# From Euler to AI: Unifying Formulas for Mathematical Constants

**Tomer Raz    Michael Shalyt    Elyasheev Leibtag    Rotem Kalisch**
**Shachar Weinbaum    Yaron Hadad    Ido Kaminer** *
Technion – Israel Institute of Technology,
Haifa 3200003, Israel

## Abstract

The constant $\pi$ has fascinated scholars throughout the centuries, inspiring numerous formulas for its evaluation, such as infinite sums and continued fractions. Despite their individual significance, many of the underlying connections among formulas remain unknown, missing unifying theories that could unveil deeper understanding. The absence of a unifying theory reflects a broader challenge across math and science: knowledge is typically accumulated through isolated discoveries, while deeper connections often remain hidden. In this work, we present an automated framework for the unification of mathematical formulas. Our system combines large language models (LLMs) for systematic formula harvesting, an LLM-code feedback loop for validation, and a novel symbolic algorithm for clustering and eventual unification. We demonstrate this methodology on the hallmark case of $\pi$, an ideal testing ground for symbolic unification. Applying this approach to 455,050 arXiv papers, we validate 385 distinct formulas for $\pi$ and prove relations between 360 (94%) of them, of which 166 (43%) can be derived from a single mathematical object—linking canonical formulas by Euler, Gauss, Brouncker, and newer ones from algorithmic discoveries by the Ramanujan Machine. Our method generalizes to other constants, including $e$, $\zeta(3)$, and Catalan's constant, demonstrating the potential of AI-assisted mathematics to uncover hidden structures and unify knowledge across domains.

## 1    Introduction

The earliest rigorous approximation for $\pi$ dates back to Archimedes around 250 BCE, who established the bounds $\frac{223}{71} < \pi < \frac{22}{7}$ [16]. Modern $\pi$ approximations employ more sophisticated formulas. For example, the Chudnovsky algorithm [15], derived from a formula by Ramanujan [48], remains instrumental for precision records. Similarly, the BBP formula [6] is notable for enabling computation of specific $\pi$ digits without requiring prior digits. Such breakthroughs inspired fundamental advances in computer science, such as high-precision arithmetic [7], evolutionary optimization [35], and elliptic curve cryptography [39]. Recent efforts led to the development of computer algorithms capable of generating numerous formula hypotheses and sometimes proofs for mathematical constants [9, 17, 46].

---

* Corresponding author: kaminer@technion.ac.il
  Project repository: https://github.com/RamanujanMachine/euler2ai

The plethora of results related to $\pi$ discovered over the centuries leads to a persistent question: How are they all connected? This question is important not only for preventing accidental rediscoveries (e.g., Lange's formula from 1999 [37] had already been derived by Lord Brouncker in 1656 [42]). Many equivalent formulas appear vastly different at first glance. A simple example is Euler's continued fraction that provides equivalent representations of infinite sums [23]. This complex situation underscores the need for a systematic approach to unify these relationships.

So far, efforts in AI for mathematics have focused on automated theorem proving [44, 56], automated conjecture generation [3, 24, 25, 40, 46], regression [31, 32, 55], and, recently, LLM–tool integrations for mathematical discovery [27, 29, 50, 59]. However, to date, no work has addressed the problem of symbolic unification of mathematical knowledge.

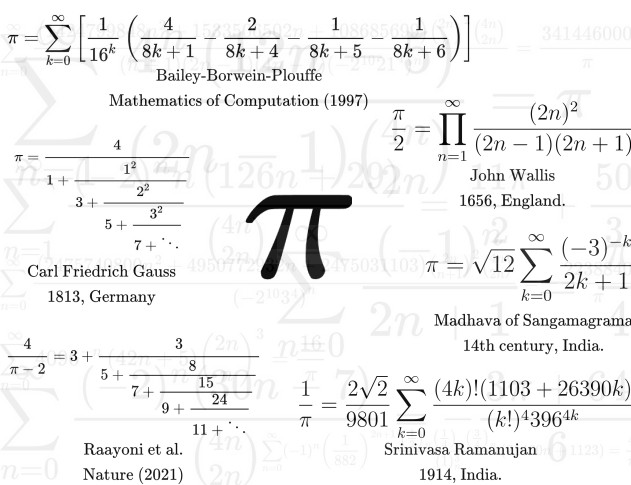

Figure 1: Selected $\pi$ formulas across centuries.

In this work, we propose a system for the large-scale harvesting, identification, and unification of mathematical formulas (Fig. 2). This effort leverages recent advances in content understanding based on large language models (LLMs), the newly discovered concept of Conservative Matrix Fields (CMFs) [21, 58], and a novel mathematical algorithm that we call UMAPS for unification via mapping across symbolic structures, using the math of coboundary equivalence for finding and proving relations between formulas. To demonstrate this methodology, we selected the symbolic case study of formulas calculating $\pi$. A total of 385 unique $\pi$ formulas were extracted from the literature and validated, of which 43% were found to correspond to different trajectories within a single CMF (Section 5). We expect near-future improvements of our algorithm to classify all the $\pi$ formulas into one or just a few unique CMFs that unify all knowledge about $\pi$ calculation.

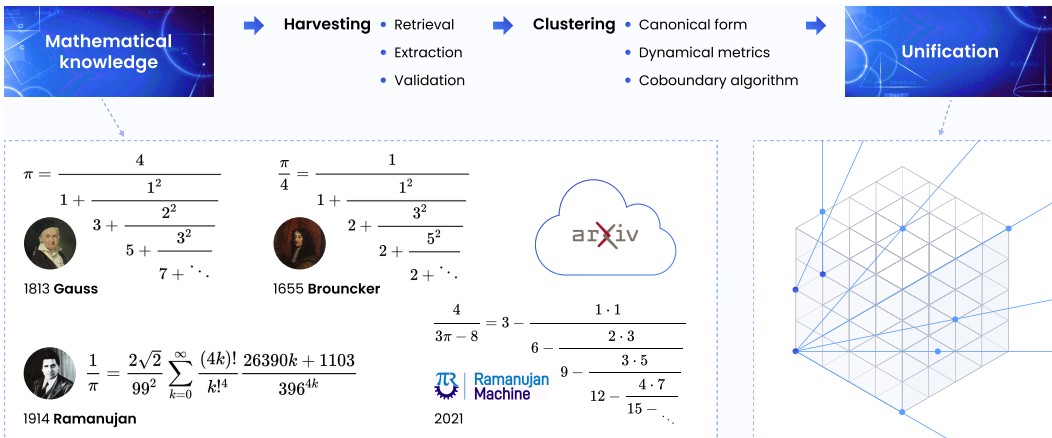

Figure 2: **Automated methodology for unifying mathematical knowledge.** A large corpus of mathematical formulas is harvested, retrieving formulas that are each translated to executable code for validation. The formulas are then clustered by conversion into their canonical forms, and unified using a novel symbolic computational algorithm that proves their relations.

To the best of our knowledge, this work is the first LLM–symbolic-tool integration for discovery in number theory, and possibly the first LLM integration with a proprietary research-grade computer algebra system. The success of this study highlights the prospects of automated unification of vast mathematical knowledge. Beyond the example of $\pi$, we applied our algorithm to other mathematical constants like $e$, $\zeta(3)$, and Catalan's constant, and to a variety of formula structures, showcasing its broad potential. Appendix A provides a glossary of key terms used throughout the paper.

## 2 Mathematical background

### 2.1 Recurrences as universal representations of formulas for mathematical constants

A wide range of formulas—including infinite sums, products, and continued fractions—can be converted into recurrences, providing a cohesive framework for unification. A function $u_n$ satisfies a recurrence of order $m$ if $u_n = a_{1,n}u_{n-1} + a_{2,n}u_{n-2} + \ldots + a_{m,n}u_{n-m}$, which can be represented via the associated companion matrix:

$$\mathrm{CM}(n) := \begin{pmatrix} 0 & 0 & \ldots & 0 & a_{m,n} \\ 1 & 0 & \ldots & 0 & a_{m-1,n} \\ 0 & 1 & \ldots & 0 & a_{m-2,n} \\ \vdots & \vdots & \ddots & \vdots & \vdots \\ 0 & 0 & \ldots & 1 & a_{1,n} \end{pmatrix} \tag{1}$$

By incrementally multiplying the companion matrix over $n$ steps, we get the matrix:

$$\prod_{i=1}^{n} \mathrm{CM}(i) = \begin{pmatrix} p_{1,n-m} & \cdots & p_{1,n-1} & p_{1,n} \\ p_{2,n-m} & \cdots & p_{2,n-1} & p_{2,n} \\ p_{3,n-m} & \cdots & p_{3,n-1} & p_{3,n} \\ \vdots & \ddots & \vdots & \vdots \\ p_{m,n-m} & \cdots & p_{m,n-1} & p_{m,n} \end{pmatrix} \tag{2}$$

$p_{1,n}, \ldots p_{m,n}$ are solutions to the recurrence for the initial conditions $p_{i,j} = \delta_i^j$. Other solutions for different initial conditions can be written as linear combinations of these.

Recurrences evaluate a desired constant $L$ either directly $\lim_{n\to\infty} u_n = L$ (e.g., for infinite sums), or as ratios $\lim_{n\to\infty} \frac{p_n}{q_n} = L$ with $p_n$ and $q_n$ being two solutions for the recurrence (e.g., for continued fractions). In the special case of a second-order recurrence, $u_n = a_n u_{n-1} + b_n u_{n-2}$, and any pair of solutions is associated with a formula in the form of a continued fraction:

$$\cfrac{b_1}{a_1 + \cfrac{b_2}{\ddots + \cfrac{b_n}{a_n}}} = \frac{p_n}{q_n} \tag{3}$$

When the functions $a_n = a(n)$ and $b_n = b(n)$ are polynomials, the formula above is known as a Polynomial Continued Fraction (PCF), denoted by $\mathrm{PCF}(a(n), b(n))$. See details in Appendix E.

### 2.2 The dynamical metrics describing each formula

A formula of a mathematical constant $L$ provides a converging sequence of rational numbers $\frac{p_n}{q_n}$ (known as a *Diophantine approximation*). The formula can be characterized by *dynamical metrics* capturing properties such as its convergence rate. A recent paper [52] proposed using such metrics for formula discovery and clustering. Here we use the metrics of *convergence rate* and *irrationality measure*. The *convergence rate* is defined as:

$$r = \lim_{n\to\infty} \frac{1}{n} \log \left| L - \frac{p_n}{q_n} \right| \tag{4}$$

When examining the connection of two candidate formulas, the ratio of their $r$ values can hint whether one is a transformation of a subsequence of the other (see Appendix E.5 for an example). The *irrationality measure* of $\frac{p_n}{q_n}$ is defined as the limit $\delta = \lim_{n\to\infty} \delta_n$, where

$$\delta_n = -1 - \frac{\log \left| L - \frac{p_n}{q_n} \right|}{\log |q_n|} \tag{5}$$

We found that two formulas sharing the same $\delta$ is the strongest indication of a possible relation, since $\delta$ is invariant under many transformations and choice of subsequences. Below, our UMAPS algorithm is used to derive and prove a relation once a pair of formulas share the same $r$ and $\delta$.

### 2.3 Conservative Matrix Fields (CMFs)

The CMF is the mathematical structure that generalizes formulas of a particular constant, originally found by generalizing PCFs [21], and later realized to be more general (Appendix G). To exemplify the

concept, we focus on the CMF of $\pi$. This CMF is 3D, i.e., consisting of three matrices $M_{\mathbf{x}}, M_{\mathbf{y}}, M_{\mathbf{z}}$ with rational function entries in the variables $(x, y, z)$, satisfying:

$$M_{\mathbf{x}}(x, y, z)M_{\mathbf{y}}(x + 1, y, z) = M_{\mathbf{y}}(x, y, z)M_{\mathbf{x}}(x, y + 1, z)$$
$$M_{\mathbf{x}}(x, y, z)M_{\mathbf{z}}(x + 1, y, z) = M_{\mathbf{z}}(x, y, z)M_{\mathbf{x}}(x, y, z + 1)$$
$$M_{\mathbf{y}}(x, y, z)M_{\mathbf{z}}(x, y + 1, z) = M_{\mathbf{z}}(x, y, z)M_{\mathbf{y}}(x, y, z + 1)$$

This property describes the path-independence of the transition between two points in a 3D lattice (lattice illustrated in Fig. 5b), in analogy to a conservative vector field. The CMF satisfies the properties of a discrete flat connection [11]. For an explanation of how formulas reside as directions within the CMF, see Appendix G.1. A notable feature of the CMF is that pairs of formulas found to be parallel trajectories with different initial points correspond to two matrices that are coboundary equivalent.

Many of the known $\pi$ formulas will be shown to reside within a single CMF (details in Appendix G.2):

$$M_{\mathbf{x}} = \begin{pmatrix} 1 & y \\ \frac{1}{x} & \frac{2x+y-2z+2}{x} \end{pmatrix} \quad M_{\mathbf{y}} = \begin{pmatrix} 1 & x \\ \frac{1}{y} & \frac{x+2y-2z+2}{y} \end{pmatrix} \quad M_{\mathbf{z}} = \begin{pmatrix} \frac{z(-x-y+z)}{(y-z)(x-z)} & \frac{zxy}{(y-z)(x-z)} \\ \frac{z}{(y-z)(x-z)} & \frac{-z^2}{(y-z)(x-z)} \end{pmatrix} \quad (6)$$

## 3 Methodology for symbolic unification of formulas

### 3.1 Harvesting: large-scale retrieval of formulas from the literature

The first challenge lies in the natural language processing of formulas. We analyze the LaTeX source code of 455,050 arXiv articles by combining regular expressions and LLMs, extracting all mathematical expressions, resulting in 278,242,506 strings. Filtering for expressions containing the $\pi$ symbol retrieves 121,662 $\pi$-related equations. The widespread use of the symbol $\pi$ in scientific literature means that most occurrences are unrelated to calculating the constant itself. To address this, and keeping in mind that there is a priori very little data on what successful formulas' LaTeX looks like, each potential formula is classified as computing $\pi$ or not, using GPT-4o mini [41] (chosen for its cost-effectiveness), reducing the number of candidates to 3367. Next, GPT-4o categorizes formulas by type: series, continued fraction, or neither—resulting in 1656 formula candidates.

### 3.2 Harvesting: extraction and validation

The extraction and validation stages rely on an LLM-code feedback loop that feeds a PSLQ algorithm. Each equation, represented as a LaTeX string, must then be parsed into a Computer Algebra System (CAS) for further manipulations (in our case, SymPy [38]). Automatically extracting algebraic forms from LaTeX strings is especially complicated due to varied LaTeX patterns, which are difficult

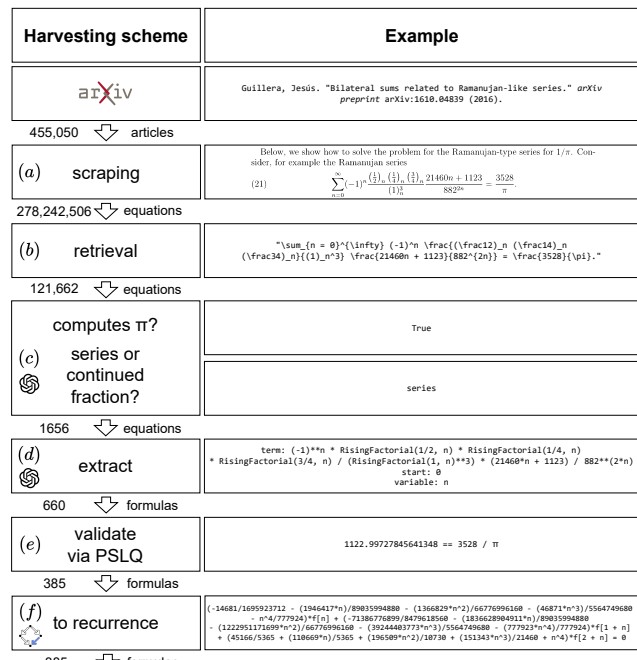

Figure 3: **Pipeline for automated harvesting of mathematical formulas (left), exemplified using one of the analyzed $\pi$ formulas (right)**. (a) Equations are scraped from papers on arXiv. (b) Regular expressions on the LaTeX strings retrieve series and continued fraction patterns that contain $\pi$ as the only irrational number (see Appendix I.3). (c) Zero-shot classification using OpenAI's GPT-4o mini identifies formulas calculating the constant $\pi$. Then, OpenAI's GPT-4o identifies the formula type (series, continued fraction, or neither). (d) Extraction of the series' summand or the continued fraction's partial numerator and partial denominator, using GPT-4o. The formula is then converted to code. (e) Formulas are computed and validated using the integer relation finder algorithm PSLQ. (f) The formulas are converted to canonical recurrences using RISC's tool for fitting recurrences [33].

to systematically convert to executable code using a predefined logic. LLMs help us overcome these obstacles by processing text contextually and attending to relevant parts of the text, solving the natural language processing task that may have required elaborate rules [14, 47]. Specifically, we use OpenAI's GPT-4o to translate relevant LaTeX into executable mathematical code [22, 43, 61] (see Appendix I for the exact prompts used). To correct for (common) mistakes in the LLM-generated formula code, we apply an LLM-code feedback loop for code validation: errors are sent back to the LLM along with the faulty code to correct it, up to three times (see Appendix I.6.3).

We validate that each extracted formula computes the constant $\pi$ by running the formula code to get a numerical approximation and then applying PSLQ, an integer relation algorithm [26]. Limit values are not extracted directly from the LaTeX string for validation, since we found that GPT-4o got them wrong in some cases (see Table 14). Instead, the PSLQ approach fixes these critical GPT mistakes and reproduces the intended formulas. Out of the 660 candidates, 385 were validated as $\pi$ formulas and passed on for canonicalization (details in Appendix I.5).

### 3.3 Clustering: using the canonical form

The first unification step is converting each formula to its canonical form: the simplest linear recurrence with polynomial coefficients (Appendix E.4.1). Automated algebraic capabilities are unpredictable in solving such tasks. Thus, we use a computational method for converting the formulas to polynomial recurrences: a Mathematica package by RISC [33] that fits polynomial-coefficient linear recurrences to each sequence of rational numbers. The resulting recurrences are validated numerically and passed to a Maple package to guarantee order minimality [57, 60], thus finding the provably minimal polynomial recurrence. Out of the 385 validated formulas (Section 3.1), 380 are found to have representations as order-2 recurrences, and 5 as order-3 recurrences, which can also be addressed as we show in Appendix B.6 and Appendix C.3.

The same canonical form captures a wide range of formulas, continued fractions and infinite sums. Thus, the conversion to canonical forms automatically unifies different formulas, yielding 149 different order-2 canonical forms and 4 order-3 canonical forms for $\pi$, 153 in total, from 385 formulas (selected examples in Table 1).

Table 1: **Canonical form representation**. Converting formulas to their canonical forms shows equivalence of different-looking expressions (e.g. 1,2), leaving the less-trivial connections for the later steps of the algorithm. Additional details in Appendix C.4.

| | Formula | Value | arXiv source | Canonical form (CF) | CF value | Initial conditions |
|---|---|---|---|---|---|---|
| 1 | $\sum_{n=0}^{\infty} \frac{n!}{\prod_{i=1}^{n}(2i+1)}$ | $\frac{\pi}{2}$ | 1806.03346 | $\mathrm{PCF}(3n+1, n(1-2n))$ | $\frac{2}{\pi}$ | $\begin{bmatrix} 0 & 1 \\ 1 & 1 \end{bmatrix}$ |
| 2 | $\sum_{n=1}^{\infty} \frac{2^n}{n\binom{2n}{n}}$ | $\frac{\pi}{2}$ | 2010.05610 | $\mathrm{PCF}(3n+1, n(1-2n))$ | $\frac{2}{\pi}$ | $\begin{bmatrix} 0 & 1 \\ 1 & 1 \end{bmatrix}$ |
| 3 | $\sum_{n=0}^{\infty} \frac{(-1)^n}{2n+1}$ | $\frac{\pi}{4}$ | 2404.15210 | $\mathrm{PCF}(2, (2n-1)^2)$ | $1+\frac{4}{\pi}$ | $\begin{bmatrix} 0 & 1 \\ 1 & 1 \end{bmatrix}$ |
| 4 | $\sum_{n=1}^{\infty} \frac{(-1)^{n+1}}{n(n+1)(2n+1)}$ | $\pi-3$ | 2206.07174 | $\mathrm{PCF}(6, (2n+1)^2)$ | $\frac{1}{\pi-3}$ | $\begin{bmatrix} 0 & 1 \\ 1 & 6 \end{bmatrix}$ |
| 5 | $\sum_{n=1}^{\infty} \frac{4^n(12n-5)}{(2n-1)\binom{4n}{2n}}$ | $\frac{3\pi+4}{2}$ | 2204.08275 | * | $\frac{-42\pi-196}{3\pi+4}$ | $\begin{bmatrix} 0 & 70 \\ -1 & 15 \end{bmatrix}$ |

\* $\mathrm{PCF}(240n^3 + 164n^2 - 54n - 29, -9216n^6 + 12288n^5 + 11264n^4 - 15520n^3 - 764n^2 + 3802n - 714)$

### 3.4 Clustering: using the dynamical metrics

The clustering stage is a heuristic to guide which formulas should be attempted to be proven equal using UMAPS. Formulas with the same metrics are likely to be related to the same constant [52]. The metrics also indicate a more intricate connection, enabling the unification of formulas in a systematic way that proves an analytical transformation between them. Canonical-form formulas are first compared to each other using the irrationality measure $\delta$ (Fig. 4a), which is the most reliable indicator for a potential equivalence. Every new formula is first evaluated relative to directions in the CMF corresponding to recurrences with the same $\delta$. This search can be improved by using gradient descent on the direction parameters, because $\delta$ is found to be continuous [21].

We found that $\delta$ is not sufficient to imply equivalence, and thus we complement it using the ratio of convergence rates $r_A : r_B$. Canonical form A is *folded* (Appendix E.5) by $r_B$ and canonical form B is folded by $r_A$ (Fig. 4b), making them converge at the same rate. The next step is finding their precise algebraic relation using UMAPS.

## 3.5 Unification: using the UMAPS algorithm for coboundary equivalence

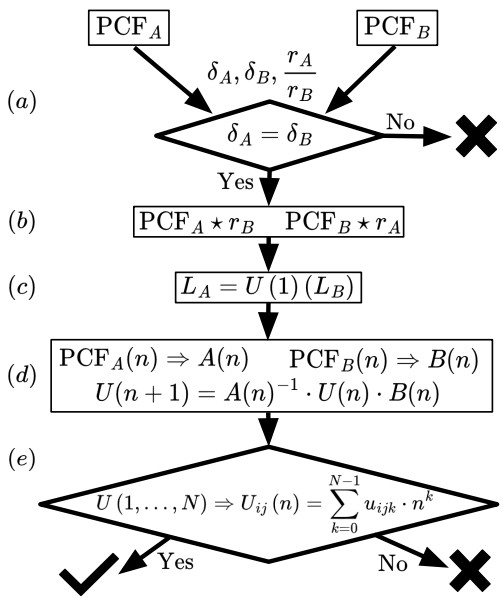

Figure 4: **The matching algorithm: connecting polynomial linear recurrences.** This algorithm is demonstrated here for polynomial continued fractions (PCFs) but can be generalized to any linear polynomial recurrence. (a) Compute the dynamical metrics [52] for the two PCFs (irrationality measures $\delta_A$, $\delta_B$ and the convergence rates ratio $r_A/r_B$). The $\delta$ metrics are used to identify possible connections, as only if $\delta_A = \delta_B$, the PCFs can be related via coboundary (in practice, we test for them to be within 0.06 of each other). (b) *Fold* $\mathrm{PCF}_A$ by $r_B$ and $\mathrm{PCF}_B$ by $r_A$ (Appendix E.5). **UMAPS (c)-(e):** (c) Solve for a general Möbius transform (a $2\times 2$ matrix $U(1)$) that once applied to the limit of $\mathrm{PCF}_B$ equates it to the limit of $\mathrm{PCF}_A$. (d) Representing the PCFs in matrix form ($A(n)$ and $B(n)$), propagate the coboundary matrix via the relation $U(n+1) = A(n)^{-1} \cdot U(n) \cdot B(n)$ up to $U(N)$ ($N = 40$ was sufficient for our runs, see Appendix D). (e) Assume the general form of $U(n)$ to have rational-function entries with polynomial degree up to $\left\lfloor \frac{N-1}{2} \right\rfloor$ and solve for their coefficients using normalized $U(1, \ldots, N)$. If such a solution is found and validated, the PCFs are coboundary-related. See Appendix C for more details.

Our algorithm for unification via mapping across symbolic structures (UMAPS) relies on the established concept of coboundary equivalence (Appendix E.4), however, no specialized coboundary solver existed prior to this work.

$A(n), B(n) \in \mathrm{PGL}_m(\mathbb{Q}(n))$ are *coboundary equivalent* if there exist a matrix $U(n)$ such that

$$A(n) \cdot U(n+1) = U(n) \cdot B(n) \qquad (7)$$

This definition carries to recurrences when their companion matrices (Eq. (1)) are coboundary equivalent (Fig. 5a,d) and then: $\left(\prod_{i=1}^{n} A(i)\right) \cdot U(n+1) = U(1) \cdot \left(\prod_{i=1}^{n} B(i)\right)$.

Since any matrix with rational function coefficients can be scaled to have polynomial coefficients, we can write that $A(n), B(n) \in \mathrm{GL}_m(\mathbb{Q}[n])$ are *coboundary equivalent* if there exist a matrix $U(n) \in \mathrm{GL}_m(\mathbb{Q}[n])$ and polynomials $p_A(n), p_B(n) \in \mathbb{Q}[n]$ such that

$$p_A(n) \cdot A(n) \cdot U(n+1) = p_B(n) \cdot U(n) \cdot B(n) \quad (8)$$

Finding a coboundary between two polynomial matrices is inherently a non-linear problem due to the product of unknown polynomials $p_A$ and $p_B$ with unknown coboundary matrix $U$. Moreover, the degree of each polynomial is not known. Despite the non-linearity, we found a coboundary solver algorithm for general order $m$ (Appendix C.3).

UMAPS finds the solution without solving non-linear equations, instead leveraging the recurrence limits to compute a sequence of empirical coboundary matrices, whose elements are fitted to rational functions [53]. The algorithm relies on the following lemma:

**Lemma 1.** *(A necessary condition on the coboundary equivalence matrix.) Let $L_A = \lim_{n \to \infty} PCF(a(n), b(n))$ and $L_B = \lim_{n \to \infty} PCF(c(n), d(n))$ be converging PCFs with associated companion matrices $A(n), B(n) \in \mathrm{PGL}_2(\mathbb{Q}(n))$. If $A(n)$ is coboundary to $B(n)$, then $L_A$ and $L_B$ are related through a rational Möbius transformation. Moreover, if $U(n)$ is the coboundary matrix, then $L_A = U(1)(L_B)$ ($U(1)$ applied to $L_B$ as a Möbius transformation).*

A proof and generalization to higher-order recurrences (Lemma 4), as well as a proof of the uniqueness of the coboundary matrix (Lemma 5), are detailed in Appendix F. These combine to show that UMAPS is sufficient to solve for the coboundary matrix, as stated in Corollary 1 (proof in Appendix C.3).

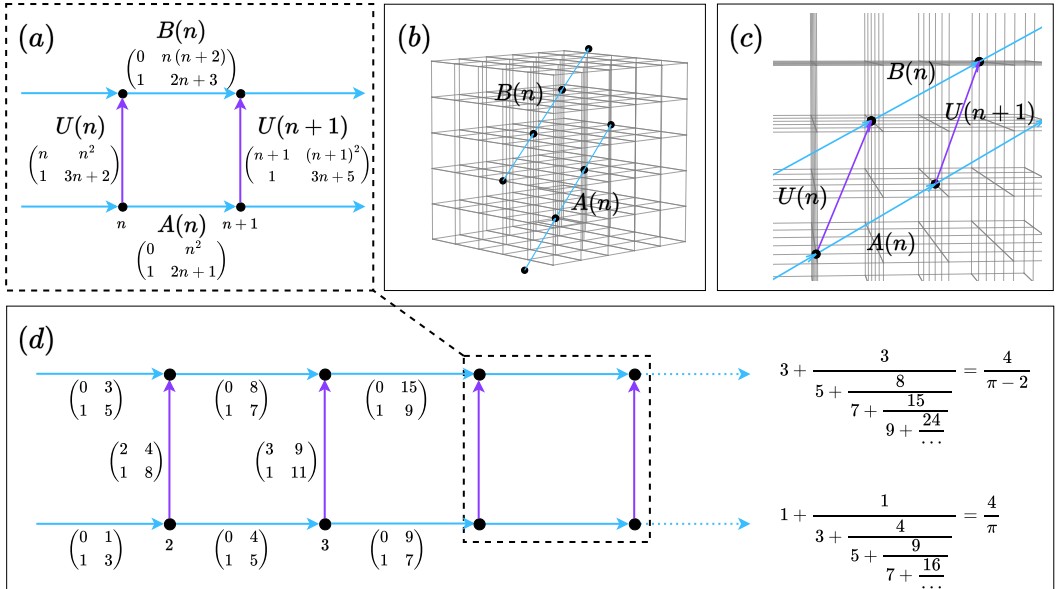

Figure 5: **Coboundary equivalence**: the mathematical framework connecting different formulas once cast into their canonical forms. (a) The coboundary condition $A(n) \cdot U(n+1) = U(n) \cdot B(n)$ recasts formulas as (b,c) parallel trajectories in a CMF. (d) Example of two coboundary-equivalent formulas, presenting their coboundary matrices and limits, which constitute proof of a novel equivalence.

**Corollary 1.** *(Sufficiency of UMAPS.) If a coboundary matrix exists for two matrices and every rational-function entry of the coboundary matrix has polynomials of degree at most $d$, then running UMAPS with $N \geq 2d+1$ suffices to recover the coboundary matrix.*

Fig. 4 summarizes the flow of matching two canonical form formulas. Using this method, we find that formulas 1,2 and 5 from Table 1 are equivalent and that formulas 3,4 are also equivalent. Refer to Appendix B.1 for descriptions of how the algorithm is applied to these formulas, and refer to Appendix C for a listing of the algorithms involved. A study of the algorithms' sensitivity to hyperparameters is provided in Appendix D. The same procedure is applied to each canonical form formula, measuring its $\delta$ value and relying on its continuity as a function of direction in the CMF to locate worthy directions that yield potential formula pairs for the coboundary algorithm. The matching algorithm is then applied between formulas and representative recurrences from the CMF. Finding a match between a formula and a CMF representative proves the formula is generated by the CMF. The full list of results is provided in Appendix J. Selected results for $\pi$ are detailed in Section 5.

## 4 Benchmarking

### 4.1 Comparison to other methods for symbolic unification

Our work is the first to address the problem of symbolic unification at scale, thus there are no standard benchmarks for performance comparison. Leading LLMs are generally unable to address the full challenge. As an example, we compare our equivalence detection and proving capabilities to those of LLMs: We tasked 2 leading LLMs—GPT-4o and Gemini 2.5 Pro Preview—with identifying and proving 10 formula-pair equivalences proven by our algorithm (Table 2). The formulas are chosen such that each pair has equal dynamical metrics $(r, \delta)$ after folding, which is the simpler situation to prove (parallel trajectories in the CMF). Even with these simpler tasks, the LLMs exhibit only limited success. We did not find cases in which any LLM succeeded in finding relations between a pair of formulas without equal dynamical metrics.

### 4.2 Comparison of LLM model performance

We utilize LLMs for classification and extraction in two different ways. Table 3 compares the performance of three choices for the extractor LLM, which we found to be the more sensitive choice, as it is used for the more advanced LLM-code feedback loop. A ground truth is established by merging the validated formulas (Section 3.2) found by the three compared LLMs.

Table 2: LLM performance when detecting and proving equivalence in a random set of 10 formula pairs of equal dynamical parameters (Appendix H). All LLM proofs were validated manually.

| LLM | Successful detections | Correct proofs |
|---|---|---|
| GPT-4o | 1/10 | 2/10 |
| Gemini 2.5 Pro Preview | 8/10 | 5/10 |

Table 3: Performance of different extractor LLM choices in terms of successfully harvested formulas. The LLM errors are split to "faulty code" for code that did not run, and "symbolic mistake" for incorrect identification of some of the formula constituents like continued fraction polynomials. The bold row marks the choice of LLMs used for all the rest of the results in this paper.

| LLM classifier | LLM extractor | Successful extractions | Faulty code | Symbolic mistake |
|---|---|---|---|---|
| **GPT-4o mini** | **GPT-4o** | **289 (97.6%)** | **2 (0.7%)** | **5 (1.7%)** |
| GPT-4o mini | Claude 3.7 Sonnet | 266 (89.9%) | 21 (7.1%) | 9 (3.0%) |
| GPT-4o mini | GPT-4o mini | 206 (69.6%) | 70 (23.6%) | 20 (6.8%) |

## 5 Results

### 5.1 Example equivalences among famous formulas

Our automated system proves previously unknown equivalences between formulas. Among the formulas connected are famous examples, such as one of Ramanujan's 1914 formulas, as well as Lord Brouncker's, Euler's, and Gauss's PCFs from the 17th, 18th, and 19th centuries [23, 28, 42]. For example, the following series found by Ramanujan in 1914 [48],

$$\frac{4}{\pi} = \sum_{k=0}^{\infty} (-1)^k \frac{\left(\frac{1}{2}\right)_k \left(\frac{1}{4}\right)_k \left(\frac{3}{4}\right)_k}{(1)_k^3} \cdot (1123 + 21460k) \cdot \left(\frac{1}{882}\right)^{2k+1} \tag{9}$$

was proven equivalent (Appendix B.4) to a newer series from a paper published in 2020 [54]:

$$\frac{341446000}{\pi} = \sum_{k=0}^{\infty} \frac{\binom{2k}{k}^2 \binom{4k}{2k}}{(k+1)(2k-1)(4k-1)(-2^{10}21^4)^k} \cdot \left(1424799848k^2 + 1533506502k + 1086885699\right) \tag{10}$$

This equivalence demonstrates how two previously distinct mathematical expressions, discovered over a century apart, are now proven to be equivalent by an automated process.

Fig. 5d proves the equivalence of another pair of famous formulas: (1) $\text{PCF}(2n+3, n(n+2))$, one of the first computer-discovered $\pi$ formulas from 2021 [46]. (2) $\text{PCF}(2n+1, n^2)$, published by Gauss in 1813 [28] and provided at the time an especially efficient way to calculate digits of $\pi$.

### 5.2 Formulas unified by a Conservative Matrix Field (CMF)

The CMF of $\pi$, Eq. (6), captures most of the harvested formulas (Table 4), with selected examples presented graphically in Fig. 6 along with their corresponding trajectories.

Table 4: **Summary of unification results**, among all validated formulas (left columns) and among the canonical forms (right columns). Formulas are harvested from 140 arXiv papers (Table 15), of which 137/140 (98%) have at least one formula proved connected by UMAPS and 70/140 (50%) have a formula residing in the same CMF.

| Found relation | Same CMF | Found relation (canonical) | Same CMF (canonical) |
|---|---|---|---|
| 360/385 (94%) | 166/385 (43%) | 136/153 (89%) | 81/153 (53%) |

The full list of canonical forms captured by the CMF appears in Table 16. Improvements in UMAPS are likely to connect additional formulas (Table 17) to the same CMF.

### 5.3 Unification of formulas beyond $\pi$

Going beyond $\pi$, we automatically identified equivalent formulas for $e$, $\zeta(3)$, and Catalan's constant—demonstrating the generality of the approach. Consider these two formulas for Apéry's constant, the Riemann zeta function value $\zeta(3)$:

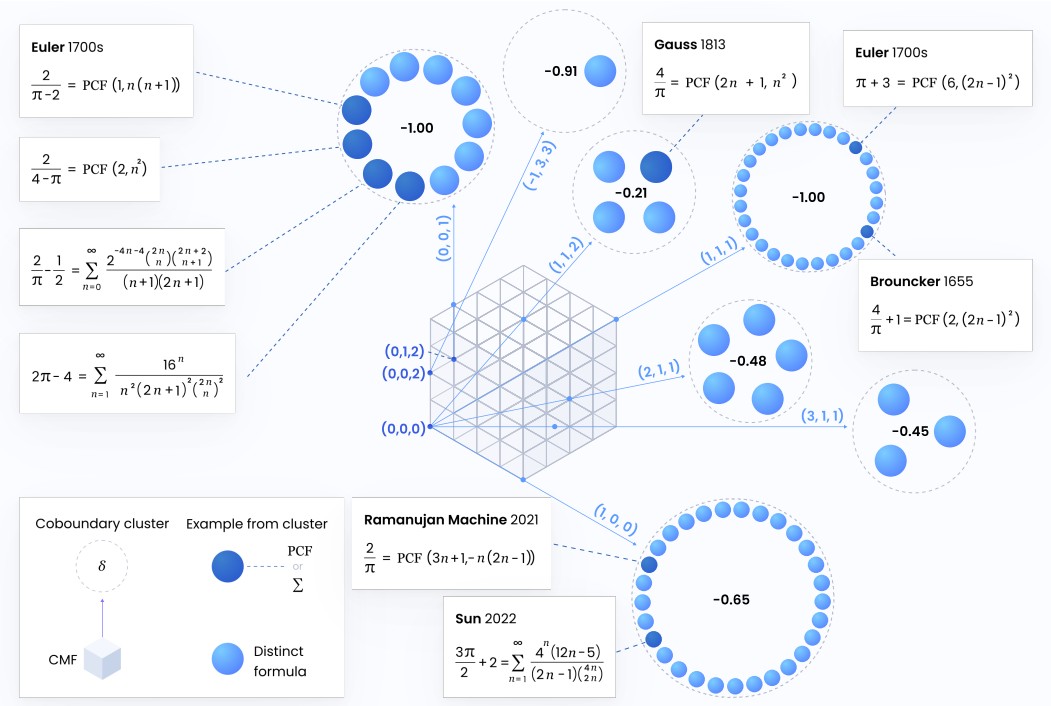

Figure 6: **Formula unification by a Conservative Matrix Field (CMF)**. Numerous $\pi$ formulas harvested from the literature are automatically arranged as trajectories in a 3D CMF. These formulas include famous ones by Gauss, Euler, and Lord Brouncker. The full list of unified formulas and their canonical forms is available in Table 16. Each cluster (large dashed circles) denotes formulas connected by coboundary, representing parallel trajectories or overlapping trajectories. The number at each cluster center is the $\delta$ of all formulas in that cluster. Arrows show trajectory directions. Note that multiple formula clusters can have the same $\delta$ value without being coboundary, showing that sharing $\delta$ is necessary but not sufficient for formulas being coboundary-related.

$$\zeta(3) = \sum_{n=1}^{\infty} \frac{1}{n^3} \qquad\qquad \frac{5}{4} - \zeta(3) = \sum_{n=2}^{\infty} \frac{1}{n^3(n^2-1)} \tag{11}$$

The second formula [36] has faster convergence compared to the classical definition of $\zeta(3)$, though both converge polynomially. Our automatic procedure proves their equivalence by the coboundary transform and unifies them in the $\zeta(3)$ CMF (detailed in Appendix B.2).

As another example, the following two PCFs for Catalan's constant [13] are also proven equivalent by UMAPS (Appendix B.5).

$$\frac{1}{2-2G} = 3 + \cfrac{2^2}{1 + \cfrac{2^2}{3 + \cfrac{4^2}{1 + \cfrac{4^2}{\dots}}}} \qquad\qquad \frac{1}{2G-1} = \frac{1}{2} + \cfrac{1^2}{\frac{1}{2} + \cfrac{1 \cdot 2}{\frac{1}{2} + \cfrac{2^2}{\frac{1}{2} + \cfrac{2 \cdot 3}{\dots}}}} \tag{12}$$

A natural next step is to conduct exhaustive searches for other well-known constants, and fundamental mathematical structures in fields such as physics and computer science. See Appendix B.3 for $e$ examples.

## 5.4 Formulas generated via a Conservative Matrix Field (CMF)

A sample of 1693 distinct $\pi$ formulas was generated from the $\pi$-CMF (per Appendix C.5). The CMF permits a new method of comparing between formulas, using the *normalized convergence rate*, defined $r/\ell^1(t)$, where $r$ is the convergence rate from Eq. (4), $t$ is a trajectory and $\ell^1$ is the $\ell^1$-norm. 57 of the formulas generated in our runs shared the best normalized $r$ of 1.76, such as the formula

$$\frac{10}{4-\pi} = 12 + \cfrac{-81}{238 + \cfrac{-6500}{968 + \cfrac{-67473}{\ddots + \cfrac{n^2(-64n^4 - 96n^3 + 12n^2 + 52n + 15)}{48n^3 + 108n^2 + 70n + 12 + \ddots}}}}$$

arising from trajectory $(-1, -1, 0)$. By comparison, the best pre-existing formula unified by our CMF has normalized convergence of 0.88 (the $(1, 1, 2)$ direction, see Table 16). Details in Appendix G.6.

# 6 Discussion

## 6.1 Limitations

Currently, the harvesting step relies on the LLM's ability to interpret and contextualize mathematical LaTeX strings. This step likely introduces data loss and false negatives in formula classification. Improvements in prompt engineering and in validation techniques will enhance the robustness of this LLM use. As more advanced LLMs become available, this step will become increasingly reliable.

Formulas often include additional symbols other than summation indices, like variables defined in the text surrounding the formula, which should be extracted and substituted into formula evaluation. We made several such substitutions manually to test the rest of the pipeline for these special cases. Future improvements of the unification pipeline can address this limitation by more advanced use of LLMs and automated validation.

Most formulas analyzed in this work are series or continued fractions. However, UMAPS and all the other steps in our harvesting and clustering processes are applicable more broadly (to any formula that generates a sequence of rational approximations for a given constant, e.g., deeper recurrences). Expanding the system to accommodate additional cases is a promising direction for future work.

The same unification pipeline shown here could apply to the vast family of constants derived from D-finite functions by finding their corresponding CMFs [58].

## 6.2 Outlook

Increasing the dimension and rank of the $\pi$-CMF, along with further improvements to UMAPS [2], is likely to yield a higher percentage of unified formulas in the near future. A planned future study will employ the CMF to systematically search for fast-converging and irrationality-proving formulas.

Looking forward, the same approach of collection, analysis, and organization of mathematical knowledge could help establish rigorous connections between different branches of mathematics. The methodology presented in this work could help develop more general frameworks for identifying connections between different scientific theories through their mathematical representations. As the volume of information grows at an accelerating pace, finding automated ways to unify knowledge will become increasingly essential, especially with the goal of providing more intuitive understanding on complex concepts.

Pairing LLMs with pre-existing and novel tools for symbolic and numerical mathematics enabled the automated discoveries in this paper. We believe this LLM–tool integration scheme will continue to advance AI for mathematics and science in the coming years.

## Acknowledgments

This research received support through Schmidt Sciences, LLC.

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

# Appendix

## A  Glossary

**Canonical form** – The simplest polynomial linear recurrence that generates a given formula. It is found by running RISC's Mathematica package [33], then validating numerically, and finally minimizing the recurrence with a Maple package [57].

**Coboundary equivalence** – An equivalence between two matrices via a third matrix called the *coboundary matrix*. For two matrices $A(n), B(n) \in \mathrm{PGL}_m(\mathbb{Q}(n))$ the equivalence takes the form $A(n) \cdot U(n+1) = U(n) \cdot B(n)$. In this study we combine a novel coboundary solver (UMAPS) with folding to create the matching algorithm (Appendix C.2), which finds equivalences between formulas.

**Coboundary transformation** – A transformation of the form $A(n) \to U^{-1}(n) \cdot A(n) \cdot U(n+1)$, for $A(n), U(n) \in \mathrm{PGL}_m(\mathbb{Q}(n))$.

**Companion matrix** – A matrix with a structure suitable for directly representing a linear recurrence, see Eq. (1).

**Conservative Matrix Field (CMF)** – A recently discovered mathematical structure consisting of a lattice of matrices that enables unification of many formulas discovered over the years. Originally discovered in [20] as a result of a large-scale experimental mathematics effort in formula generation, the structure was found to be generalizable to any D-finite function in [58].

**CMF dimension** – The number of parameterized matrices composing the CMF (3 for the $\pi$-CMF).

**CMF rank** – The dimension of the CMF matrices (number of rows, equal to the number of columns; 2 for the $\pi$-CMF).

**Convergence rate (r)** – The exponential convergence rate of a formula, measured empirically in this study, see Eq. (4). Used in the formula matching algorithm to ascertain the folds needed to find an equivalence between two formulas.

**Dynamical metrics** – Numerical values derived from formulas, used to cluster and find relations between the formulas (see convergence rate $r$ and irrationality measure $\delta$) [52].

**Extraction** – The stage in the formula harvesting pipeline translating formulas given in LaTeX into their components, resulting in executable SymPy code.

**Folding** – A fold by $k$ is a transformation mapping a matrix $M(n) \in \mathrm{PGL}_m(\mathbb{Q}(n))$ to the matrix $\Pi_{i=1}^{k} M(k(n-1)+i)$. In the context of recurrences, the new matrix represents taking $k$ steps of the original recurrence at a time.

**Irrationality measure ($\delta$)** – The irrationality measure of a formula, measured empirically in this study, see Eq. (5). $\delta$ is used in the matching algorithm to find candidate pairs for matching formulas, as $\delta$ is invariant under coboundary transformations. The result of this clustering phase was a 20-fold decrease in runtime, see Appendix D.

**Polynomial Continued Fraction (PCF)** – An equivalent representation of second-order recurrences with polynomial coefficients. Any second-order recurrence generating a mathematical constant can be represented as a companion matrix (Lemma 2), and thus as a continued fraction.

**PSLQ** – A numerical integer relation algorithm that finds integer coefficients $(a_i)$ such that $a_1 x_1 + a_2 x_2 + \cdots + a_n x_n = 0$ for given real numbers $(x_i)$, revealing algebraic relationships among them [9, 26].

**Retrieval** – The stage of the formula harvesting pipeline in which regular expressions are applied to filter for equations that may compute the constant of interest.

**Scraping** – The first stage of the formula harvesting pipeline.

**Trajectory** – A direction in a CMF, a vector of integers with length equal to the CMF dimension.

**Trajectory matrix** – The parameterized matrix resulting from walking in a trajectory in a CMF. Every trajectory matrix corresponds to a polynomial linear recurrence, which has a canonical form. Matches between the canonical forms of trajectory matrices and the canonical forms representing formulas can thus be searched. These matrices act as representatives of the CMF, enabling unification of formulas via the CMF.

**UMAPS** – Our novel coboundary-solving algorithm.

**Unification** – Embedding formulas in a single CMF to show there is an underlying theory behind them, the main goal of the work.

**Validation** – The stage of the harvesting pipeline after extraction in which the limits of formulas are recovered in terms of the constant of interest by running PSLQ on a numerical evaluation of the formula. This stage has two roles: (1) recovering the limits and (2) validating convergence to the constant of interest (or a Möbius transformation thereof).

# B  Special results

The examples shown below were found completely automatically via the algorithms discussed in Section 3 and organized in Appendix C. The actions at each step are described as if being applied by a human for clarity, but we emphasize that the connections were made automatically. A template for proofs could have been written once and filled in by a computer as it executed the algorithms. Some interactive examples are available at the online algorithm demonstration [4].

## B.1  Unification by the $\pi$ Conservative Matrix Field (CMF)

### B.1.1  Unification example: formulas 1, 2 and 5 from Table 1

Formulas 1 and 2 have the same canonical form, showing they are equivalent. We will next prove formulas 1,5 equivalent via the matching algorithm (Appendix C.2). We compute the irrationality measure ($\delta$) for formulas 1 and 5, resulting in $-0.65$ for both canonical forms. Next, we compute convergence rates ($r$), resulting in $0.69$ and $1.38$ for formulas 1 and 5, respectively. Dividing the two we get $\frac{r_1}{r_5} = \frac{1}{2}$. Folding canonical form 1 by 2 and canonical form 5 by 1 (meaning no change) results in

$$\text{PCF}_{1,\text{folded}} = \text{PCF}\left(60n^3 + 34n^2 - 11n - 3, 2n\left(-288n^5 + 624n^4 - 230n^3 - 225n^2 + 158n - 24\right)\right)$$

Applying UMAPS (Appendix C.3), the following coboundary matrix and polynomials connecting $\text{PCF}_{1,\text{folded}}$ and $\text{PCF}_5$ are discovered:

$$U(n) = \begin{pmatrix} 48n^3 - 85n^2 + 28n & 2304n^6 - 9792n^5 + 15440n^4 - 11100n^3 + 3586n^2 - 408n \\ -1 & -48n^3 + 200n^2 - 223n + 51 \end{pmatrix}$$

$$p_A(n) = 12n - 17$$
$$p_B(n) = 3n + 2$$

This means that the coboundary condition (Eq. (8)) holds:

$$p_A(n) \cdot \text{CM}_{1,\text{folded}} \cdot U(n+1) = p_B(n) \cdot U(n) \cdot \text{CM}_5$$

Multiplying out the terms, we indeed get the expected relation:

$$(12n - 17) \cdot \begin{pmatrix} 0 & 2n\left(-288n^5 + 624n^4 - 230n^3 - 225n^2 + 158n - 24\right) \\ 1 & 60n^3 + 34n^2 - 11n - 3 \end{pmatrix} \cdot$$
$$\cdot \begin{pmatrix} 48n^3 + 59n^2 + 2n - 9 & 2304n^6 + 4032n^5 + 1040n^4 - 1180n^3 - 434n^2 + 88n + 30 \\ -1 & -48n^3 + 56n^2 + 33n - 20 \end{pmatrix}$$
$$=$$
$$(3n + 2) \cdot \begin{pmatrix} 48n^3 - 85n^2 + 28n & 2304n^6 - 9792n^5 + 15440n^4 - 11100n^3 + 3586n^2 - 408n \\ -1 & -48n^3 + 200n^2 - 223n + 51 \end{pmatrix} \cdot$$
$$\cdot \begin{pmatrix} 0 & -9216n^6 + 12288n^5 + 11264n^4 - 15520n^3 - 764n^2 + 3802n - 714 \\ 1 & 240n^3 + 164n^2 - 54n - 29 \end{pmatrix}$$

We have therefore found a transformation from one canonical form to the other:
Coboundary( Fold(PCF$_1$, 2) ) = Fold(PCF$_5$, 1). Formulas 1,2 and 5 are equivalent—having a proof for any one formula out of these three immediately proves the other two.

Are these formulas contained in the CMF, Eq. (6)? We find that the recurrence of trajectory $(1, 0, 0)$ in the CMF also has $\delta = -0.65$. The recurrence generated by this direction is precisely the recurrence $\text{PCF}_1$, meaning a trivial coboundary equivalence $(U(n) = I_{2 \times 2})$ exists between formulas 1,2 and the CMF. Note that in the general case, showing unification requires finding a nontrivial coboundary equivalence $(U(n) \neq I_{2 \times 2})$ between a representative canonical form of the equivalent formulas, and the canonical form of the recurrence generated by the CMF (per Appendix C.1). See Appendix B.1.3 for an example.

### B.1.2  Unification example: formulas 3 and 4 from Table 1

The canonical forms of formulas 3 and 4 have $\delta = -1$ and convergence rates 0 (they converge slowly). Given the similarity in $\delta$, we conjecture that the formulas are coboundary. Applying UMAPS yields the coboundary matrix:

$$U(n) = \begin{pmatrix} 4n^2 - 4n + 1 & 8n^3 + 4n^2 - 10n + 3 \\ 2n + 1 & 4n^2 + 8n + 7 \end{pmatrix}$$

and trivial external polynomials - $p_A(n) = 1, p_B(n) = 1$. So formulas 3 and 4 are equivalent. The trajectory $(1, 1, 1)$ of the CMF generates a recurrence with $\delta = -1$ and upon inspection is found to be precisely the canonical form of formula 3, $\text{PCF}_3$, unifying formulas 3,4.

The next example requires a nontrivial coboundary equivalence between a recurrence of the CMF and the formula of interest.

### B.1.3  Unification example: cluster $(-1, 3, 3)$ ($\delta = -0.91$)

The following example pertains to the $\delta = -0.91$ cluster in Table 16. The canonical form generated by the CMF:

$$\begin{aligned} \text{PCF}_{\text{CMF}} =& \text{PCF}(-7568n^5 - 11664n^4 + 6992n^3 + 6036n^2 - 279n - 162, \\ & - 24n(2n+1)(4n-3)(4n-1)(6n-7)(6n-5)(22n^2 - 39n - 1)(22n^2 + 49n + 9)) \end{aligned}$$

and the canonical form corresponding to the series

$$2\pi = \sum_{k=1}^{\infty} \frac{16^k (22k^2 - 17k + 3)\binom{4k}{2k}}{k(4k-3)(4k-1)\binom{3k}{k}\binom{6k}{3k}}$$

is

$$\begin{aligned} \text{PCF}_{42} =& \text{PCF}(3784n^4 + 156n^3 - 1942n^2 + 261n + 45, \\ & - 24n(2n-3)(4n-3)(4n-1)(6n-5)(6n-1)(11n-14)(11n+8)) \end{aligned}$$

Computing convergence rates, we find both have $r = 0.52$, so they are not folded. Applying UMAPS results in coboundary matrix and polynomials:

$$U(n) = \begin{pmatrix} 1848n^5 - 7676n^4 + 10730n^3 - 5605n^2 + 682n + 21 & 6690816n^9 - 50485248n^8 + 157736064n^7 \\ 1 & 2860n^4 - 10680n^3 + 13481n^2 - 6348n + 756 \end{pmatrix}$$

$$p_A(n) = -22n^2 + 61n - 42$$
$$p_B(n) = 44n^3 + 120n^2 + 67n + 9$$

showing that the series is contained within the CMF.

### B.1.4  Unification example: cluster $(0, 0, 1)$ ($\delta = -1.00$)

Here we show the four formulas listed explicitly in Fig. 6 for trajectory $(0, 0, 1)$ are all equivalent, these correspond to indices 71, 72, 75 and 76 of Table 16. All formulas have $\delta = -1.00$ of course, so they proceed to the convergence rate matching stage.

First, consider the two polynomial continued fractions (formulas 72 and 71):

$$\text{PCF}_{72} = \text{PCF}(1, n(n+1))$$
$$\text{PCF}_{71} = \text{PCF}(2, n^2)$$

These have convergence rates $0$, so all combinations of folding up to 2 are passed to UMAPS. Applying UMAPS, the coboundary algorithm (Appendix C.3), the two turn out to be coboundary to each other with no folds necessary:

$$U(n) = \begin{pmatrix} n & -n^2 \\ -1 & n-1 \end{pmatrix}$$

with trivial "external" polynomials $p_A(n), p_B(n) = 1$.

Next, consider formulas 75 and 76: Applying the same steps as above shows they have convergence rates of $0$. Passing all three combinations of folding by 2 to UMAPS, we obtain a coboundary matrix relating the two formulas with no folds necessary:

$$U(n) = \begin{pmatrix} 4n^2 - 4n + 1 & 16n^2 - 16n^3 + 4n^2 \\ -1 & -4n^2 - 4 \end{pmatrix}$$

these too with trivial "external" polynomials.

At this point there are two clusters. Can they be united? Consider the pair 72, 75. Passing all three combinations for folds to the coboundary algorithm, a coboundary matrix comes up:

$$U(n) = \begin{pmatrix} 4n^2 - 4n + 2 & 16n^4 - 16n^3 + 4n^2 \\ -1 & 1 - 4n^2 \end{pmatrix}$$

with external polynomials

$$p_A(n) = 2n - 1$$
$$p_B(n) = 2n + 1$$

In conclusion, we have found that formulas 71, 72, 75, 76 are equivalent to each other. Only one need be proven to prove all of the others.

## B.2   Unification by the $\zeta(3)$ Conservative Matrix Field (CMF)

The following matrices define a 2D CMF that computes the constant $\zeta(3)$.

$$M_{\mathbf{x}} = \begin{pmatrix} 0 & -x^3 \\ (x+1)^3 & x^3 + 2y(2x+1)(y-1) + (x+1)^3 \end{pmatrix}$$
$$M_{\mathbf{y}} = \begin{pmatrix} -x^3 + 2x^2y - 2xy^2 + y^3 & -x^3 \\ x^3 & x^3 + 2x^2y + 2xy^2 + y^3 \end{pmatrix} \tag{13}$$

Converting the two formulas for $\zeta(3)$, Eq. (11), to canonical forms, respectively yields

$$\frac{2}{5 - 4\zeta(3)} = 12 + \cfrac{-48}{40 + \cfrac{-648}{98 + \cfrac{-3840}{\ddots + \cfrac{-n(n+1)^4(n+2)}{2n^3 + 9n^2 + 17n + 12 + \ddots}}}}$$

$$\frac{\zeta(3)}{\zeta(3) - 1} = 9 + \cfrac{-64}{35 + \cfrac{-729}{91 + \cfrac{-4096}{\ddots + \cfrac{-(n+1)^6}{2n^3 + 9n^2 + 15n + 9 + \ddots}}}}$$

Applying our methods, we find a coboundary matrix of degree 6 with linear external polynomials,

$$U(n) = \begin{pmatrix} n^3 + n^2 + n + 1 & n^6 + 5n^5 + 10n^4 + 10n^3 + 5n^2 + n \\ -1 & -n^3 - 4n^2 - 5n \end{pmatrix}$$

$$p_A(n) = n$$
$$p_B(n) = n + 1$$

which are together a certificate of equivalence for the canonical forms, and hence for the original formulas too.

Both of these formulas are found in the $(1, 0)$ direction of the $\zeta(3)$ CMF (Eq. (13)), which corresponds to the continued fraction:

$$\frac{1}{\zeta(3)} = 1 + \cfrac{-1}{9 + \cfrac{-64}{35 + \cfrac{-729}{\ddots + \cfrac{-n^6}{2n^3 + 3n^2 + 3n + 1 + \ddots}}}}$$

### B.3 Unification by the $e$ Conservative Matrix Field (CMF)

The following matrices define a 2D CMF that computes the constant $e$.

$$M_{\mathbf{x}} = \begin{pmatrix} 1 & -y-1 \\ -1 & x+y+2 \end{pmatrix} \qquad\qquad M_{\mathbf{y}} = \begin{pmatrix} 0 & -y-1 \\ -1 & x+y+1 \end{pmatrix} \qquad (14)$$

We present below the unification of all 15 $e$ formulas from [46].

Table 5: Unification of all 15 $e$ formulas from [46] by CMF Eq. (14), via UMAPS. Because these formulas converge super-exponentially, the convergence rates are unbounded and depend on the depth to which they are computed. The folds needed to match the formulas in the (0,1) trajectory were luckily still uncovered.

| Cluster | | Formula | Value | Convergence rate |
|---|---|---|---|---|
| (1, 1) $\delta = 1.00$ | | | | |
| | 1 | PCF($4n + 2$,1) | $\frac{1+e}{-1+e}$ | 17.36 |
| (1, 0) $\delta = 0.00$ | | | | |
| | 2 | PCF($n + 2, -n$) | $\frac{e}{-1+e}$ | 7.30 |
| | 3 | PCF($n + 3, -n$) | $e$ | 7.30 |
| | 4 | PCF($n^2 + 3n + 3, -n^3 - 2n^2$) | $\frac{4e}{-1+2e}$ | 7.30 |
| | 5 | PCF($n^2 + 4n + 3, -n^3 - 3n^2$) | $\frac{3e}{2(-1+e)}$ | 7.30 |
| | 6 | PCF($n + 4, -n$) | $\frac{e}{-2+e}$ | 7.31 |
| | 7 | PCF($n + 5, -n$) | $\frac{e}{6-2e}$ | 7.31 |
| | 8 | PCF($n + 6, -n$) | $\frac{e}{-24+9e}$ | 7.31 |
| | 9 | PCF($n^2 + 6n + 7, -n^3 - 3n^2$) | $\frac{6e}{-3+2e}$ | 7.31 |
| (0, 1) $\delta = 0.00$ | | | | |
| | 10 | PCF($n$,$n$) | $\frac{1}{-1+e}$ | 7.30 |
| | 11 | PCF($n + 1$,$n$) | $\frac{1}{-2+e}$ | 7.30 |
| | 12 | PCF($n + 2$,$n$) | $\frac{1}{-5+2e}$ | 7.31 |
| | 13 | PCF($n + 3$,$n$) | $\frac{1}{-16+6e}$ | 7.31 |
| | 14 | PCF($4n^2 + 14n + 11, -4n^2 - 6n$) | $\frac{3}{3-e}$ | 15.98 |
| | 15 | PCF($4n^2 + 10n + 5, -4n^2 - 2n + 2$) | $1 + \frac{e}{-2+e}$ | 15.98 |

For example, consider these two polynomial continued fractions:

$$\frac{6e}{2e-3} = 7 + \cfrac{-4}{14 + \cfrac{-20}{23 + \cfrac{-54}{\ddots + \cfrac{n^2(-n-3)}{n(n+6)+7+\ddots}}}} \qquad\qquad \frac{4e}{2e-1} = 3 + \cfrac{-3}{7 + \cfrac{-16}{13 + \cfrac{-45}{\ddots + \cfrac{n^2(-n-2)}{n(n+3)+3+\ddots}}}}$$

Applying UMAPS, we find coboundary matrix

$$U(n) = \begin{pmatrix} n^3 + 4n^2 + 6n + 6 & n^4 + 4n^3 + 4n^2 \\ -n - 1 & -n^2 - n + 2 \end{pmatrix}$$

and external polynomials

$$p_A(n) = n + 2$$
$$p_B(n) = n + 3$$

proving that the two formulas are equivalent.

In similar fashion, we arrive at the other equivalences summarized in Table 5.

### B.4 Equivalence example: notable formulas for $\pi$

Eq. (9) [48] and Eq. (10) [54] are converted to recurrences, both of order 2, after which they are converted to canonical form PCFs, respectively:

$$\frac{239018472}{-3528 + 1123\pi} = \text{PCF}_{\text{Ramanujan}} = \text{PCF}(a_1(n), b_1(n))$$

$$\frac{1047212167162854000}{-341446000 + 108685699\pi} = \text{PCF}_{\text{Sun}} = \text{PCF}(a_2(n), b_2(n))$$

$$a_1(n) = 534215282560n^4 + 1630601631968n^3 + 1686512782328n^2 + 618081838666n + 27955409115$$
$$b_1(n) = n^3(366856790423961600n^5 + 588680355780034560n^4 - 56045383774765056n^3$$
$$- 487988770034755584n^2 - 247923828204062976n - 34298642100691584)$$
$$a_2(n) = 35468306308982528n^5 + 180047738533689024n^4 + 332745102731042192n^3$$
$$+ 272631301503072468n^2 + 89876772716256332n + 5411146610376015$$
$$b_2(n) = n^2(1617129676787301327212642304n^8 + 42895855268945734350604861 44n^7$$
$$- 2833662109815845910282240 00n^6 - 578121362136863737845475 7376n^5$$
$$- 10392789775942675228520171 52n^4 + 195228587262173057883521 2800n^3$$
$$+ 65692626394504296555019008n^2 - 10048226342191391688515596 8n$$
$$- 15998802007913316345 60)$$

The canonical forms share $\delta = -0.29$ and $r = 13.56$, so the recurrences are not folded and UMAPS is applied, resulting in a coboundary matrix of degree 10, coupled with external polynomials of degree 4, rendering Eqs. 9 and 10 equivalent:

$$p_A(n) = 11398398784n^4 - 19077544640n^3 + 9321191372n^2 - 1315967464n - 20955$$
$$p_B(n) = 171680n^3 + 395264n^2 + 290210n + 67749$$

$$U(n) = \begin{pmatrix} U_{11}(n) & U_{12}(n) \\ U_{21}(n) & U_{22}(n) \end{pmatrix}$$

$$U_{11}(n) = 28876576000n^5 - 61950059840n^4 + 1926362087953808n^3$$
$$- 1678583497631500n^2 - 139251745359750n$$
$$U_{12}(n) = 10242045593095285103984640 00n^{10} - 21191237220245887903275417 60n^9$$
$$+ 1056569453502166636426985472n^8 + 24497499562221163441 2208128n^7$$
$$- 20556483493578159808474 2144n^6 - 7035268079364204755 916288n^5$$
$$+ 847052781450583359776947 2n^4 + 1348682584079729606 40n^3$$
$$U_{21}(n) = 42050n - 29337$$
$$U_{22}(n) = 1491444197503390771200n^6 - 926743682638889031168n^5$$
$$- 132917008783804435411 2n^4 + 980655193799148492576n^3$$
$$- 117379649957600136708n^2 - 9013576532170267008n - 143483055820335$$

### B.5 Equivalence example: formulas for Catalan's constant G

The two formulas for Catalan's constant in Eq. (12) are equivalent via coboundary matrix

$$U(n) = \begin{pmatrix} 4n^2 + 2n & 16n^4 \\ -1 & -4n^2 + 2n - 1 \end{pmatrix}$$

and trivial external polynomials—$p_A(n) = 1$ and $p_B(n) = 1$. Note that the formulas in Eq. (12) are not polynomial continued fractions in their current form due to a periodicity of 2 in the $a_n$, $b_n$ functions. To convert them into polynomial form, they are first inflated to make them integer continued fractions, then folded by 2 to make them polynomial, resulting in the canonical forms:

$$\frac{1}{2-2G} = 7 + \cfrac{-16}{23 + \cfrac{-256}{55 + \cfrac{-1296}{\ddots + \cfrac{-16n^4}{8n^2 + 8n + 7 + \ddots}}}} \qquad \frac{2}{2G-1} = 5 + \cfrac{-32}{25 + \cfrac{-384}{61 + \cfrac{-1728}{\ddots + \cfrac{16n^3(-n-1)}{8n^2 + 12n + 5 + \ddots}}}}$$

## B.6  Example polynomial recurrences for formulas, some of order greater than 2

One of Ramanujan's 1914 formulas (shown in Fig. 3) is represented by the following order-2 polynomial recurrence using RISC's tool for finding minimal recurrences [33]:

$$0 = \left(-\frac{18426177}{3162112} - n \cdot \frac{1603904319}{63242240} - n^2 \cdot \frac{185504787}{3952640} - n^3 \cdot \frac{605532897}{12648448} - n^4 \cdot \frac{22985937}{790528} - n^5 \cdot \frac{83133297}{7905280} - n^6 \cdot \frac{2072547}{988160} - n^7 \cdot \frac{729}{4096}\right) f(n)$$
$$+ \left(\frac{569520571}{15810560} - n \cdot \frac{1927156365}{12648448} - n^2 \cdot \frac{1076882413}{3952640} - n^3 \cdot \frac{3379580191}{12648448} - n^4 \cdot \frac{122831663}{790528} - n^5 \cdot \frac{424008847}{7905280} - n^6 \cdot \frac{10066461}{988160} - n^7 \cdot \frac{3367}{4096}\right) f(1+n)$$
$$+ \left(\frac{40384}{965} + n \cdot \frac{171504}{965} + n^2 \cdot \frac{61640}{193} + n^3 \cdot \frac{60808}{193} + n^4 \cdot \frac{35600}{193} + n^5 \cdot \frac{61907}{965} + n^6 \cdot \frac{23709}{1930} + n^7\right) f(2+n).$$

Some formulas are generated by recurrences of higher order. The methods presented in this work can be generalized to higher degree recurrences (see Lemma 4 of Appendix C.3). For example, these two series for Catalan's constant [13]

$$G = \sum_{n=0}^{\infty} \frac{1}{2^{n+1}} \sum_{k=0}^{n} \binom{n}{k} \frac{(-1)^k}{(2k+1)^2}, \qquad 2G = \sum_{n=0}^{\infty} \frac{2^n}{(2n+1)\binom{2n}{n}} \sum_{k=0}^{n} \frac{1}{2k+1},$$

are given by the same recurrence of order 3:

$$0 = \left(-\frac{3}{2} - \frac{5n}{4} - \frac{n^2}{4}\right) f(n) + \left(\frac{21}{2} + \frac{29n}{4} + \frac{5n^2}{4}\right) f(1+n)$$
$$+ \left(-\frac{85}{4} - 13n - 2n^2\right) f(2+n) + \left(\frac{49}{4} + 7n + n^2\right) f(3+n)$$

meaning their recurrence matrices, which are in general companion matrices (Eq. (1)), are trivially coboundary, with the identity coboundary matrix. Other cases in which the recurrence is not precisely the same require a generalization of the coboundary algorithm Appendix C.3 to solve for the coboundary matrix. Additional examples of high-order recurrences, for $\pi$:

$$\frac{2}{\pi} = \sum_{k=0}^{\infty} \sum_{i=0}^{k} \binom{2k-2i}{k-i}^2 \binom{2i}{i}^2 \cdot k \left(\frac{1}{32}\right)^k$$

is given by the order-3 recurrence

$$0 = \left(-4 - 8n - 6n^2 - 2n^3 - \frac{n^4}{4}\right) f(n) + \left(\frac{81}{4} + \frac{173n}{4} + \frac{65n^2}{2} + \frac{21n^3}{2} + \frac{5n^4}{4}\right) f(n+1)$$
$$+ \left(-\frac{137}{4} - \frac{297n}{4} - \frac{111n^2}{2} - \frac{35n^3}{2} - 2n^4\right) f(n+2) + \left(18 + 39n + 29n^2 + 9n^3 + n^4\right) f(n+3)$$

$$\frac{2}{\pi} = \sum_{n=0}^{\infty} (-1)^n \frac{(3n+1)}{32^n} \sum_{k=0}^{n} \binom{2n-2k}{n-k} \binom{2k}{k} \binom{n}{k}^2$$

is given by the order-3 recurrence

$$0 = \left(-\frac{35}{9} - \frac{26n}{3} - \frac{23n^2}{3} - \frac{121n^3}{36} - \frac{35n^4}{48} - \frac{n^5}{16}\right) f(n) + \left(-\frac{365}{9} - \frac{181n}{2} - \frac{1879n^2}{24} - \frac{1589n^3}{48} - \frac{55n^4}{8} - \frac{9n^5}{16}\right) f(n+1)$$
$$+ \left(-\frac{356}{9} - \frac{503n}{6} - \frac{1633n^2}{24} - \frac{1279n^3}{48} - \frac{81n^4}{16} - \frac{3n^5}{8}\right) f(n+2) + \left(84 + 183n + 154n^2 + \frac{568n^3}{9} + \frac{38n^4}{3} + n^5\right) f(n+3)$$

# C Algorithms

This section contains an in-depth description of the algorithms discussed in Section 3. The algorithms are ordered top-down, from the highest level algorithm to the lowest. A hyperparameter sensitivity study supporting the hyperparameter choices below is included in Appendix D. All algorithms used in the pipeline were run on a 13th Gen i5-13500H Intel Core and are available at https://github.com/RamanujanMachine/euler2ai. Runs required for the sensitivity study were conducted on the Technion High Performance Computing Zeus Cluster.

Table 6: **Algorithms enabling unification by Conservaitve Matrix Field (CMF)**. The matching algorithm and UMAPS are depicted also in Fig. 4.

|  | Name | Dependencies | Input | Output |
|---|---|---|---|---|
| C.1 | Coboundary graph growing algorithm | C.2, C.4, C.5 | canonical-form formulas and CMF-derived representative recurrences | a *coboundary graph*—a forest of equivalent formulas with equivalence-proving transformations stored in the edges |
| C.2 | Matching algorithm | C.3 | two polynomial recurrences $A(n), B(n)$ | a triple of three transformations: a fold transform for each recurrence $(F_A, F_B)$ and a cobandary transform $(U(n), p_A(n), p_B(n))$ s.t. $\frac{p_A(n)}{p_B(n)} \cdot U(n)^{-1} \cdot F_A(A(n)) \cdot U(n+1) = F_B(B(n))$ |
| C.3 | UMAPS: The coboundary solving algorithm | | two polynomial recurrences $A(n), B(n)$ | a cobandary transform $(U(n), p_A(n), p_B(n))$ s.t. $\frac{p_A(n)}{p_B(n)} \cdot U(n)^{-1} \cdot A(n) \cdot U(n+1) = B(n)$ |
| C.4 | Conversion to canonical form | RISC's tool [33] | Diophantine formula | minimal polynomial recurrence (PCF for second-order recurrences) |
| C.5 | Recurrence generation by CMF | | CMF matrices, start point, trajectory | polynomial trajectory matrix (polynomial recurrence) |

## C.1 The coboundary graph growing algorithm

**Input:** (1) Initialized graph with no edges, where each node is a canonical-form recurrence for a formula. Each node has precomputed attributes: irrationality measure ($\delta$) and convergence rate ($r$) (Section 2). (2) Dataframe of canonical forms generated by the CMF (see Appendix C.5), with precomputed attributes as in (1).

**Output:** The graph as a forest, where each edge contains the rigorous transformation between the two nodes it connects. This forest is termed a *coboundary graph*. Every tree groups formulas that are rigorously-equivalent together—the trees found by the algorithm are actually subgraphs of cliques; not all clique edges are computed during the matching phase to make the algorithm more efficient, but they all exist. If a tree contains a CMF-generated canonical form, all of the formulas within are unified.

**Steps:**

1. Cluster nodes according to $\delta$: Initialize an empty list for each value between $-1.00$ and $0.05$ (or higher upper bound, depending on the PCF with highest $\delta$) at intervals of $0.05$ ($\delta$ "granularity") including the edge values. For every cluster value $\delta_c$, insert every node with $\delta$ obeying $|\delta - \delta_c| < 0.03$ ($\delta$ "similarity threshold") into the cluster's corresponding list. Note that a node can appear in two clusters initially. The maximum distance between two nodes in the same cluster is $2 \cdot 0.03 = 0.06$. This is intentional to prevent missing matches. Additionally, initialize an empty list "nonhubs."

2. For every cluster of nodes indexed by $\delta_c$:

- For each node, attempt to match it to all nodes appearing at higher indices that are not in "nonhubs," using Appendix C.2. Store successful matches as new graph edges containing the transformations. Every formula matched to the current node is placed in the global "nonhubs" list.

- At the end of this process, nodes not in the "nonhubs" list are deemed final non-CMF "hubs." They are currently the roots of a forest since a node is added to "nonhubs" as soon as it is matched with the current node during the loop, and it is subsequently skipped.

3. Now, for each "hub," attempt to match it to all CMF-generated canonical forms with $\delta_{CMF}$ that obey $|\delta - \delta_{CMF}| < 0.03$. When an edge is found, break for the current hub—the hub and all nodes connected to it have been unified in the CMF.

4. Output the resulting graph.

## C.2   The matching algorithm

Equivalence is a binary relation between two formulas. To find one, formulas represented as canonical form recurrences (Appendix E.4.1) are clustered based on dynamical metrics (Section 3.4). Promising pairs, those with similar irrationality measure $\delta$ (Eq. (5)), are folded (Appendix E.5) according to the ratio of their convergence rates $r$ (Eq. (4)), then sent to UMAPS—the coboundary solving algorithm (Appendix C.3). The result is a triple: the fold transform needed to be applied to each of the recurrences, and the coboundary matrix and polynomials as outputted by UMAPS. This algorithm is also summarized in Fig. 4.

**Input:** Two linear polynomial-coefficient recurrences with matrices $A(n)$ and $B(n)$.

**Output:** A list of three transformations connecting the recurrences (if found)

- fold transformation for each of the recurrences $(F_A, F_B)$
- coboundary transformation linking the two recurrences after they are folded $(U(n), p_A(n), p_B(n))$:

$$\frac{p_A(n)}{p_B(n)} \cdot U(n)^{-1} \cdot F_A(A(n)) \cdot U(n+1) = F_B(B(n))$$

**Steps:**

1. If unknown, compute the convergence rates of each of the recurrences using Eq. (4) at $n = 2000$. If the convergence rate is less than $5 \cdot 10^{-2}$, set it to 0.

2. Compute the ratio of the convergence rates of the recurrences, when defined, as $R = \frac{|r_A|}{|r_B|}$. Assume this number can be approximated as a rational with low-enough denominator to good degree. Set $R = 0$ if either of the convergence rates is 0 (this is normally the case when $\delta = -1$, such PCFs converge polynomially).

3. Fold recurrence $A$ by $r_B$ and recurrence B by $r_A$ if $R \neq 0$. If $R = 0$, all combinations of folding the recurrences by up to 2 are passed to the next phase for a total of 3 options: fold neither or (one option) or fold one of them by 2 (two options). This is because the convergence rate does not contain fold information for PCFs with slow convergence.

4. Apply the coboundary solving algorithm Appendix C.3 between the recurrences. If a coboundary transform is successfully found, output fold transforms from 4 and the coboundary transform.

## C.3   UMAPS: The coboundary solving algorithm

We aim to find a coboundary relation between two recurrences, yet as mentioned in Section 3.5, this constitutes a nonlinear problem. To see why, let us write the equations for the coboundary matrix and polynomials in full for order $m = 2$. We have, given two polynomial matrices $A(n)$ and $B(n)$

$$p_A(n) \cdot A(n) \cdot U(n+1) = p_B(n) \cdot U(n) \cdot B(n) \iff$$

$$\sum_{i=0}^{d_{p_A}} p_{A,i} n^i \cdot \begin{pmatrix} \sum_{i=0}^{d_{A_{11}}} A_{11,i} n^i & \sum_{i=0}^{d_{A_{12}}} A_{12,i} n^i \\ \sum_{i=0}^{d_{A_{21}}} A_{21,i} n^i & \sum_{i=0}^{d_{A_{22}}} A_{22,i} n^i \end{pmatrix} \cdot \begin{pmatrix} \sum_{i=0}^{d_{U_{11}}} U_{11,i}(n+1)^i & \sum_{i=0}^{d_{U_{12}}} U_{12,i}(n+1)^i \\ \sum_{i=0}^{d_{U_{21}}} U_{21,i}(n+1)^i & \sum_{i=0}^{d_{U_{22}}} U_{22,i}(n+1)^i \end{pmatrix} =$$

$$\sum_{i=0}^{d_{p_B}} p_{B,i} n^i \cdot \begin{pmatrix} \sum_{i=0}^{d_{U_{11}}} U_{11,i} n^i & \sum_{i=0}^{d_{U_{12}}} U_{12,i} n^i \\ \sum_{i=0}^{d_{U_{21}}} U_{21,i} n^i & \sum_{i=0}^{d_{U_{22}}} U_{22,i} n^i \end{pmatrix} \cdot \begin{pmatrix} \sum_{i=0}^{d_{B_{11}}} B_{11,i} n^i & \sum_{i=0}^{d_{B_{12}}} B_{12,i} n^i \\ \sum_{i=0}^{d_{B_{21}}} B_{21,i} n^i & \sum_{i=0}^{d_{B_{22}}} B_{22,i} n^i \end{pmatrix}$$

The unknowns of this equation are the coefficients of $p_A(n), p_B(n)$ and the coefficients of the four polynomials of $U(n)$: $\{p_{A,k}|k = 0, \ldots, d_{p_A}\}$, $\{p_{B,k}|k = 0, \ldots, d_{p_B}\}$, $\{U_{ij,k}|k = 0, \ldots, d_{U_{ij}}; i, j \in \{1, 2\}\}$. From the equation written above, it is clear that the coefficients are

coupled (e.g. the product $p_{A,0} \cdot U_{21,0}$ is in the expansion), making the equations resulting from equating powers of $n$ nonlinear.

An empirical method that takes care of nonlinearity and the ambiguity in unknown polynomial degrees would be much preferred. Luckily, such a method is possible: we use Lemma 1 (proven below), a necessary condition a coboundary matrix between two matrices must obey, to solve for $U(1)$ and then reconstruct the coboundary matrix in full. Lemma 4 generalizes this condition to general matrix order $m$, enabling a coboundary-solving algorithm for recurrences of any depth.

**Input:** Two order-$m$ recurrences that converge to the same irrational constants ($m-1$ constants at most) up to Möbius transformations (Eq. (17) and a requested depth up to which a coboundary matrix will be fit to "measurements" of the coboundary matrix. Denote the recurrence matrices by $A(n)$ and $B(n)$, their limits by $L_A$ and $L_B$ (integer Möbius transformations of the irrational constant $\varphi$ for $m = 2$ and projective vectors for higher order recurrences—see the setup of Lemma 4) and the requested depth as $N$.

**Output:** A polynomial matrix $U(n)$ and two additional polynomials $p_A(n), p_B(n)$, if found, satisfying the coboundary condition Eq. (8):

$$p_A(n) \cdot A(n) \cdot U(n+1) = p_B(n) \cdot U(n) \cdot B(n).$$

**Steps:**

1. Solve for the first coboundary matrix $U(1)$ up to a multiplicative factor using :

   - (order $m = 2$) From Lemma 1, write the equation $L_A = U(1)(L_B)$, where $U(1)(\cdot)$ is the Möbius transformation, for the four unknowns of the first coboundary matrix

   $$U(1) = \begin{pmatrix} u_{11} & u_{12} \\ u_{21} & u_{22} \end{pmatrix}.$$

   Writing the Möbius transformation explicitly, the equation reads $L_A = \frac{u_{11}L_B + u_{12}}{u_{21}L_B + u_{22}}$. Without loss of generality, the above equation can be written as $\frac{\alpha\varphi + \beta}{\gamma\varphi + \delta} = \frac{a\varphi + b}{c\varphi + d}$, where $a, b, c, d$ depend on $\{u_{ij}\}_{i,j \in \{1,2\}}$ and $L_B$ and $L_A = \frac{\alpha\varphi + \beta}{\gamma\varphi + \delta}$. By equating coefficients of powers of $\varphi$ in the numerator and denominator independently, we obtain four equations for the four unknowns of $U$. (This must hold assuming the irrational number does not solve a quadratic equation, a condition met by all non-algebraic constants like $\pi$ and $\zeta(3)$.) The solution is the four rational numbers of $U(1)$ (up to a multiplicative factor).

   - (general order $m$) From Lemma 4, write the equation $L_A = U(1)L_B$. Every row of this equation can be written as $L_{A_i} = U(1)_i \cdot L_B$ where $U(1)_i$ is the $i^{\text{th}}$ row of $U(1)$. Since we are solving over the rationals, PSLQ [26] can be used to find a solution: $a_i \cdot L_{A_i} - \sum_{j=1}^{m} b_{ij}L_{B_j} = 0$ (where $a_i, \{b_{ij}\}_{j=1}^{m}$ were found over the integers via PSLQ) $\implies$ $U(1)_{ij} = \frac{b_{ij}}{a_i}$. If the symbolic limits are known in terms of a set of algebraically independent constants, $U(1)$ can also be found by equating coefficients.

2. Propagate the coboundary matrix to the requested depth $N$ using the necessary condition for a coboundary equivalence:

   - $A(n) \cdot U(n+1) \propto U(n) \cdot B(n) \implies U(n+1) \propto A(n)^{-1} \cdot U(n) \cdot B(n)$
   - The resulting matrices are again the rational matrices $\{U_i\}_{i=2}^{N}$, known only up to independent multiplicative factors.

3. Divide each of the "measured" $\{U(i)\}_{i=1}^{N}$ by, e.g., the $U_{11}$ element of each matrix.

   - Take care to pick an element $U_{11}, U_{12}, U_{21}, U_{22}$ (etc. for orders $> 2$) that does not zero out on $1, 2, \ldots, N$. If all do then pick the element whose last 0 arrives at the earliest index and toss out all measurements preceeding this index.
   - The result is a new list of matrices. If a polynomial coboundary relation exists between the two recurrences, $p_A(n) \cdot A(n) \cdot \tilde{U}(n+1) = p_B(n) \cdot \tilde{U}(n) \cdot B(n)$ for some polynomials $p_A(n)$ and $p_B(n)$, then the matrices we have found are precisely the result of dividing $\tilde{U}(n)$ by its (e.g.) first element. Meaning our measurements should be of a rational matrix.

4. Fit rational functions to each of the elements of the measured $U$:

- Writing a general $m \times m$ rational matrix requires $2m^2$ polynomials $\{p_{ij}(n), q_{ij}(n)\}_{i,j=1}^m$. Equating the rational function of each element $\frac{p_{ij}(n)}{q_{ij}(n)}$ to the measurements of that element $U_{ij}(n)$, yields a system of linear equations for the coefficients of the numerator polynomial and the denominator polynomial: $\frac{p_{ij}(n)}{q_{ij}(n)} = U_{ij}(n) \implies p_{ij}(n) - U_{ij}(n)q_{ij}(n) = 0$ for $n = 1, \ldots, N$.
- The result is a rational coboundary matrix *hypothesis* $U_h(n)$. Hypothesis—because this is a rational fit to our empirical coboundary matrices.

5. The final stage of the algorithm is to verify the hypothesis.

- Multiply out $A(n) \cdot U_h(n+1)$ and $U_h(n) \cdot B(n)$. If the resulting matrices differ by a multiplicative factor (not a matrix, a rational function), meaning the condition $A(n) \cdot U_h(n+1) \propto U_h(n) \cdot B(n)$ holds, then the coboundary matrix hypothesis $U_h(n)$ is a valid, though still rational, coboundary matrix.
- Multiply both sides of $A(n) \cdot U_h(n+1) \propto U_h(n) \cdot B(n)$ by the least common multiple of the denominators of $U_h(n+1), U_h(n)$ to convert the coboundary relation into a polynomial one of the form $p_A(n) \cdot A(n) \cdot U(n+1) = p_B(n) \cdot U(n) \cdot B(n)$, where $p_A(n), p_B(n)$ are polynomials and $U(n)$ is a polynomial matrix.

6. Output $U(n)$, $p_A(n)$ and $p_B(n)$ (which are defined only if $U_h$ was valid).

This algorithm (for $m = 2$) is also available on our interactive, online algorithm demonstration [4].

### C.3.1  Sufficiency of UMAPS

We can show that UMAPS is sufficient to recover the coboundary matrix, provided an upper bound on the degrees of the polynomials in its rational entries. We now proceed to prove Corollary 1. According to this corollary, the sensitivity study (Appendix D) indicates that any coboundary matrix among the harvested formulas that has not been recovered must involve rational entries with polynomials of degree at least 60. Let us restate the corollary:

**Corollary 1.** *(Sufficiency of UMAPS.) If a coboundary matrix exists for two matrices and every rational-function entry of the coboundary matrix has polynomials of degree at most $d$, then running UMAPS with $N \geq 2d + 1$ suffices to recover the coboundary matrix.*

*Proof.* By Lemma 1 (and the generalization to any recurrence order, Lemma 4), two matrices that are coboundary have limits related by the transformation $U(1)$. Therefore, during the execution of UMAPS, the quantities $U(1), \ldots, U(N)$ correspond to correct measurements of $U(n)$.

Assume that each rational-function entry of the coboundary matrix has numerator and denominator polynomials of degree at most $d$. This gives a total of $2d + 2$ coefficients. Since multiplying both the numerator and denominator by a common nonzero factor yields the same rational function, there are effectively $2d + 1$ independent degrees of freedom.

Choosing $N = 2d + 1$ produces a linear homogeneous system of $2d + 1$ equations in $2d + 2$ unknowns for each element of the coboundary matrix. If a coboundary matrix exists, then it is unique up to scale by Lemma 5. So this system has a one-dimensional null space, corresponding to a unique solution up to overall scale. Increasing $N$ beyond $2d + 1$ introduces linearly dependent equations, leaving the null space unchanged.

Hence, a rational fit to the empirical measurements of $U(n)$ will recover the rational-function entries whenever $N \geq 2d + 1$. $\qquad \square$

### C.3.2  Rational function fitting details

A rational function fit is attempted in step 4 of UMAPS, detailed above. Consult [53] for a thorough treatment of rational fitting. The rational fit is conducted by finding the null space of a linear system of equations:

When fitting rational functions, the equations are of the form $\frac{P(i)}{Q(i)} = \frac{p_i}{q_i}$, where $P, Q$ are the polynomials of the rational function and $\frac{p_i}{q_i}$ is the empirical value at $i$. This leads to equations of the form $q_i \sum_{k=0}^{deg_P} P_k i^k - p_i \sum_{k=0}^{deg_Q} Q_k i^k = 0$, for the $deg_P + deg_Q + 2$ polynomial coefficients. These coefficients are determined by finding the null space of the system of equations.

In practice, the system is a SymPy matrix of integers (e.g. $q_i i^k$ for $P_k$). SymPy does this via the linear system's reduced row-echelon form, and uses rational number objects (not floats), so precision is guaranteed.

Since the rational fit is not a least-squares fit, noise and finite precision are not sources of error. Analytical correctness of the fit is guaranteed by subsequent substitution into the coboundary condition. Once a matrix is successfully recovered, Lemma 5 guarantees this is *the* coboundary matrix, due to $U(n)$'s projective uniqueness.

### C.4 Conversion of formulas to canonical form

We use RISC's *Guess* tool [33] to convert formulas into recurrences, after which correctness is validated numerically. Minimality is then guaranteed using Maple's *MinimalRecurrence* package [57] (additional details available in [60]). Finally, formulas yielding second-order recurrences are represented as continued fractions. The proper initial conditions needed for the continued fractions to generate the formulas are found (Appendix C.4.1).

Our use of RISC's *Guess* algorithm and the following steps can be summarized by:

**Input:** First $N$ approximants of a formula.

**Output:** Minimal polynomial recurrence generating the formula.

As more terms of a sequence are considered, the likelihood that a guessed recurrence fits only the initial terms but fails for later ones decreases. In our experiments, we use $N = 200$ partial sums when converting each series into a corresponding recurrence. The hyperparameter sensitivity study (Appendix D) supports this choice.

Once a recurrence is identified for a formula with a known closed form, we verify it by substituting the formula back into the recurrence. If $\mathcal{R}$ is the recurrence operator and $f(n)$ is the formula, then the goal is to show $\forall n : \mathcal{R}[f](n) = 0$. This verification is done numerically, since substituting full series expressions or complex closed forms into the recurrence does not always allow algebraic simplification. We perform 30 random numerical substitutions into the expression $\mathcal{R}[f](n)$, all of which must agree to within $10^{-10}$ for the recurrence to be accepted. All recurrences were successfully validated under this criterion.

Formulas given by a polynomial recurrence of order 2 can be represented as polynomial $2 \times 2$ companion matrices (polynomial continued fractions): given a recurrence $x_n$

$$c(n)x_n = a(n)x_{n-1} + b(n)x_{n-2}$$

the corresponding canonical form is

$$\text{PCF}\,(a(n), b(n)c(n-1))$$

Which is achieved using inflation by $c(n)$ (see E.4).

Converting a general series to a continued fraction can be done using a technique devised by Euler, however, this technique relies on algebraic manipulation of the term of the series and also does not necessarily yield *polynomial* recurrences, let alone recurrences of order 2. A formula can have a recurrence of order 2 that is not polynomial, but have minimal polynomial recurrence of order 3 (see for example Appendix B.6).

### C.4.1 Conversion of series limit to Polynomial Continued Fraction (PCF) limit

**Input:** Series, with partial sums (not summand) to index $n$ given by $S(n)$, and continued fraction $CF(a_n, b_n)$ from the recurrence fit to the series using RISC's tool.

**Output:** $x \in \mathbb{Q}$ such that the initial conditions

$$\begin{pmatrix} S_0 & xS_1 \\ 1 & x \end{pmatrix}$$

generate the partial sums of the entire series when the continued fraction's recurrence is applied from index 2. By demanding the second convergent of the continued fraction is equal to the second partial sum of the series (recall our notation for the Möbius transform defined by matrix $U$ - $U(\cdot)$):

$$\left(\begin{pmatrix} S_0 & xS_1 \\ 1 & x \end{pmatrix} \cdot \begin{pmatrix} 0 & b_2 \\ 1 & a_2 \end{pmatrix}\right)(0) = S_2$$

we arrive at the equation

$$\frac{S_0 b_2 + xS_1 a_2}{b_2 + xa_2} = S_2$$

Solving for x, we obtain

$$x = \frac{-b_2}{a_2}\left(\frac{S_2 - S_0}{S_2 - S_1}\right)$$

Note that only the first three partial sums of the series $S_0$, $S_1$ and $S_2$ (more generally $n_0$, $n_0 + 1$, $n_0 + 2$, where $n_0$ is the start index of the series summation) and a single partial-numerator and partial-denominator pair of the continued fraction are needed. This is because the continued fraction values $\frac{p_n}{q_n}$ satisfy the recurrence, hence when equating the first two values to the values of the sum we get the exact same series.

The algorithm's result can be converted to initial conditions for when the continued fraction's recurrence is applied from any other index by multiplying by recurrence matrices of appropriate indices and their inverses, i.e. the initial condition that calculates the series starting from $n = 1$ (using the same notation as Eq. (2)) is

$$\underbrace{\begin{pmatrix} S_0 & xS_1 \\ 1 & x \end{pmatrix} \cdot \begin{pmatrix} 0 & b_1 \\ 1 & a_1 \end{pmatrix}^{-1}}_{\text{new initial conditions}} \cdot \begin{pmatrix} 0 & b_1 \\ 1 & a_1 \end{pmatrix} \cdot \begin{pmatrix} 0 & b_2 \\ 1 & a_2 \end{pmatrix} \cdot \begin{pmatrix} 0 & b_3 \\ 1 & a_3 \end{pmatrix} \cdots$$

This method is convenient since there is no need to keep track of transformations applied to the series' recurrence in the process of conversion to continued fraction. In summary, the correct initial conditions for recreating the series from the recurrence are found using only the recurrence and first few partial sums of the series.

This algorithm is also available on our online algorithm demonstration [4].

### C.5  Recurrence generation by Conservative Matrix Field (CMF)

A recurrence is generated from a CMF by selecting a starting point and a trajectory in the CMF. We calculate their corresponding trajectory matrix $M(n)$ (see G.1), and finally convert it to the canonical companion form (see E.2) in order to generate a recurrence.

**Input:** A CMF, a starting point $p$ and a trajectory $t$, both vectors of dimension equal to that of the CMF. Denote the number of matrices in the CMF (the CMF's *dimension*) by $d$ and the number of rows and columns in these square matrices (the CMF's *rank*) by $r$.

**Output:** Polynomial recurrence of order $r$.

**Steps:**

1. Compute a single trajectory step from $p + (n - 1)t$ to $p + nt$ (see Appendix G.1 for a more rigorous definition):

- initialize $M = \mathbf{I}_{r \times r}$ and $cur\_pos = p + (n - 1)t$.
- for i in $1, \ldots, d$:
-       for j in range($t_i$):
-             $M = M \cdot M_{x_i}(cur\_pos)$
-             $cur\_pos = cur\_pos + \mathbf{e}_i$ (where $\mathbf{e}_i$ is 1 at index $i$ and zero elsewhere)

By the end of this loop $cur\_pos = p + nt$ and $M$ is the work matrix from $p + (n-1)t$ to $p + nt$.

2. If interested in companion form (Eq. (1), which corresponds to a PCF), bring the resulting rational matrix $M$ into polynomial companion form per Lemma 2.

3. Return $M$.

It is worth mentioning that there are degrees of freedom in the calculation of the trajectory matrix, due to the conserving property of the CMF. The order of matrix multiplication can be selected, which is advantageous in preventing matrix singularities.

Our search space for formulas in the CMF defined in Eqs. (6) and (25) consisted of the trajectory subspace $T_3 = \{(a, b, c) \in \mathbb{Z}^3 : max\{|a|, |b|, |c|\} \leq 3\}$ and the initial positions $P_1 = (\frac{1}{2}, \frac{1}{2}, \frac{1}{2}) + \{(a, b, c) \in \mathbb{Z}^3 : max\{|a|, |b|, |c|\} \leq 1\}$. In cases where $M(n)$ had a singularity at some point along the trajectory $p + kt$ (where $t \in T_3$ and $p \in P_1$), we shifted the initial position to $p' = p + (k+1)t$ to avoid it.

As can be seen from the definition of $P_1$, starting points are actually non-integer. The CMF starting points appearing in dark blue in Fig. 6 are shown in terms of the difference from $(\frac{1}{2}, \frac{1}{2}, \frac{1}{2})$ which is the true $(0, 0, 0)$ point.

The full dataset of recurrences generated by the CMF for this study is included in the supplementary material (described in Appendix J).

# D   Hyperparameter sensitivity study

Below we provide a sensitivity study showing the linkage percentage and runtime for different $\delta$-clustering granularities and similarity thresholds, values of UMAPS's fit depth, as well as an analysis of the sensitivity of RISC's Guess algorithm to its fit depth. Parameters used in the paper are denoted in bold. Runtimes are normalized by the runtimes for the parameters used in the paper. Runs required for the sensitivity study were conducted on the Technion High Performance Computing Zeus Cluster.

Table 7: Linkage rate via UMAPS for different $\delta$ clustering granularities and similarity thresholds in the graph growing algorithm (Appendix C.1). The numbers listed are % of max linkage achieved (136) between canonical form formulas. The linkage percentage with no clustering criterion is found by setting similarity threshold = 2 and is 100%, as expected.

| Similarity threshold | Clustering granularity | | | | | |
| --- | --- | --- | --- | --- | --- | --- |
| | 0.01 | **0.05** | 0.1 | 0.2 | 0.3 | 0.5 |
| 0.001 | 85 | 76 | 68 | 58 | 54 | 64 |
| 0.003 | 95 | 79 | 74 | 64 | 60 | 70 |
| 0.005 | 100 | 80 | 75 | 64 | 60 | 70 |
| 0.01 | 100 | 97 | 83 | 64 | 62 | 73 |
| 0.02 | 100 | 100 | 86 | 64 | 65 | 73 |
| **0.03** | 100 | **100** | 86 | 64 | 65 | 73 |
| 0.05 | 100 | 100 | 100 | 73 | 71 | 73 |
| 0.1 | 100 | 100 | 100 | 100 | 90 | 73 |
| 0.2 | 100 | 100 | 100 | 100 | 100 | 87 |
| 0.3 | 100 | 100 | 100 | 100 | 100 | 100 |
| 0.5 | 100 | 100 | 100 | 100 | 100 | 100 |

Table 8: Runtime for different $\delta$ clustering granularities and similarity thresholds. The algorithm (Appendix C.1) first matches formulas to find hubs (stage 1), then matches each hub with CMF representatives (stage 2). For coarser granularities ($> 0.1$), the runtime shows three phases across similarity thresholds (regular text, bold, regular). This can be explained by the number of hubs from stage 1 (decrease as similarity threshold increases) vs. the number of CMF representatives that meet the similarity criterion in stage 2 (increase as similarity threshold increases). The runtime value results from their product, which is largest for medium similarity threshold values. Runtime without clustering (similarity threshold = 2): 20.0.

| | Clustering granularity | | | | | |
|---|---|---|---|---|---|---|
| Similarity threshold | 0.01 | **0.05** | 0.1 | 0.2 | 0.3 | 0.5 |
| 0.001 | 0.8 | 0.6 | 0.8 | 0.8 | 0.7 | 0.7 |
| 0.003 | 0.9 | 0.8 | 1.1 | 1.1 | 1.1 | 1.0 |
| 0.005 | 1.1 | 0.9 | 1.3 | 1.2 | 1.3 | 1.2 |
| 0.01 | 1.0 | 0.7 | 1.2 | **2.0** | **2.0** | 1.3 |
| 0.02 | 1.0 | 1.0 | 1.5 | **3.4** | **3.5** | **1.6** |
| **0.03** | 1.0 | **1.0** | 1.4 | **4.7** | **4.9** | **2.4** |
| 0.05 | 1.0 | 1.0 | 1.0 | **4.0** | **3.9** | **3.2** |
| 0.1 | 1.0 | 1.0 | 0.9 | 1.0 | **2.9** | **5.4** |
| 0.2 | 0.9 | 0.9 | 0.9 | 0.9 | 0.9 | **6.1** |
| 0.3 | 1.0 | 0.9 | 0.9 | 1.0 | 0.9 | 1.0 |
| 0.5 | 1.0 | 0.9 | 1.1 | 1.0 | 0.9 | 0.9 |

Table 9: Linkage rate and runtime for different maximum fitting depths in UMAPS. The total runtime of the graph growing algorithm (Appendix C.1) has a clear minimum. The two forces at play: the average UMAPS time increases with $N$, while the number of hubs from stage 1 decreases with $N$; the runtime (their product) is smallest at intermediate $N$. Note the maximum polynomial degree found as a function of $N$ closely follows the upper bound stipulated by Corollary 1.

| UMAPS $N$ | Found relation (canonical) (x/153) | Same CMF (canonical) (x/153) | Max poly deg found | Avg UMAPS time | Total runtime |
|---|---|---|---|---|---|
| 4 | 28 | 22 | 1 | 0.5 | 6.7 |
| 8 | 45 | 38 | 3 | 0.5 | 3.7 |
| 12 | 80 | 67 | 5 | 0.7 | 3.1 |
| 16 | 95 | 78 | 7 | 0.8 | 2.5 |
| 20 | 130 | 81 | 9 | 0.9 | 1.2 |
| 24 | 132 | 81 | 11 | 0.9 | 1.0 |
| 29 | 136 | 81 | 13 | 0.8 | 0.8 |
| 30 | 136 | 81 | 14 | 1.1 | 0.9 |
| **40** | **136** | **81** | **14** | **1.0** | **1.0** |
| 50 | 136 | 81 | 14 | 1.0 | 1.3 |
| 65 | 136 | 81 | 14 | 1.1 | 2.9 |
| 80 | 136 | 81 | 14 | 1.7 | 9.7 |
| 100 | 136 | 81 | 14 | 7.1 | 48.8 |
| 120 | 136 | 81 | 14 | 18.3 | 137.3 |

Table 10: Success rate and runtime of RISC's Guess [33] for different fit depths $N$ over 100 random formulas. These findings indicate that Guess generates robust fits that generalize well.

| Guess $N$ | Recurrences found (x/100) | Of which correct | Avg runtime |
|---|---|---|---|
| 8 | 0 | 0 | 0.02 |
| 10 | 24 | 24 | 0.05 |
| 15 | 30 | 30 | 0.08 |
| 20 | 57 | 57 | 0.18 |
| 25 | 76 | 76 | 0.26 |
| 30 | 92 | 92 | 0.31 |
| 50 | 99 | 99 | 0.46 |
| 100 | 100 | 100 | 0.67 |
| **200** | **100** | **100** | **1.00** |
| 500 | 100 | 100 | 2.79 |

# E    Recurrences and limit-preserving transformations

## E.1    On linear recurrences

Recurrences, also known as "difference equations," [34] are the discrete analog of differential equations. They play a prominent role in various areas of mathematics and science, including Newton's approximation algorithms, counting problems (combinatorics), special functions, and the modeling of economic and biological systems. In this section, we revisit the notion of a linear recurrence that we introduced in Section 2.

A function $u_n$ satisfies a recurrence of order $m$ if it is a solution to the equation:

$$u_n = a_{1,n} u_{n-1} + a_{2,n} u_{n-2} + \ldots a_{m,n} u_{n-m}$$

It is customary to represent the recurrence via the associated companion matrix:

$$\mathrm{CM}(n) := \begin{pmatrix} 0 & 0 & \ldots & 0 & a_{m,n} \\ 1 & 0 & \ldots & 0 & a_{m-1,n} \\ 0 & 1 & \ldots & 0 & a_{m-2,n} \\ \vdots & \vdots & \ddots & \vdots & \vdots \\ 0 & 0 & \ldots & 1 & a_{1,n} \end{pmatrix} \tag{15}$$

Observe that $[u_{n-m}, \ldots, u_{n-2}, u_{n-1}] \, \mathrm{CM}(n) = [u_{n-(m-1)}, \ldots, u_{n-1}, u_n]$. Thus, the companion matrix represents a single step in the recurrence at time $n$. By incrementally multiplying the companion matrix over $n$ steps, we get the matrix:

$$M_n := \prod_{i=1}^{n} \mathrm{CM}(i) = \begin{pmatrix} p_{1,n-m} & \cdots & p_{1,n-1} & p_{1,n} \\ p_{2,n-m} & \cdots & p_{2,n-1} & p_{2,n} \\ p_{3,n-m} & \cdots & p_{3,n-1} & p_{3,n} \\ \vdots & \ddots & \vdots & \vdots \\ p_{m,n-m} & \cdots & p_{m,n-1} & p_{m,n} \end{pmatrix} \tag{16}$$

which we call the $n$-th step matrix. The functions $p_{1,n}, \ldots p_{m,n}$ are solutions to the recurrence equation with initial conditions $p_{i,j} = \delta_i^j$, and any other solution is a linear combination of these. Hence this matrix encapsulated all the information about the recurrence. Explicitly to get a different set of solutions we multiply the step matrix $M_n$ from left by the initial condition matrix $U_{\mathrm{init}}$ where each row represents the initial condition for a solution.

Alternatively, let $A(n) \in \mathrm{PGL}_m(\mathbb{Q}(n))$ be a matrix with coefficient functions. Similar to the process done for the companion matrix, we incrementally multiply the matrix:

$$\mathcal{A}_n := \prod_{i=1}^{n} A(i)$$

We regard $\mathcal{A}_n$ as a "cocycle" (see E.4.2 and E.4.3). We are fundamentally interested in the situation where the matrix $A(n)$ is coboundary equivalent (Eq. (7)) to the companion matrix of a recurrence.

In the context of Diophantine approximation formulas for constants, we examine the ratios of elements in the last column of the step matrix $M_n$. Diophantine approximation formulas can also be derived by taking ratios of elements in $\mathcal{A}_n$.

We will now provide a detailed analysis for the case of second-order recurrences. In this case, we utilize the fact that a 2-by-2 matrix acts by Möbius transformation (linear fractional transformation) on $\mathbb{R} \cup \infty$, and we provide a complete characterization of coboundary equivalence between matrices.

## E.2    Second-order recurrences

Recall that any 2-by-2 matrix $M$ can perform a fractional linear transformation. The action of $M$ on the number $l$ is defined as:

$$\begin{pmatrix} a & b \\ c & d \end{pmatrix}(l) = \frac{a \cdot l + b}{c \cdot l + d} \tag{17}$$

This is known as the Möbius transformation of the matrix acting on $l$. It is customary to extend this action to $\mathbb{R} \cup \infty$ by defining $\begin{pmatrix} a & b \\ c & d \end{pmatrix}(\infty) = \frac{a}{c}$.

Let $a_n, b_n \in \mathbb{Q}(n)$ be rational functions, and consider the recurrence relation $u_n = a_n u_{n-1} + b_n u_{n-2}$. By incrementally multiplying the companion matrix $\mathrm{CM}(n)$ and examining the ratio of the elements in the first (or second) column, we effectively apply a Möbius transformation. Explicitly, we have:

$$M_N(0) = \frac{p_n}{q_n}, \quad M_N(\infty) = \frac{p_{n-1}}{q_{n-1}}$$

Alternatively, we can start with a general matrix $A(n) \in \mathrm{PGL}_2(\mathbb{Q}(n))$ and incrementally multiply to obtain the matrix:

$$\mathcal{A}_n := \prod_{i=1}^{n} A(i)$$

We can then examine the ratios of elements in the first or second column of this matrix, or even apply it as a Möbius transformation at a value $l \in \mathbb{Q}$. This can potentially provide a Diophantine approximation to some constant, similar to how a recurrence relation provides a formula via a Möbius transformation of the step matrices.

We call such a matrix $A(n)$ a *formula generating matrix* if, for any $l \in \mathbb{Q} \cup \{\infty\}$, the sequence $\mathcal{A}_n(l)$ provides an approximation formula for a constant $L$.

Indeed as the following lemma asserts any such matrix is coboundary equivalent (see Definition (7)) to a companion matrix of a recurrence equation.

**Lemma 2.** *A matrix $A(n) \in \mathrm{PGL}_2(\mathbb{Q}(n))$ is a formula-generating matrix if and only if there exist a matrix $U(n) \in \mathrm{PGL}_2(\mathbb{Q}(n))$ and a companion matrix $\mathrm{CM}(n)$ associated to a recurrence $u_n$, such that*

$$U(n) \cdot A(n) = \mathrm{CM}(n) \cdot U(n+1)$$

*Proof.* First, assume

$$A(n) = \begin{pmatrix} \alpha(n) & \beta(n) \\ \gamma(n) & \delta(n) \end{pmatrix}$$

is a formula-generating matrix, and denote by $\mathcal{A}_n = \prod_{i=1}^{n} A(i)$. If $\gamma(n) = 0$ then the bottom left entry in $\mathcal{A}_n$ will remain 0 for any $n$, resulting in $\lim_{n \to \infty} \mathcal{A}_n . \infty = \infty$. In this case it must be that $\beta(n)$ is nonzero since if it is zero then so is the top right entry of $\mathcal{A}_n$, resulting in $\lim_{n \to \infty} \mathcal{A}_n . 0 = 0 \neq \infty$. Hence we may assume that either $\gamma(n)$ or $\beta(n)$.

If $\gamma(n) \neq 0$ then define

$$U(n) = \begin{pmatrix} \gamma(n) & -\alpha(n) \\ 0 & 1 \end{pmatrix}$$

And get that

$$U(n) \cdot A(n) \cdot U^{-1}(n+1) = \begin{pmatrix} 0 & -\alpha(n)\delta(n) + \beta(n)\gamma(n) \\ \frac{\gamma(n)}{\gamma(n+1)} & \frac{\alpha(n+1)\gamma(n)}{\gamma(n+1)} + \delta(n) \end{pmatrix}$$

And if $\gamma(n) = 0$ define

$$U(n) = \begin{pmatrix} 1 & 0 \\ \alpha(n-1) & \beta(n-1) \end{pmatrix}$$

to get that

$U(n) \cdot A(n) \cdot U^{-1}(n+1)$

$$= \begin{pmatrix} 0 & 1 \\ -\frac{\alpha(n)(\alpha(n-1)\beta(n) + \beta(n-1)\delta(n))}{\beta(n)}) + \alpha(n-1)\alpha(n) & \frac{\alpha(n-1)\beta(n) + \beta(n-1)\delta(n)}{\beta(n)}) \end{pmatrix}$$

Where these matrices are projectively equivalent to a matrix in a companion form [49].

To conclude the proof, it is left to show that companion form matrices are formula-generating.

Let $\mathrm{CM}(n) = \begin{pmatrix} 0 & b_n \\ 1 & a_n \end{pmatrix}$ be a companion form matrix with $\lim\limits_{n \to \infty} \dfrac{p_n}{q_n} = L$.

$$M_n(l) = \frac{p_{n-1} \cdot l + p_n}{q_{n-1} \cdot l + q_n} = \frac{p_n}{q_n} \cdot \frac{\frac{p_{n-1}}{p_n} \cdot l + 1}{\frac{q_{n-1}}{q_n} \cdot l + 1}$$

And since the ratio of $\frac{p_n}{q_n}$ converges, the growth rate of these solutions is the same, implying that

$$\lim_{n \to \infty} \left( \frac{\frac{p_{n-1}}{p_n} \cdot l + 1}{\frac{q_{n-1}}{q_n} \cdot l + 1} \right) = 1 \text{ making } \lim_{n \to \infty} M_n(l) = L. \qquad \square$$

### E.3 Proof of expected limit value

We call a companion matrix $A(n)$ Poincaré-Perron type if the recurrence relation represented by $A(n)$ is of Poincaré-Perron type, that is

$$u_n = a_{1,n} u_{n-1} + a_{2,n} u_{n-2} + \ldots + a_{m,n} u_{n-m}$$

where the limit

$$\lim_{n \to \infty} a_{i,n} = \alpha_i \in \mathbb{R}$$

exists and is finite, and the characteristic roots, i.e., the solutions $\lambda_i$ to the equation

$$\sum_i \alpha_i \cdot X^{m-i} = X^m$$

are all with different absolute values

$$|\lambda_1| > \cdots > |\lambda_m|$$

In this case, the theorem of Poincaré-Perron [19, Chapter 8, Theorem 10] states that there are $m$-linearly independent solutions $P_i$ to the recurrence $u_n$ such that

$$\lim_{n \to \infty} \frac{P_i(n+1)}{P_i(n)} = \lambda_i$$

**Lemma 3.** *Given a companion matrix $A(n)$ with the step matrix:*

$$\mathcal{A}_n = \prod_{i=1}^{n} A(i) = \begin{pmatrix} p_{1,n-m} & \cdots & p_{1,n-1} & p_{1,n} \\ p_{2,n-m} & \cdots & p_{2,n-1} & p_{2,n} \\ p_{3,n-m} & \cdots & p_{3,n-1} & p_{3,n} \\ \vdots & \ddots & \vdots & \vdots \\ p_{m,n-m} & \cdots & p_{m,n-1} & p_{m,n} \end{pmatrix}$$

*Assume $A(n)$ represents a Poincaré-Perron companion matrix with fundamental roots*

$$|\lambda_1| > \cdots > |\lambda_m|.$$

*Then any column of the limit matrix*

$$\widetilde{\mathcal{A}} = \lim_{n \to \infty} \mathcal{A}_n^{-t}$$

*is projectively equivalent to the vector* $\hat{X} = \begin{pmatrix} X_1 \\ X_2 \\ X_3 \\ \vdots \\ X_m \end{pmatrix}$, *where $\hat{X}$ is the initial condition for the*

*"slowest" growing solution and $X_i \neq 0$, i.e, the one that grows like $|\lambda_m|^n$. In particular*

$$\frac{\widetilde{\mathcal{A}}_{i,j}}{\widetilde{\mathcal{A}}_{i',j}} = \frac{X_i}{X_{i'}}$$

*Proof.* Let $y_n$ be a solution for the recurrence defined by $A(n)$, setting

$$Y_n = \begin{pmatrix} y_n & y_{n+1} & y_{n+2} & \cdots & y_{n+m-1} \end{pmatrix}$$

We get that

$$Y_0 \mathcal{A}_n = Y_n$$

Hence

$$\mathcal{A}_n^{-t} \cdot \begin{pmatrix} y_n \\ y_{n+1} \\ y_{n+2} \\ \cdots \\ y_{n+m-1} \end{pmatrix} = \begin{pmatrix} y_0 \\ y_1 \\ y_2 \\ \cdots \\ y_{m-1} \end{pmatrix}$$

Each row gives the equation

$$\left( \left( \mathcal{A}_n^{-t} \right)_{i,1} \cdot y_n \right) \cdot \left( \sum_{j=1}^{m} \frac{\left( \mathcal{A}_n^{-t} \right)_{i,j}}{\left( \mathcal{A}_n^{-t} \right)_{i,1}} \cdot \frac{y_{n+j-1}}{y_n} \right) = y_i$$

Assuming $y_n$ is the "slowest" solution to the recurrence, we get that

$$\lim_{n \to \infty} \frac{y_{n+1}}{y_n} = \lambda_m$$

Note that each row of $\widetilde{A}$ is a general solution to the system defined by $A^{-t}(n)$ so by Poincare-Perron theorem $\frac{\left( \mathcal{A}_n^{-t} \right)_{i,j+1}}{\left( \mathcal{A}_n^{-t} \right)_{i,1}} = \lambda_m^{-j}$ is independent of $i$, so we get that

$$\lim_{n \to \infty} \frac{\sum_{j=1}^{m} \frac{\left( \mathcal{A}_n^{-t} \right)_{i,j}}{\left( \mathcal{A}_n^{-t} \right)_{i,1}} \cdot \frac{y_{n+j-1}}{y_n}}{\sum_{j=1}^{m} \frac{\left( \mathcal{A}_n^{-t} \right)_{i',j}}{\left( \mathcal{A}_n^{-t} \right)_{i',1}} \cdot \frac{y_{n+j-1}}{y_n}} = 1$$

Hence

$$\lim_{n \to \infty} \frac{\left( \left( \mathcal{A}_n^{-t} \right)_{i,1} \cdot y_n \right) \cdot \left( \sum_{j=1}^{m} \frac{\left( \mathcal{A}_n^{-t} \right)_{i,j}}{\left( \mathcal{A}_n^{-t} \right)_{i,1}} \cdot \frac{y_{n+j-1}}{y_n} \right)}{\left( \left( \mathcal{A}_n^{-t} \right)_{i',1} \cdot y_n \right) \cdot \left( \sum_{j=1}^{m} \frac{\left( \mathcal{A}_n^{-t} \right)_{i',j}}{\left( \mathcal{A}_n^{-t} \right)_{i,1}} \cdot \frac{y_{n+j-1}}{y_n} \right)} = \frac{y_i}{y_{i'}}$$

As needed. □

### E.4  Coboundary transform

Given a recurrence relation $u_n$, we are interested in analyzing the dynamics of its solutions, which translates to analyzing the behavior of the companion matrix $\mathrm{CM}(n)$ increments, i.e., the behavior of the step matrix $M_n$ as $n$ approaches infinity.

We are interested in the ratio between solutions to the recurrence. Consequently, we look at these matrices as elements in the projective group $\mathrm{PGL}_m\left( \mathbb{Q}(n) \right)$ We will recall the definition of a coboundary equivalence:

Two matrices $A(n), B(n) \in \mathrm{PGL}_m\left( \mathbb{Q}(n) \right)$ are said to be *coboundary equivalent* if there exist a matrix $U(n)$ such that

$$A(n) \cdot U(n+1) = U(n) \cdot B(n) \tag{18}$$

From the equation above we see that $A(n) = U(n) \cdot B(n) \cdot U^{-1}(n+1)$ thus exhibiting a "telescoping effect" on the product, resulting in the equation:

$$\left( \prod_{i=1}^{n} A(n) \right) \cdot U(n+1) = U(1) \cdot \left( \prod_{i=1}^{n} B(n) \right)$$

This suggests that these matrices share similar dynamics.

We denote $A(n) \sim B(n)$, to indicate that they are coboundary equivalent, and we often call the matrix $U(n)$ a *coboundary transform* from $A(n)$ to $B(n)$.

**Index shift**:

Notice that the matrices $A(n)$ and $A(n+1)$ are coboundary equivalent by simply taking $U(n) = A(n)$. This implies that an index shift is a coboundary operation. Since coboundary is an equivalence, by transitivity, any integer index shift is also a coboundary operation.

**Inflation**:

Given a function $u_n$ satisfying the relation

$$u_n = a_{1,n}u_{n-1} + a_{2,n}u_{n-2} + \ldots + a_{m,n}u_{n-m}$$

We can define $\widetilde{u}_n = u_n \cdot \prod_{i=1}^{n} c_i$ for some function $c_n$. The function $\widetilde{u}_n$ satisfied the relation

$$\widetilde{u}_n = a_{1,n}c_n\widetilde{u}_{n-1} + a_{2,n}c_nc_{n-1}\widetilde{u}_{n-2} + \ldots + a_{m,n}c_nc_{n-1}\cdots c_{n-(m-1)}\widetilde{u}_{n-m}$$

Let $\mathrm{CM}(n)$ denote the companion form of $u_n$ and $\widetilde{\mathrm{CM}}(n)$ denote the companion form of $\widetilde{u}_n$, then the matrix

$$U(n) = \begin{pmatrix} \prod\limits_{i=1}^{m-1} c(n-i) & 0 & \ldots & 0 & 0 \\ 0 & \prod\limits_{i=1}^{m-2} c(n-i) & \ldots & 0 & 0 \\ \vdots & \vdots & \ddots & \vdots & \vdots \\ 0 & 0 & \ldots & c(n-1) & 0 \\ 0 & 0 & \ldots & 0 & 1 \end{pmatrix}$$

Is a coboundary transform between $\mathrm{CM}(n)$ and $\widetilde{\mathrm{CM}}(n)$. This process is called inflation and it allows one to construct a Polynomial Continued Fraction (PCF) from a recurrence given by rational function coefficients. In [52], this is done by taking $c_n$ to be the LCM of the denominators. In the other direction starting from a $PCF(a(n), b(n))$ and taking $C_n = n^{-\deg(a)}$, the inflation by $c_n$ is called factorial reduction [10].

### E.4.1 Canonical form

We have seen that a recurrence relation with rational function coefficients can be transformed via inflation to a recurrence with polynomial coefficients. We say that a recurrence

$$u_n = a_{1,n}u_{n-1} + a_{2,n}u_{n-2} + \ldots + a_{m,n}u_{n-m}$$

is in canonical form if the coefficients $a_{i,n}$ are polynomials, and for any other recurrence with polynomial coefficients $a'_{i,n}$ which is coboundary equivalent to it, the degree of each $a_{i,n}$ is less than or equal to the degree of the corresponding $a'_{i,n}$. Appendix C.4 explains how the minimal recurrence is found for each formula.

Lemma 2 asserts that in the case of second-order recurrences, any formula-generating matrix is equivalent to a companion matrix. In turn, this is equivalent to a companion matrix of a recurrence $u_n = a(n)u_{n-1} + b(n)u_{n-2}$ with polynomial coefficients through the process of inflation. We denote such recurrences by $PCF(a(n), b(n))$ as they are associated with a polynomial continued fraction

$$M(n)(0) = \cfrac{b(1)}{a(1) + \cfrac{b(2)}{a(2) + \cfrac{}{\ddots \quad + \cfrac{b(n)}{a(n)}}}} = \frac{p_n}{q_n}$$

The canonical form of $PCF\left(n^2 + n + 1, n^4 + n^2 + 1\right)$, for example, is equal to $PCF\left(1, 1\right)$, the simplest continued fraction for the Golden Ratio.

### E.4.2  Group cocycle and coboundary

We now describe the mathematical context that motivates our definition of coboundary equivalence.

Let $\Gamma$ be a group acting by automorphisms on a group $G$ via the map:

$$\varphi\colon \Gamma \to \operatorname{Aut}(G)$$

A $\Gamma$-cocycle with respect to $\varphi$ is a map $\mathcal{M}\colon \Gamma \to G$ satisfying the cocycle condition:

$$\mathcal{M}(\gamma_1\gamma_2) = \mathcal{M}(\gamma_1) \cdot \varphi_{\gamma_1}(\mathcal{M}(\gamma_2)) \tag{19}$$

We declare two cocycles, $\mathcal{M}$ and $\mathcal{M}'$, to be coboundary equivalent if there exists an element $g \in G$ such that:

$$g \cdot \mathcal{M}(\gamma) = \mathcal{M}'(\gamma) \cdot \varphi_\gamma(g) \tag{20}$$

For further details, we refer the interested reader to Chapter 5 of [51].

Here, we focus on the case where $\Gamma = \mathbb{Z}$ and $G = \operatorname{PGL}_m\left(\mathbb{Q}(n)\right)$.

For any $k \in \Gamma$, and $A(n) \in G$, the action map is defined by $\varphi_m(A(n)) = A(n + k)$. Since $\mathbb{Z}$ is generated by a single element, any cocycle $\mathcal{M}\colon \Gamma \to G$ is determined by the value $\mathcal{M}_1(n) \in G$. For $k > 0$ the cocycle condition implies the

$$\mathcal{M}_k(n) = \prod_{i=0}^{k-1} \mathcal{M}_1(n + i)$$

Moreover since $\mathcal{M}_0(n) = \operatorname{Id}_m$, have that $\mathcal{M}_k(n) = \left(\mathcal{M}_{-k}(n - k)\right)^{-1}$

From the definition of the coboundary equivalence in the context of group cocycles, we get that two $\Gamma$-cocycles, $\mathcal{M}$ and $\mathcal{M}'$, are coboundary equivalent if there exists $U(n) \in G$ such that:

$$U(n) \cdot \mathcal{M}_1(n) = \mathcal{M}'_1(n) \cdot U(n + 1)$$

When regarding a matrix $A(n) \in \operatorname{PGL}_m\left(\mathbb{Q}(n)\right)$ as the matrix $\mathcal{M}_1(n)$ generating the cocycle, this equivalence is precisely the definition given in Definition (7).

### E.4.3  Cocyles in dynamical systems

Let $T$ be a homeomorphism $T\colon X \to X$ of some topological space. In dynamical systems, one studies the evolution of the system under the repeated application of the transformation $T$.

A function

$$\mathcal{A}\colon \mathbb{Z} \times X \to \operatorname{GL}_m(\mathbb{R})$$

satisfying

$$\mathcal{A}(k_1 + k_2, x) = \mathcal{A}(k_1, x)\mathcal{A}(k_2, T^{k_1}(x))$$

is called a continuous linear cocycle [12].

Also in this context, a single matrix $A(n) \in \operatorname{GL}_m\left(\mathbb{Q}(n)\right)$ defines a cocycle. This is done by taking $X = \mathbb{R}$, and constructing a cocycle by defining

$$\mathcal{A}(k, x) = \prod_{i=0}^{k-1} A(x + i)$$

Typically, it is assumed that the topological space $X$ is a compact space or a probability measure space. [18] Our definition of coboundary equivalence (7) is not only mathematically natural but also captures the essence of our dynamic framework. The vast data collected on related formulas suggest that this notion is effective for equating formulas, as it preserves the measure of irrationality.

## E.5 Fold transform

Consider the matrix $A(n)$ as encoding a process defined by incrementally multiplying the matrix. We might wish to associate a matrix with an accelerated process, one that goes "k steps at a time." That is, a matrix $A_{k\text{-fold}}(n)$ such that taking one step in $A_{k\text{-fold}}(n)$ is equivalent to taking $k$-steps in $A$. This requirement can be expressed as:

$$\prod_{i=1}^{nk} A(i) = \prod_{i=1}^{n} A_{k\text{-fold}}(i)$$

And the k-fold matrix is given by the product:

$$A_{k\text{-fold}}(n) := \prod_{j=1}^{k} A\left(k(n-1)+j\right)$$

When the matrix $A(n)$ is a trajectory matrix within CMF (as described in Appendix G.1 ), the $k$-folding of $A(n)$ corresponds to the trajectory matrix associated with a direction that is $k$-times the original direction.

### E.5.1 An example of folding per convergence rate

The ratio of $r$ (Eq. (4)) between formulas, when they exist and are non-zero, hints that one may be a transformation of a subsequence of the other. To see why, consider two series; one with summand $s(n)$ and the other with summand $s(3n-2) + s(3n-1) + s(3n)$. These series have the same limit, as the latter produces a subseries of the former, but its convergence rate may be 3 times higher. Evidently, their summands do not appear directly equivalent. Our algorithm thus applies *folding* (Appendix E.5) to equate the $r$ values of each pair of formulas that are candidates for equivalence. following is an example of two such series for $\pi$ (120 and 121 from Table 17):

$$\sum_{n=0}^{\infty} a_n := \sum_{n=0}^{\infty} (-1)^n 4^{-n} \left(\frac{1}{4n+3} + \frac{2}{4n+2} + \frac{2}{4n+1}\right) = \pi \tag{21}$$

featured in [1, 30], with a convergence rate of 1.39, and

$$\sum_{k=0}^{\infty} b_k := \sum_{k=0}^{\infty} 16^{-k} \left(-\frac{1}{4(8k+7)} - \frac{1}{2(8k+6)} - \frac{1}{2(8k+5)} + \frac{1}{8k+3} + \frac{2}{8k+2} + \frac{2}{8k+1}\right) = \pi \tag{22}$$

featured in [8], with a convergence rate of 2.77, are not directly coboundary equivalent. However, one can examine that when summing over Eq. (21) in pairs, meaning:

$$\sum_{n=0}^{\infty} a_n = \sum_{k=0}^{\infty} a_{2k} + a_{2k+1}$$

it is easily verified that $b_k = a_{2k} + a_{2k+1}$. Thus, these formulas ought to be considered equivalent, and indeed were matched by our algorithm (Appendix C.2) by first folding the sum in Eq. (21) by $\frac{2.77}{1.39} = 2$.

## F Coboundary matrix properties

### F.1 A necessary condition on the coboundary matrix

Below we include a proof of the necessary condition a coboundary matrix must obey that was leveraged to create UMAPS Appendix C.3. We start with the case of $2 \times 2$ matrices (second-order recurrences), then generalize to any order.

**Lemma 1.** *(A necessary condition on the coboundary equivalence matrix.) Let $L_A = \lim_{n\to\infty} PCF\left(a(n), b(n)\right)$ and $L_B = \lim_{n\to\infty} PCF\left(c(n), d(n)\right)$ be converging PCFs with associated companion matrices $A(n), B(n) \in \mathrm{PGL}_2\left(\mathbb{Q}(n)\right)$. If $A(n)$ is coboundary to $B(n)$, then $L_A$ and $L_B$ are related through a rational Möbius transformation, moreover, if $U(n)$ is the coboundary matrix then $L_A = U(1)(L_B)$ ($U(1)$ applied to $L_B$ as a Möbius transformation).*

*Proof.* Let $\mathcal{A}_n$ and $\mathcal{B}_n$ be the step matrices of the PCF recurrence.

$$\mathcal{A}_n = \prod_{i=1}^{n} A(i) = \begin{pmatrix} p_{n-1} & p_n \\ q_{n-1} & q_n \end{pmatrix}, \quad \mathcal{B}_n = \prod_{i=1}^{n} B(i) = \begin{pmatrix} s_{n-1} & s_n \\ t_{n-1} & t_n \end{pmatrix}$$

Let

$$U(n+1) = \begin{pmatrix} \alpha(n) & \beta(n) \\ \gamma(n) & \delta(n) \end{pmatrix}$$

with $\alpha, \beta, \gamma, \delta$ polynomials. Since $A(n)$ and $B(n)$ are coboundary equivalent

$$\mathcal{A}_n \cdot U(n+1) = U(1) \cdot \mathcal{B}_n$$

This implies equality when applying a factional linear transformation to 0.

$$\left( \begin{pmatrix} p_{n-1} & p_n \\ q_{n-1} & q_n \end{pmatrix} \cdot U(n+1) \right)(0) = \left( U(1) \cdot \begin{pmatrix} s_{n-1} & s_n \\ t_{n-1} & t_n \end{pmatrix} \right)(0) \tag{23}$$

Taking the limit, The left hand side of equation (23) above yields

$$\lim_{n \to \infty} \frac{p_{n-1}\beta(n) + p_n\delta(n)}{q_{n-1}\beta(n) + q_n\delta(n)} = \lim_{n \to \infty} \frac{p_{n-1}}{q_{n-1}} \cdot \left( \frac{\frac{\beta(n)}{\delta(n)} + \frac{p_n}{p_{n-1}}}{\frac{\beta(n)}{\delta(n)} + \frac{q_n}{q_{n-1}}} \right)$$

While on the right-hand side:

$$\lim_{n \to \infty} \left( U(1) \cdot \begin{pmatrix} s_{n-1} & s_n \\ t_{n-1} & t_n \end{pmatrix} \right)(0) = U(1) \left( \lim_{n \to \infty} \begin{pmatrix} s_{n-1} & s_n \\ t_{n-1} & t_n \end{pmatrix}(0) \right) = U(1)(L_B)$$

The matrix $A(n)$ is a companion matrix for a second-order linear recurrence, any solution for the recurrence is composed of two prime solutions $x_n, y_n$, where the growth of $x_n$ is dominant over the growth of $y_n$. Since $\frac{p_n}{q_n}$ converge, we know that $p_n$ and $q_n$ both have an $x_n$ component to them,

making $\frac{p_n}{p_{n-1}}$ and $\frac{q_n}{q_{n-1}}$ both grow asymptotically the same, this renders $\lim_{n \to \infty} \left( \frac{\frac{\beta(n)}{\delta(n)} + \frac{p_n}{p_{n-1}}}{\frac{\beta(n)}{\delta(n)} + \frac{q_n}{q_{n-1}}} \right) = 1$

and we may conclude that

$$\lim_{n \to \infty} \frac{p_{n-1}\beta(n) + p_n\delta(n)}{q_{n-1}\beta(n) + q_n\delta(n)} = \lim_{n \to \infty} \frac{p_{n-1}}{q_{n-1}} \cdot \left( \frac{\frac{\beta(n)}{\delta(n)} + \frac{p_n}{p_{n-1}}}{\frac{\beta(n)}{\delta(n)} + \frac{q_n}{q_{n-1}}} \right) = L_A \cdot 1$$

Hence $L_A = U_1(L_B)$ as needed. $\qquad \square$

Let us generalize Lemma 1 for matrices of higher dimensions.

**Lemma 4.** *(Generalized necessary condition on the coboundary equivalence matrix.) Given two coboundary equivalent companion matrices $A(n), B(n) \in \mathrm{PGL}_m(\mathbb{Q}(n))$, $A(n)U(n+1) = U(n)B(n)$ with $m \geq 2$ and the step matrices:*

$$\mathcal{A}_n = \prod_{i=1}^{n} A(i) = \begin{pmatrix} p_{1,n-m} & \cdots & p_{1,n-1} & p_{1,n} \\ p_{2,n-m} & \cdots & p_{2,n-1} & p_{2,n} \\ p_{3,n-m} & \cdots & p_{3,n-1} & p_{3,n} \\ \vdots & \ddots & \vdots & \vdots \\ p_{m,n-m} & \cdots & p_{m,n-1} & p_{m,n} \end{pmatrix}, \qquad \mathcal{B}_n = \prod_{i=1}^{n} B(i) = \begin{pmatrix} q_{1,n-m} & \cdots & q_{1,n-1} & q_{1,n} \\ q_{2,n-m} & \cdots & q_{2,n-1} & q_{2,n} \\ q_{3,n-m} & \cdots & q_{3,n-1} & q_{3,n} \\ \vdots & \ddots & \vdots & \vdots \\ q_{m,n-m} & \cdots & q_{m,n-1} & q_{m,n} \end{pmatrix}$$

*s.t the right most columns of $\mathcal{A}_n, \mathcal{B}_n$ both converge projectively:*

$$L_A, L_B \in \mathbb{P}^{n-1}\mathbb{R}, \ L_A := \lim_{n \to \infty} \begin{pmatrix} p_{1,n} \\ \vdots \\ p_{m,n} \end{pmatrix}, \ L_B := \lim_{n \to \infty} \begin{pmatrix} q_{1,n} \\ \vdots \\ q_{m,n} \end{pmatrix}$$

*and all entries of $L_A, L_B$ are non-zero, we have:*

$$L_A = U(1)L_B$$

*Proof.* Let

$$U(n+1) = \begin{pmatrix} u_{1,1}(n) & \cdots & u_{1,m-1}(n) & u_{1,m}(n) \\ u_{2,1}(n) & \cdots & u_{2,m-1}(n) & u_{2,m}(n) \\ u_{3,1}(n) & \cdots & u_{3,m-1}(n) & u_{3,m}(n) \\ \vdots & \ddots & \vdots & \vdots \\ u_{m,1}(n) & \cdots & u_{m,m-1}(n) & u_{m,m}(n) \end{pmatrix}$$

With $u_{i,j}(n)$ rational functions. Given $A(n)U(n+1) = U(n)B(n)$, we have $\mathcal{A}_n U(n+1) = U(1)\mathcal{B}_n$. This implies the following equality:

$$\mathcal{A}_n U(n+1) \begin{pmatrix} 0 \\ 0 \\ \vdots \\ 1 \end{pmatrix} = U(1)\mathcal{B}_n \begin{pmatrix} 0 \\ 0 \\ \vdots \\ 1 \end{pmatrix} \implies \mathcal{A}_n \begin{pmatrix} u_{1,m}(n) \\ u_{2,m}(n) \\ \vdots \\ u_{m,m}(n) \end{pmatrix} = U(1) \begin{pmatrix} q_{1,n} \\ q_{2,n} \\ \vdots \\ q_{m,n} \end{pmatrix} \tag{24}$$

Taking the projective limit on both sides of this equation yields on the right:

$$\lim_{n\to\infty} U(1) \begin{pmatrix} q_{1,n} \\ q_{2,n} \\ \vdots \\ q_{m,n} \end{pmatrix} = U(1)L_B$$

On the left, we first need to claim a few things regarding the growth rate of solutions to the recurrence represented by $A(n)$. Using Poincare Perron asymptotics, we can claim that there are $m$ different canonical solutions $x_n^{(i)}$, with different growth rates. Thus any solution is a linear combination of these canonical solutions. For a solution $r_n = \sum a_i x_n^{(i)}$, we shall call the fastest growing canonical solution $x_n^{(i)}$ s.t $a_i \neq 0$ the *dominating solution*. The projective convergence of $\begin{pmatrix} p_{1,n} \\ \vdots \\ p_{m,n} \end{pmatrix}$ to nonzero limits, implies that all $p_{i,n}$ have the same dominating solution. Otherwise, if the dominating solution of $p_{j,n}$ grows faster than the dominating solution of $p_{k,n}$, then $\lim_{n\to\infty} \frac{p_{k,n}}{p_{j,n}} = 0$, in contradiction to our assumption of non-zero entries in $L_A$.

Finally, let us return to the left-hand side of (24):

$$\mathcal{A}_n \begin{pmatrix} u_{1,m}(n) \\ u_{2,m}(n) \\ \vdots \\ u_{m,m}(n) \end{pmatrix} = \begin{pmatrix} \sum_{i=1}^m p_{1,n-i+1} u_{i,m}(n) \\ \sum_{i=1}^m p_{2,n-i+1} u_{i,m}(n) \\ \vdots \\ \sum_{i=1}^m p_{m,n-i+1} u_{i,m}(n) \end{pmatrix}$$

As we are interested in the projective limit, let us consider the limit of the ratio of the $k$th and $j$th elements in the left-hand side:

$$\frac{\sum_{i=1}^m p_{k,n-i+1} u_{i,m}(n)}{\sum_{i=1}^m p_{j,n-i+1} u_{i,m}(n)} = \frac{p_{k,n}}{p_{j,n}} \cdot \frac{\sum_{i=1}^m \frac{p_{k,n-i+1}}{p_{k,n}} u_{i,m}(n)}{\sum_{i=1}^m \frac{p_{j,n-i+1}}{p_{j,n}} u_{i,m}(n)}$$

Now, given that both $p_{k,n}, p_{j,n}$ have the same dominating solution, we have that $\frac{p_{k,n-i+1}}{p_{k,n}}, \frac{p_{j,n-i+1}}{p_{j,n}}$ grow asymptotically the same in $n$, making:

$$\lim_{n\to\infty} \frac{\sum_{i=1}^m \frac{p_{k,n-i+1}}{p_{k,n}} u_{i,m}(n)}{\sum_{i=1}^m \frac{p_{j,n-i+1}}{p_{j,n}} u_{i,m}(n)} = 1 \implies \lim_{n\to\infty} \frac{\sum_{i=1}^m p_{k,n-i+1} u_{i,m}(n)}{\sum_{i=1}^m p_{j,n-i+1} u_{i,m}(n)} = \lim_{n\to\infty} \frac{p_{k,n}}{p_{j,n}}$$

Which implies, by definition of projective convergence:

$$\lim_{n\to\infty} \mathcal{A}_n \begin{pmatrix} u_{1,m}(n) \\ u_{2,m}(n) \\ \vdots \\ u_{m,m}(n) \end{pmatrix} = L_A$$

Which finally gives

$$L_A = U(1)L_B$$

$\square$

### F.2 Uniqueness

**Lemma 5.** *(Projective uniqueness of coboundary matrices.) Let $A(n), B(n) \in \mathrm{PGL}_m\left(\mathbb{Q}(n)\right)$ be two rational matrices as in Lemma 4. Suppose further that the projective limit $L_B$ of $B(n)$ has no algebraic relations over the rationals. Then, if $A(n), B(n)$ are coboundary equivalent, there exists a unique coboundary matrix realizing this equivalence.*

*Proof.* Let $U_1(n), U_2(n)$ be two coboundary matrices between $A(n), B(n)$. We therefore have $\Rightarrow A(n) = U_1(n) \cdot B(n) \cdot U_1(n+1)^{-1} = U_2(n) \cdot B(n) \cdot U_2(n+1)^{-1}$. So $B(n)$ is coboundary to itself with $U_3(n) = U_1(n)^{-1}U_2(n)$. We will show that $U_3(n)$ is the identity in $\mathrm{PGL}_m\left(\mathbb{Q}(n)\right)$, $I_m$, i.e. that there are no non-trivial coboundary matrices between $B(n)$ and itself. By construction

$$B(k) = U_3(k) \cdot B(k) \cdot U_3(k+1)^{-1}$$

Therefore if $U_3(1) = I_m$ then $U_3(k) = I_m$ for any $k \in \mathbb{N}$. Assuming $L_B$ has no algebraic relations over the field of rational numbers, it follows that no non-trivial matrix $U$ with rational coefficients satisfies

$$U \cdot L_B = L_B$$

By Lemma 4 we have that $L_B = U_3(1)L_B$, hence $U_3(1) = I_m$ as desired. $\qquad\square$

## G  The Conservative Matrix Field (CMF)

The CMF defined in [21] has polynomial coefficients, which is not enough for our purposes since the $\pi$-CMF (Eq. (25)) has rational function as coefficients. We redefine the CMF in a way that allows one to insert rational functions as coefficients.

**Definition 1.** *A $d$-dimensional CMF of rank $m$ is defined by a collection of $d$-matrices $M_{\mathbf{x}_1}, \ldots, M_{\mathbf{x}_d}$ in $\mathrm{PGL}_m\left(\mathbb{Q}(x_1, \ldots, x_d)\right)$ satisfying the conserving property, that is, for any pair $i \neq j$*

$$M_{\mathbf{x}_i}(x_1, \ldots, x_i, \ldots, x_j, \ldots, x_d)M_{\mathbf{x}_j}(x_1, \ldots, x_i + 1, \ldots, x_j, \ldots, x_d)$$
$$=$$
$$M_{\mathbf{x}_j}(x_1, \ldots, x_i, \ldots, x_j, \ldots, x_d)M_{\mathbf{x}_i}(x_1, \ldots, x_i, \ldots, x_j + 1, \ldots, x_d)$$

Envision a $d$-dimensional lattice where each edge has a displacement function representing the "work" of moving between vertices. The conserving property is that: the work is *path-independent*.

### G.1  Trajectory matrices in a CMF

Let $\mathcal{M}$, be a $d$-dimensional CMF of rank $m$. Given a point $x \in \mathbb{Q}^d$ and a direction $v \in \mathbb{Z}^d$ the evaluation map $v \mapsto \mathcal{M}_v(x)$ describes the displacement from point $x$ to point $x + v$. By the conserving property, the total work for a displacement along a broken path, say, first from $x$ to $x + v$ and then from $x + v$ to $(x + v) + w$, is equal to the total displacement from $x$ to $x + v + w$, in terms of the evaluation map this is translated to:

$$\mathcal{M}_{v+w}(x) = \mathcal{M}_v(x)\mathcal{M}_w(x + v)$$

We can construct a general matrix representing the work going from the point $x + (n - 1)v$ to the point $x + nv$, this matrix is in $\mathrm{PGL}_2\left(\mathbb{Q}(n)\right)$ and is given by

$$T_{x,v}(n) = \mathcal{M}_v(x + (n - 1)v)$$

We call this matrix the trajectory matrix associated with the point $x$ in direction $v$.

Note that by the conserving property $\mathrm{Id} = M_{-v+v}(x) = M_{-v}(x)M_v(x - v)$, this provides us with the identity

$$M_{-v}(x) = M_v^{-1}(x - v)$$

Taking a point $x'$ in the lattice $x + \mathbb{Z}^d$ with the same direction $v$ yields the trajectory matrix

$$T_{x',v}(n) = \mathcal{M}_v(x' + (n - 1)v)$$

Since $x'$ is in the same lattice, the difference $w = x' - x$ is in $\mathbb{Z}^d$. The direction $w$ represents the direction from $x$ to $x'$. By the conserving property taking $U(n) = T_{x,w}(n)$, the trajectory matrix from $x$ in the direction $w$ is a coboundary transform between $T_{x,v}(n)$ and $T_{x',v}(n)$.

In terms of the evaluation functions we see explicitly :

$$\mathcal{M}_{v+w}(x + (n-1)v) = \mathcal{M}_{v+w}(x + (n-1)v)\mathcal{M}_w(x + (n-1)v + v) = T_{x,v}(n)T_{x,w}(n+1)$$

and

$$\mathcal{M}_{v+w}(x + (n-1)v) = \mathcal{M}_w(x + (n-1)v)\mathcal{M}_v(x + (n-1)v + (x' - x)) = T_{x,w}(n)T_{x',v}(n)$$

Appendix C.5 describes how to construct a trajectory matrix given a CMF and a trajectory.

## G.2 The $\pi$-CMF

The following three matrices describe a 3D rank 2 CMF (same as Eq. (6)).

$$M_{\mathbf{x}} = \begin{pmatrix} 1 & y \\ \frac{1}{x} & \frac{2x+y-2z+2}{x} \end{pmatrix}$$

$$M_{\mathbf{y}} = \begin{pmatrix} 1 & x \\ \frac{1}{y} & \frac{x+2y-2z+2}{y} \end{pmatrix} \tag{25}$$

$$M_{\mathbf{z}} = \begin{pmatrix} \frac{z(-x-y+z)}{(y-z)(x-z)} & \frac{zxy}{(y-z)(x-z)} \\ \frac{z}{(y-z)(x-z)} & \frac{-z^2}{(y-z)(x-z)} \end{pmatrix}$$

The cocycle $\mathcal{M} \colon \mathbb{Z}^3 \to \mathrm{PGL}_2\left(\mathbb{Q}(n)\right)$ is defined on the generators of $\mathbb{Z}^3$ as

$$\mathcal{M}_{e_1} = M_{\mathbf{x}}, \quad \mathcal{M}_{e_2} = M_{\mathbf{y}}, \quad \mathcal{M}_{e_3} = M_{\mathbf{z}}$$

### G.2.1 Fundamental properties of the $\pi$-CMF

Let $_2F_1(a_1, a_2, a_3; z)$ represent the Gauss hypergeometric function

$$_2F_1(a_1, a_2, a_3; z) = \sum_{n=0}^{\infty} \frac{(a)_n(b)_n}{(c)_n} \frac{z^n}{n!}$$

Denoting the Euler operator by $\Theta = z \cdot \frac{d}{dz}$, the Gauss hypergeometric function has well-known contiguous relations that are encoded by the $\mathcal{M}_{e_i}$ matrices as detailed in [58]. In particular, we have the following property:

$$[_2F_1(a_1, a_2, a_3; \tfrac{1}{2}), \Theta_2F_1(a_1, a_2, a_3; \tfrac{1}{2})] \cdot \mathcal{M}_{e_1} = [_2F_1(a_1+1, a_2, a_3; \tfrac{1}{2}), \Theta_2F_1(a_1+1, a_2, a_3; \tfrac{1}{2})]$$

$$[_2F_1(a_1, a_2, a_3; \tfrac{1}{2}), \Theta_2F_1(a_1, a_2, a_3; \tfrac{1}{2})] \cdot \mathcal{M}_{e_2} = [_2F_1(a_1, a_2+1, a_3; \tfrac{1}{2}), \Theta_2F_1(a_1, a_2+1, a_3; \tfrac{1}{2})]$$

$$[_2F_1(a_1, a_2, a_3; \tfrac{1}{2}), \Theta_2F_1(a_1, a_2, a_3; \tfrac{1}{2})] \cdot \mathcal{M}_{e_3} = [_2F_1(a_1, a_2, a_3+1; \tfrac{1}{2}), \Theta_2F_1(a_1, a_2, a_3+1; \tfrac{1}{2})]$$

### G.2.2 Example of a formula arising as a trajectory in the CMF

Let $x = (\tfrac{1}{2}, \tfrac{-1}{2}, \tfrac{3}{2})$ be an point in space defining the lattice $x + \mathbb{Z}^3$. Directions between points in this lattice correspond to a vast collection of formulas for $\pi$ as seen in Fig. 6.

For completeness of the exposition, we take the simplest direction $e_3 = (0, 0, 1)$ and show how the famous Euler formula $PCF(1, n(n+1)) = \frac{2}{\pi-2}$ sits as a trajectory matrix over the point $x$ in the direction $e_3$.

$$\mathrm{CM}(n) = \begin{pmatrix} 0 & n(n+1) \\ 1 & 1 \end{pmatrix}$$

is the companion form of Euler's PCF.

The trajectory matrix is equal to

$$Tx, e_3(n) = \mathcal{M}_{e_3}(\frac{1}{2}, \frac{-1}{2}, \frac{3}{2} + (n-1)) = \begin{pmatrix} \frac{(2n+1)^2}{4n(n+1)} & \frac{-2n-1}{8n(n+1)} \\ \frac{2n+1}{2n(n+1)} & -\frac{(2n+1)^2}{4n(n+1)} \end{pmatrix}$$

Following the process described in Lemma 2, we define

$$U(n) = \begin{pmatrix} \frac{2n+1}{2n(n+1)} & -\frac{(2n+1)^2}{4n(n+1)} \\ 0 & 1 \end{pmatrix}$$

and get that

$$E(n) = U(n)T(n)U^{-1}(n+1) = \begin{pmatrix} 0 & \frac{n^2+2n+\frac{3}{4}}{n^2+3n+2} \\ 1 & \frac{n+\frac{3}{2}}{n^2+3n+2} \end{pmatrix} \cdot \frac{2n^2+5n+2}{n(2n+3)}$$

Is in companion form. We let $I(n)$ be the inflation by $c_n = n^2 + 3n + 2$ (see E.4) and we get the relation

$$I(n) \cdot E(N) = \mathrm{CM}(n) \cdot I(n+1)$$

Concluding $T(n) \sim E(n) \sim \mathrm{CM}(n)$, which states the equivalence of the trajectory with Euler's PCF.

### G.3 The CMF proves the convergence formula

Recall that the $\pi$-CMF records the data of the contiguous relations of the Gauss hypergeometric function as shown in G.2.1.

For a trajectory $v \in \mathbb{Z}^3$, we denote

$$F_n = {}_2F_1(\frac{1}{2} + v_1 \cdot n, \frac{1}{2} + v_2 \cdot n, \frac{1}{2} + v_3 \cdot n; \frac{1}{2})$$

$$G_n = \Theta_2 F_1(\frac{1}{2} + v_1 \cdot n, \frac{1}{2} + v_2 \cdot n, \frac{1}{2} + v_3 \cdot n; \frac{1}{2})$$

Under this notation, we get that the trajectory matrix $T_{x,v}(n)$ at the initial point $x = (\frac{1}{2}, \frac{1}{2}, \frac{1}{2})$ in direction $v$ is the matrix that represents a forward shift in the parameter $n$, i.e

$$T_{x,v}(n) : [F_n, G_n] \mapsto [F_{n+1}, G_{n+1}]$$

Following the procedure in Lemma 2, we get that $C_{x,v}(n)$, the companion form of $T_{x,v}(n)$, is the matrix that encodes the step going from $[F_n, F_{n+1}]$ to $[F_{n+1}, F_{n+2}]$

$$
\begin{array}{ccc}
[F_n, G_n] & \xrightarrow{T_{x,v}(n)} & [F_{n+1}, G_{n+1}] \\
{\scriptstyle U(n)}\big\uparrow & & \big\downarrow{\scriptstyle U^{-1}(n+1)} \\
[F_n, F_{n+1}] & \xrightarrow{C_{x,v}(n)} & [F_{n+1}, F_{n+2}]
\end{array}
$$

In the case $C_{x,v}(n)$ is Poincaré-Perron type (see Appendix E.3), it follows by Lemma 3 that the projective limit

$$\lim_{n \to \infty} \prod_{i=1}^{n} C_{x,v}(i)^{-t} = [1, \frac{F_2}{F_1}],$$

where $F_n$ is the solution growing like $|\lambda_2|^n$ where $|\lambda_1| > |\lambda_2|$ the characteristics roots of the recurrence equation.

Notice that

$$(\prod_{i=1}^{n} C_{x,v}(i)^{-1})(\prod_{i=1}^{n} C_{x,v}(i)) = Id$$

Hence, the first column of $\prod_{i=1}^{n} C_{x,v}(i)^{-t}$ is perpendicular to the second column of $\prod_{i=1}^{n} C_{x,v}(i)$. So in the limit, the second column is projectively equal to $[-\frac{F_2}{F_1}, 1]$, hence the limit is equal to $-\frac{F_2}{F_1}$.

In conclusion, given a trajectory matrix $T(n)$ with companion form matrix $C(n)$ that is Poincaré-Perron type, we get that the limit of $C(n)$ is $-\frac{F_2}{F_1}$ and consequently the limit of the trajectory matrix $T(n)$ is $U(1) \cdot \left(-\frac{F_2}{F_1}\right)$ for a solution $F_n$ of the recurrence with the smaller $\lambda_2$ growth rate.

### G.4  Proof for the trajectory (1,1,2)

Taking the trajectory $v = (1, 1, 2)$ we get $F_n = {}_2F_1(\frac{1}{2} + n, \frac{1}{2} + n, \frac{1}{2} + 2n; \frac{1}{2})$. The trajectory matrix and companion form are

$$T(n) = \begin{pmatrix} \dfrac{(4n+5)(4n+7)}{2(n+1)^2} & -\dfrac{(4n+5)(4n+7)^2}{4(n+1)^2} \\[4mm] -\dfrac{(16n+28)(4n+5)^2}{4(n+1)^2(2n+3)^2} & \dfrac{(4n+5)(4n+7)(80n+104)}{(n+1)^2(32n+48)} \end{pmatrix}$$

$$C(n) = \begin{pmatrix} 0 & -\dfrac{(4n+11)(4n+7)(4n+9)^2}{(n+2)^2(2n+5)^2} \\[4mm] 1 & \dfrac{(4n+7)(4n+9)(8n+22)(6n^2+21n+17)}{(n+2)^2(2n+5)^2(4n+5)} \end{pmatrix}$$

With the limit matrix $C(\infty)$ being Poincaré-Perron type

$$C(\infty) = \begin{pmatrix} 0 & -64 \\ 1 & 48 \end{pmatrix}$$

Having the characteristic roots (eigenvalues) $\{24 - 16\sqrt{(2)}, 24 + 16\sqrt{(2)}\}$. All we need to show is that $F_n$ does not grow exponentially like $24 + 16\sqrt{(2)} \approx 46.627$.

By the Euler identity for the hypergeometric function

$$_2F_1(a, b, c; x) = \frac{\Gamma(c)}{\Gamma(a)\Gamma(c-a)} \int_0^1 t^{a-1}(1-t)^{c-a-1}(1-xt)^{-b} \, dt.$$

We get that

$$F(n) = \frac{\Gamma(\frac{1}{2} + 2n)}{\Gamma(\frac{1}{2} + n)\,\Gamma(n)} \int_0^1 t^{n-\frac{1}{2}}(1-t)^{n-1}\left(1 - \tfrac{t}{2}\right)^{-(n+\frac{1}{2})} dt.$$

Setting

$$f(t) = \log\left(\frac{t(1-t)}{1-\frac{t}{2}}\right), \qquad \psi(t) = t^{-1/2}(1-t)^{-1}\left(1-\tfrac{t}{2}\right)^{-1/2},$$

so that

$$_2F_1 = \frac{\Gamma(\frac{1}{2} + 2n)}{\Gamma(\frac{1}{2} + n)\,\Gamma(n)} \int_0^1 e^{nf(t)}\,\psi(t)\,dt.$$

Using Stirling's formula

$$\Gamma(z) \sim \sqrt{2\pi}\, z^{z-\frac{1}{2}} e^{-z}, \qquad (z \to \infty),$$

we find

$$\frac{\Gamma(\frac{1}{2}+2n)}{\Gamma(\frac{1}{2}+n)\,\Gamma(n)} \sim \frac{\sqrt{2\pi}\,(2n)^{2n}e^{-2n}}{\left(\sqrt{2\pi}\,n^n e^{-n}\right)\left(\sqrt{2\pi}\,n^{n-\frac{1}{2}}e^{-n}\right)} = \frac{4^n\sqrt{n}}{\sqrt{2\pi}}\left(1 + O(1/n)\right).$$

Hence, the Gamma prefactor contributes an exponential growth factor $4^n$.

The function $f(t)$ has a maximal value at $t^* \in (0, 1)$

$$t^* = 2 - \sqrt{2}, \qquad f_{\max} = \log(6 - 4\sqrt{2}) = 2\log(2 - \sqrt{2}).$$

Thus

$$\max_{t \in (0,1)} e^{nf(t)} = (6 - 4\sqrt{2})^n.$$

Note that although $\psi(t)$ diverges at the endpoints, the full integrand

$$t^{n-\frac{1}{2}}(1-t)^{n-1}\left(1-\tfrac{t}{2}\right)^{-(n+\frac{1}{2})} = e^{nf(t)}\psi(t)$$

is integrable for every $n$, therefore by the Laplace method, we get:

$$\int_0^1 e^{nf(t)}\psi(t)\,dt \le C\left(6 - 4\sqrt{2}\right)^n$$

for some constant $C > 0$.

Combining the two parts gives

$$F_n \le \frac{4^n\sqrt{n}}{\sqrt{2\pi}}\left(6 - 4\sqrt{2}\right)^n C = \mathrm{poly}(n)\left(24 - 16\sqrt{2}\right)^n,$$

$$\boxed{F_n \le \left(24 - 16\sqrt{2}\right)^n \cdot \mathrm{poly}(n)}$$

The exponential growth rate is thus governed by $\left(24 - 16\sqrt{2}\right)^n$ proving the convergence of the companion form matrix $C(n)$ is to $-\frac{F_2}{F_1}$.

### G.5 Proof for the entire $\pi$-CMF and related hypergeometric constants

To prove the convergence for the entire CMF we need to estimate $F_n$ for each trajectory $(\alpha, \beta, \gamma) \in \mathbb{Z}^3$. We will estimate the exponential growth in a way similar to the process done for direction $(1, 1, 2)$, but keep it general and symbolic. This will give us the exponential growth rate of $F_n$ in each trajectory $(\alpha, \beta, \gamma)$. Next, we will analyze the CMF to estimate the possible growth rates from the eigenvalues and show that the smallest one is equal to the estimated rate of $F_n$ we had before.

#### G.5.1 The growth rates for given directions in the CMF

Recall that the CMF is constructed as a D-finite CMF with respect to bases $\{{}_2F_1, \Theta \cdot_2 F_1\}$. We can conduct a change of basis to get an equivalent CMF with bases $\{{}_2F_1, \frac{y0}{x0x1}\Theta \cdot_2 F_1\}$. This CMF is defined by the matrices:

$$M_{\mathbf{x}} = \begin{pmatrix} 1 & \frac{z}{x+1} \\ y & \frac{2x+y-2z+2}{x+1} \\ z & \end{pmatrix}$$

$$M_{\mathbf{y}} = \begin{pmatrix} 1 & \frac{z}{y+1} \\ \frac{x}{z} & \frac{x+2y-2z+2}{y+1} \end{pmatrix} \tag{26}$$

$$M_{\mathbf{z}} = \begin{pmatrix} \frac{z(-x-y+z)}{(y-z)(x-z)} & \frac{z(z+1)}{(y-z)(x-z)} \\ \frac{xy}{(y-z)(x-z)} & \frac{-z(z+1)}{(y-z)(x-z)} \end{pmatrix}$$

The matrices here now have coefficients of balanced degree 0: When going to infinity in $x, y, z$, only the dominant factors from the denominator and numerator contribute, so at infinity we get these three commuting matrices:

$$\widetilde{M_{\mathbf{x}}} = \begin{pmatrix} 1 & \frac{z}{x} \\ \frac{y}{z} & \frac{2x+y-2z}{x} \end{pmatrix}$$

$$\widetilde{M_{\mathbf{y}}} = \begin{pmatrix} 1 & \frac{z}{y} \\ \frac{x}{z} & \frac{x+2y-2z}{y} \end{pmatrix} \tag{27}$$

$$\widetilde{M_{\mathbf{z}}} = \begin{pmatrix} \frac{z(-x-y+z)}{(y-z)(x-z)} & \frac{z^2}{(y-z)(x-z)} \\ \frac{xy}{(y-z)(x-z)} & \frac{-z^2}{(y-z)(x-z)} \end{pmatrix}$$

These matrices can be mutually diagonalized, set

$$\Delta = x^2 + 6xy - 4xz + y^2 - 4yz + 4z^2 = (x + y - 2z)^2 + 4xy$$

$$S_1 = 3x + y - 2z, \quad S_2 = x + 3y - 2z$$

We have the eigenvalues:

$$\widetilde{M}_{\mathbf{x}} \longrightarrow \frac{S_1}{2x} \pm \frac{\sqrt{\Delta}}{2x} = \{\lambda_{1,-}, \lambda_{1,+}\},$$

$$\widetilde{M}_{\mathbf{y}} \longrightarrow \frac{S_2}{2y} \pm \frac{\sqrt{\Delta}}{2y} = \{\lambda_{2,-}, \lambda_{2,+}\},$$

$$\widetilde{M}_{\mathbf{z}} \longrightarrow -\frac{z(x+y)}{2(x-z)(y-z)} \pm \frac{z\sqrt{\Delta}}{2(x-z)(y-z)} = \{\lambda_{3,-}, \lambda_{3,+}\}.$$

Hence

$$\begin{cases} |\lambda_{1,-}| \le |\lambda_{1,+}| & \text{if } \Delta = 0 \text{ or } S_1 \ge 0, \\ |\lambda_{2,-}| \le |\lambda_{2,+}| & \text{if } \Delta = 0 \text{ or } S_2 \ge 0, \\ |\lambda_{3,-}| \le |\lambda_{3,+}| & \text{if } \Delta = 0 \text{ or } z = 0 \text{ or } x + y \le 0. \end{cases}$$

The growth rates of a trajectory matrix in direction $v = (\alpha, \beta, \gamma)$ are exactly the eigenvalues of $T_v(\infty)$, which is the limit of the trajectory matrix $T_{x',v}(n)$ with $n$ going to infinity. By our construction of the "infinity matrices" $\widetilde{M}$, we have that

$$T_v(\infty) = \tilde{M}_x(\alpha, \beta, \gamma)^\alpha \cdot \tilde{M}_y(\alpha, \beta, \gamma)^\beta \cdot \tilde{M}_z(\alpha, \beta, \gamma)^\gamma$$

If we assume, for example, that

$$S_1((\alpha, \beta, \gamma) \le 0, \ S_2((\alpha, \beta, \gamma) \le 0, \ \alpha + \beta \ge 0$$

We get that the minimal growth is given by

$$B(\alpha, \beta, \gamma) = \lambda_{1,+}(\alpha, \beta, \gamma)^\alpha \cdot \lambda_{2,+}(\alpha, \beta, \gamma)^\beta \cdot \lambda_{3,+}(\alpha, \beta, \gamma)^\gamma \tag{28}$$

### G.5.2    The growth rates for the $F_n$ in each direction

We consider the Gauss hypergeometric function in the scaling form

$$F(n; x) = {}_2F_1(a, b; c; x) \qquad \text{with} \qquad a = a_0 + \alpha n, \quad b = b_0 + \beta n, \quad c = c_0 + \gamma n,$$

for fixed real slopes $(\alpha, \beta, \gamma)$ and a fixed $x \in (0, 1)$. As $n \to \infty$, the asymptotic growth of $F_n$ is governed by a Laplace–type integral representation, and takes the general form

$$|F(n; x)| \asymp B(\alpha, \beta, \gamma; x)^n,$$

where $B(\alpha, \beta, \gamma; x) > 0$ is the *exponential base*.

The base $B$ depends on which integral representation (*Euler–b, Euler–a, or Kummer at $x \approx 1$*) is valid, and whether the dominant contribution comes from an *interior saddle* or an *endpoint*. For $x = \frac{1}{2}$ these regimes can be described explicitly as follows.

Assume

$$\gamma > \alpha, \qquad \gamma \ge \beta \ge 0,$$

In this regime, the integral

$$_2F_1(a, b; c; x) = \frac{\Gamma(c)}{\Gamma(b)\Gamma(c-b)} \int_0^1 t^{b-1}(1-t)^{c-b-1}(1-xt)^{-a} \, dt$$

is convergent and admits an interior stationary point $t^* \in (0, 1)$.

The phase function is

$$\Phi_1(t) = \beta \log t + (\gamma - \beta) \log(1 - t) - \alpha \log(1 - xt),$$

and for $x = \frac{1}{2}$ the saddle equation $\Phi_1'(t) = 0$ has the exact solution

$$t^* = \frac{-(\alpha - \beta - 2\gamma) - \sqrt{\Delta}}{2(\gamma - \alpha)}, \qquad \Delta = \alpha^2 + 6\alpha\beta - 4\alpha\gamma + \beta^2 - 4\beta\gamma + 4\gamma^2.$$

Substituting $t^*$ into the integrand and adding the Stirling rate coming from the Gamma prefactors

$$S_b = \gamma \log|\gamma| - \beta \log|\beta| - (\gamma - \beta)\log|\gamma - \beta|$$

yields the simplified closed form

$$\boxed{\begin{aligned} B_{\text{int}}(\alpha, \beta, \gamma) &= \left(\frac{3\alpha + \beta - 2\gamma - \sqrt{\Delta}}{2\alpha}\right)^{\alpha} \left(\frac{\alpha + 3\beta - 2\gamma - \sqrt{\Delta}}{2\beta}\right)^{\beta} \\ &\times \left(\frac{-\gamma(\alpha + \beta - \sqrt{\Delta})}{2(\alpha - \gamma)(\beta - \gamma)}\right)^{\gamma}. \end{aligned}}$$

Thus, in this regime,

$$|y_1(n; \tfrac{1}{2})| \asymp B_{\text{int}}(\alpha, \beta, \gamma)^{\,n}.$$

Which is equal to the function $B(\alpha, \beta, \gamma)$!

To see this, recall that

$$t^* = \frac{-(\alpha - \beta - 2\gamma) - \sqrt{\Delta}}{2(\gamma - \alpha)}.$$

Then

$$1 - t^* = \frac{S}{2(\gamma - \alpha)}, \qquad 1 - \frac{t^*}{2} = \frac{L_3}{4(\gamma - \alpha)},$$

where we introduce the linear forms $(s = -\sqrt{\Delta})$

$$L_1 := 3\alpha + \beta - 2\gamma + s, \quad L_2 := \alpha + 3\beta - 2\gamma + s, \quad L_3 := (2\gamma - 3\alpha - \beta) + s, \quad S := -s - (\alpha + \beta).$$

*Step 1: LHS in linear forms.* From the definitions,

$$e^{\Phi(t^*)} = \frac{(t^*)^{\beta}(1 - t^*)^{\gamma - \beta}}{(1 - \frac{t^*}{2})^{\alpha}} = \frac{\left[-(\alpha - \beta - 2\gamma) - s\right]^{\beta} S^{\gamma - \beta}}{L_3^{\alpha}} \cdot \frac{(4(\gamma - \alpha))^{\alpha}}{(2(\gamma - \alpha))^{\gamma}}.$$

Multiplying by the Gamma prefactor gives

$$\text{LHS} = \frac{\left[-(\alpha - \beta - 2\gamma) - s\right]^{\beta} S^{\gamma - \beta}}{L_3^{\alpha}} \cdot \frac{(4(\gamma - \alpha))^{\alpha}}{(2(\gamma - \alpha))^{\gamma}} \cdot \frac{\gamma^{\gamma}}{\beta^{\beta}(\gamma - \beta)^{\gamma - \beta}}.$$

*Step 2: RHS in linear forms.* By definition of the $\lambda$'s,

$$\text{RHS} = (\lambda_{1,+})^{\alpha}(\lambda_{2,+})^{\beta}(\lambda_{3,+})^{\gamma} = \frac{L_1^{\alpha} L_2^{\beta} S^{\gamma}}{(2\alpha)^{\alpha}(2\beta)^{\beta}\left[2(\gamma - \alpha)(\gamma - \beta)\right]^{\gamma}} \, \gamma^{\gamma}.$$

*Step 3: Reduce to two linear identities.* Cancel $\gamma^{\gamma}$ and the common factor $S^{\gamma - \beta}$. We need to show

$$\frac{\left[-(\alpha - \beta - 2\gamma) - s\right]^{\beta}}{L_3^{\alpha}} \cdot \frac{(4(\gamma - \alpha))^{\alpha}}{(2(\gamma - \alpha))^{\gamma}} \cdot \frac{1}{\beta^{\beta}(\gamma - \beta)^{\gamma - \beta}} = \frac{L_1^{\alpha} L_2^{\beta} S^{\beta}}{(2\alpha)^{\alpha}(2\beta)^{\beta}\left[2(\gamma - \alpha)(\gamma - \beta)\right]^{\gamma}}.$$

After clearing constants this is equivalent to

$$\left[-(\alpha - \beta - 2\gamma) - s\right]^{\beta}\left(8\alpha(\gamma - \alpha)\right)^{\alpha} 2^{\beta}(\gamma - \beta)^{\beta} = L_1^{\alpha} L_2^{\beta} S^{\beta} L_3^{\alpha}. \qquad (*)$$

It suffices to establish the following two identities (both linear or quadratic in $s$):

$$\boxed{\text{(I)} \quad \left[-(\alpha - \beta - 2\gamma) - s\right] \cdot 2(\gamma - \beta) = L_2 \, S,}$$

$$\boxed{\text{(II)} \quad L_1 \, L_3 = 8 \, \alpha(\gamma - \alpha).}$$

*Verification of (I).* Expand the right-hand side:

$$L_2 \, S = (\alpha + 3\beta - 2\gamma + s)(s - \alpha - \beta) = s^2 - (\alpha + \beta)(\alpha + 3\beta - 2\gamma) + s(\alpha + 3\beta - 2\gamma - \alpha - \beta).$$

Using $s^2 = \Delta = (\alpha + \beta - 2\gamma)^2 + 4\alpha\beta$, a short simplification yields

$$L_2 S = 2(\gamma - \beta)\left(-\alpha + \beta + 2\gamma - s\right) = 2(\gamma - \beta)\left[-(\alpha - \beta - 2\gamma) - s\right],$$

which is exactly (I).

*Verification of (II).* Note $L_3 = (2\gamma - 3\alpha - \beta) + s = -(3\alpha + \beta - 2\gamma) + s$. Hence

$$L_1 L_3 = (s + 3\alpha + \beta - 2\gamma)(s - (3\alpha + \beta - 2\gamma)) = s^2 - (3\alpha + \beta - 2\gamma)^2.$$

But

$$(3\alpha + \beta - 2\gamma)^2 = (\alpha + \beta - 2\gamma + 2\alpha)^2 = (\alpha + \beta - 2\gamma)^2 + 4\alpha(\alpha + \beta - 2\gamma) + 4\alpha^2.$$

Therefore,

$$L_1 L_3 = \left[(\alpha + \beta - 2\gamma)^2 + 4\alpha\beta\right] - \left[(\alpha + \beta - 2\gamma)^2 + 4\alpha(\alpha + \beta - 2\gamma) + 4\alpha^2\right] = 8\,\alpha(\gamma - \alpha),$$

which is (II). $\qquad\square$

**Example.** For $(\alpha, \beta, \gamma) = (1, 1, 2)$ one obtains $B = B_{\text{int}} = 24 - 16\sqrt{2}$, matching direct evaluation.

And for example over the point $(\alpha, \beta, \gamma) = (1, 1, 3)$ one obtains $B = B_{\text{int}} = (-\frac{297}{4} + 135 \cdot \frac{\sqrt{5}}{4})$.

### G.5.3 Proof method

For each regime of the parameters $\alpha, \beta, \gamma$, one can establish the growth rates of the hypergeometric function and the possible eigenvalues of the trajectory matrix at infinity. When the growth rate of the hypergeometric match the value of the slowest eigenvalue we have by Lemma 3 that the limit is a ratio of the Gauss hypergeometric function.

This procedure is can be dome for $_2F_1(a_0 + \alpha \cdot n, b_0 + \beta \cdot n, c_0 + \gamma \cdot n; x)$ and the correspondent values of with depend only on $\alpha, \beta$ and $\gamma$ so using the same CMF for another initial point $(a_0, b_0, c_0)$ one can get a unified proof that the trajectory converges to the ratio of the hypergeometric functions. In our case, with initial points being half integers, the ratio is always a rational function of $\pi$ but we get infinitely many new proving formulas for any other constant that is a ratio of Gauss Hypergeometrics (such as $\log(2)$ if we take integer initial points), some values even have a positive irrationality measure!

### G.5.4 General D-finite CMF

This above process can be done for a general D-finite CMF (see [58]). We can base-change the CMF to have balanced degree coefficients and symbolically find the eigenvalues in each trajectory. We can also apply analysis to the D-finite function the CMF is built from in order to estimate the growth rate; all we need to do is see that the smallest eigenvalue matches the growth rate of the D-finite function.

### G.6 Scanning the dynamical metrics of the $\pi$-CMF

Fig. 7 and Fig. 8 visualize the dynamical metrics of the $\pi$-CMF on a unit sphere.

The *normalized convergence rate* is defined $r/\ell^1(t)$, where $r$ is the convergence rate from Eq. (4), $t$ is a trajectory and $\ell^1$ is the $\ell^1$-norm. It allows an unbiased comparison between formulas arising from the CMF. The need for such an unbiased convergence parameter for CMF formulas stems from the artificial convergence acceleration that taking multiples of trajectories can cause: if trajectory $(1, 2, 3)$ has (exponential) convergence rate $r$, then trajectory $(2, 4, 6)$ will have convergence rate $2r$. The convergence performance of these parallel trajectories, though, remains constant when measured by the normalized convergence rate. This enables a well-defined convergence measurement for each *direction*.

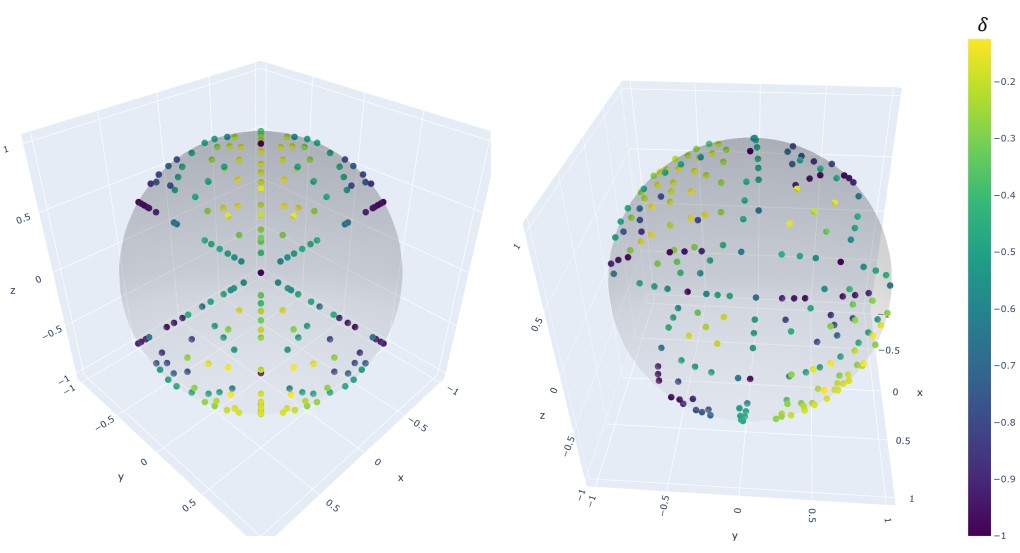

Figure 7: The irrationality measure $\delta$, Eq. (5), for different directions in the $\pi$-CMF.

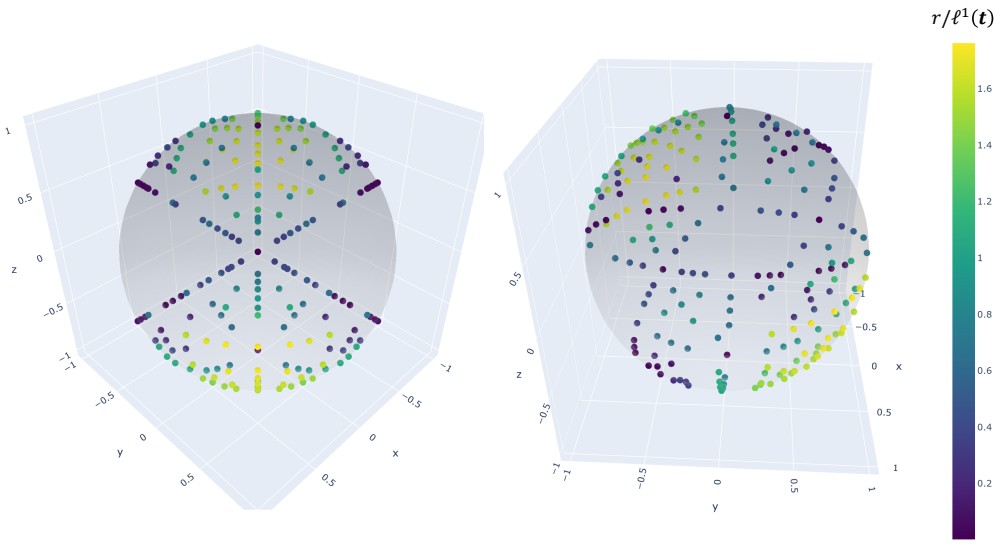

Figure 8: The normalized convergence rate $r/\ell^1(t)$ for different trajectories $t$ in the $\pi$-CMF.

# H    LLM formula equivalence detection and proving

LLMs were independently asked whether formula pairs proven equivalent by our system are equivalent (equivalence detection) and subsequently tasked with proving equivalence (equivalence prooving), regardless of the answer to the first question.

## H.1    Test formula pairs

The following equivalent formula pairs, proven by our system, were given to GPT-4o and Gemini 2.5 Pro Preview for detection and proof: (1,2), (3,4), (1,5) from Table 1; (9,10), (63,66), (75,76) from Table 16; and (82,84), (101,103), (120,121), (140,142) from Table 17. Detection is binary and easily checked, while proofs were checked manually (recall Table 2). The full results are shown in Table 11.

Table 11: **Detailed outcomes of LLM equivalence detection and proving attempts.** The three segments of the table respectively contain formulas from Tables 1, 16 and 17. Gemini 2.5 Pro Preview is remarkable in proving 50%, sometimes by finding Wilf-Zeilberger pairs [45], but still falls short of the full challenge.

| Formula pair | Note | GPT-4o detect | GPT-4o prove | Gemini 2.5 Pro Preview detect | Gemini 2.5 Pro Preview prove |
|---|---|---|---|---|---|
| (1,2) | equal term by term | - | - | - | + |
| (3,4) | partial fraction decomposition | - | + | + | + |
| (1,5) | | - | - | - | - |
| (9,10) | | - | - | + | - |
| (63,66) | equal term by term up to added constant | - | - | + | - |
| (75,76) | | - | - | + | - |
| (82,84) | Ramanujan-Sun pair from Section 5 | - | - | + | - |
| (101,103) | | - | - | + | + |
| (120,121) | direct fold by 2 (see Appendix E.5) | + | + | + | + |
| (140,142) | | - | - | + | + |

## H.2    Equivalence detection and proving prompts

**Detection - system message:**

> You are an equivalence detector. Your task is to decide whether two given formulas are mathematically equivalent.
> Two formulas are equivalent if one formula implies the other, and vice versa. In other words, you need to determine whether the following statement is true: Formula A converges to Value A if and only if Formula B converges to Value B.
> You are not required to prove the equivalence—only to determine whether the equivalence appears to hold. Base your decision on mathematical structure, known identities, or other relevant patterns. Respond with a clear answer: Yes (equivalent) or No (not equivalent).

**Detection - user message:**

> Two formulas are said to be equivalent if one holds if and only if the other does. Specifically, Formula A converges to and equals value A if and only if Formula B converges to and equals value B.
> In other words, the truth of one formula guarantees the truth of the other, and vice versa.
> Given the following two formulas:
> FORMULAA_VALUEA
> FORMULAB_VALUEB
> Are these formulas equivalent?

**Proving - system message:**

> You are an equivalence prover. Your task is to determine whether two given formulas are mathematically equivalent. That is, proving one formula should be sufficient to establish the other.
> In other words, your goal is to rigorously assess and demonstrate that:
> If formula A converges to a value A, then formula B must converge to value B, and vice versa.
> Do not assume equivalence—justify it. Use mathematical reasoning, transformations, or known identities to support your argument.
> The proof of equivalence should be rigorous and detailed.

**Proving - user message:**

> Two formulas are said to be equivalent if one holds if and only if the other does. Specifically, Formula A converges to and equals value A if and only if Formula B converges to and equals value B.
> In other words, proving the validity of one formula must be sufficient to establish the validity of the other.
> Given the following two formulas:
> FORMULAA_VALUEA
> FORMULAB_VALUEB
> These formulas are equivalent. Please provide a rigorous and detailed justification, a complete proof of equivalence.

# I Formula harvesting details

## I.1 Article retrieval

arXiv's search API was not reliable for retrieving papers with $\pi$ formulas. Some simple queries such as "formula for pi" or "$\pi$ formula" returned few results (and if the search method is not set to word-for-word results are mostly irrelevant). Not knowing all patterns in which $\pi$ tends to be calculated in, we went with a more exhaustive approach.

455,050 articles from the following categories which were indexed in the arXiv metadata dataset [5] as of 24 November, 2024, were scraped: math.CA, math.NT, math.PR, math.CO, math.GM, math.HO, cs.AI, cs.LG and cs.DC.

## I.2 LaTeX equation patterns and preprocessing

The following unnumbered or inline LaTeX math environments were scraped from all articles:

- `$ · $`,
- `$$ · $$`,
- `\[ · \]`
- `\( · \)`
- `math`

The following LaTeX equation environments were scraped from all articles:

- `equation`
- `align`
- `gather`
- `multline`
- `alignat`

- eqnarray

Starred (*) versions of the latter equation environments were also collected, for a total of 17 environments.

Equation environments were kept so strings with multiple equations could be split into distinct formulas during preprocessing. Preprocessing was mainly aimed at removing text within equations and setting uniform symbols for objects like `\ddots`, `\cdots`.

### I.3 Formula retrieval patterns

Table 12: **LaTeX formula patterns.** Each pattern was paired with both "$\pi =$" and "$= \pi$," and the `\cfrac`-based variants of the `\frac`-containing regular expressions were also included, resulting in a total of 10 patterns.

| Pattern for - | Python **re** pattern |
| --- | --- |
| Series | \ sum \ s*_{(?s:.)*}\ s*\ char˜ˆ s* |
| Sum of \ frac | (\ s*\ frac\ s*{\ s*[^{}]*\ s*}\s*{\ s*[^{}]*\ s*}) |
| | ((?:\ s*\+\ s*\ frac\ s*{\ s*[^{}]*\ s*}\ s*{\ s*[^{}]*\ s*})+) |
| Nested \ frac | (\ frac\ s*{\ s*[^{} ]*\ s*}\ s*{\ s*[^{} ]*) |
| | ((?:\ frac\ s*{\ s*[^{} ]*\ s*}\ s*{\ s*[^{} ]*}+\ s*}\ s*}) |

In addition to positive regular expressions (Table 12), presence of any of the following strings in preprocessed data halted processing: `sqrt, tan, cos, sin, log, ln, zeta, pi^`

### I.4 Formula classification and extraction

LLMs were run via API—GPT-4o with the OpenAI API, Claude 3.7 Sonnet with Anthropic's API and Gemini 2.5 Pro Preview through Google AI Studio.

As a formula is extracted, information collected from previous prompts is appended to the following prompts to reinforce context. See Appendix I.6 for the prompts used. All prompts to the LLM use a temperature of 0 to promote consistency. The OpenAI GPT API supports a return format called Structured Outputs, which guarantees a json schema of choice is returned by the LLM. Formula candidates are passed to the LLM for zero-shot binary classification, into classes: formulas that compute the constant $\pi$ and formulas that do not. Candidates are then passed again for trinary classification into classes: series, continued fraction (the targets of this study), and other formulas (which are either not $\pi$ formulas or are but are not series or continued fractions). The third class is intentionally redundant, as some strings which GPT-4o mini gets wrong may be correctly filtered out by GPT-4o: out of 3367 strings classified as $\pi$-computing series or continued fractions by GPT-4o mini, 1711 were discarded by GPT-4o, leaving 1656. All examples which were checked manually were false positives, discarded when double checked by GPT-4o. Example: $g(x) = \frac{\pi}{2} + \sum_{j=1}^{\infty} a_n x^n$ (arXiv 2206.11256).

The vast majority of formulas classified as $\pi$ formulas by GPT-4o were series rather than nested fractions (1,591 vs 65). In some instances the LLM successfully generalized sequences of numbers to polynomial expressions, see Table 13. The cost for API calls for the entire analysis—classification and extraction—was under \$50.

In the future, additional formulas could be gathered by allowing the LLM to decide whether to call a polynomial fitting function on such sequences.

### I.5 Formula validation

Validating the collected formulas presents a challenge. The safest way to validate that a formula converges to the expected constant is by computing the formula (attempting to automatically reconstruct

Table 13: OpenAI's GPT-4o extracts the partial numerator and partial denominator of a continued fraction. Explanation is by the LLM. LaTeX taken from [46] (arXiv 1907.00205).

| LaTeX | 3 + \frac{1 * 3}{5 + \frac{2 * 4}{7 + \frac{3 * 5}{9 + \frac{4 * 6}{11 + ...}}}} = 3\frac{1}{_{2}F_{1} ( 1,\frac{1}{2}; \frac{5}{2}; - 1 ) } = \frac{4}{\pi - 2} |
|---|---|
| Rendered | $3 + \cfrac{1 \cdot 3}{5 + \cfrac{2 \cdot 4}{7 + \cfrac{3 \cdot 5}{9 + \cfrac{4 \cdot 6}{11 + \dots}}}} = 3 \cdot \frac{1}{{}_2 F_1\left(1, \frac{1}{2}; \frac{5}{2}; -1\right)} = \frac{4}{\pi - 2}$ |
| Prompt | Identify the partial denominator an and partial numerator bn. Extract each of them and write them as proper SymPy expressions as a function of depth n. |
| Output | $a_n$ = 2*n + 3, $b_n$ = n * (n + 2) |
| Explanation | The partial denominator an is the sequence 5, 7, 9, 11,... which can be expressed as 2*n + 3. The partial numerator bn is given by the sequence 1*3, 2*4, 3*5, 4*6,... which can be expressed as n*(n + 2). |

and verify a proof in a formal language is beyond the scope of this study). As exemplified in Table 14, when prompted for the value of a formula, GPT-4o sometimes omitted free constants, multipliers, or simple arithmetic adjustments required for the proposed value to match the true value of the formula.

Table 14: A typical extraction by OpenAI's GPT-4o: The LLM extracts the partial numerator and partial denominator of a continued fraction, but fails to find the correct value of the formula it collects, which needs to be isolated through simple arithmetic; the correct value is $\left(\frac{2}{1 - \frac{\pi}{4} \cdot \frac{3 \cdot 3 \cdot 5 \cdot 5}{2 \cdot 4 \cdot 4 \cdot 6}} - 1\right) = \frac{256 + 75\pi}{256 - 75\pi}$. LaTeX taken from arXiv 1806.03346.

| LaTeX | \frac{\pi}{4} = \frac{2}{3} * \frac{4}{3} * \frac{4}{5} * \frac{6}{5} * [ 1 - \cfrac{2}{25 + \cfrac{1 * 3}{24 + \cfrac{3 * 5}{24 + \cfrac{5 * 7}{24 + ... }}}} ] |
|---|---|
| Rendered | $\frac{\pi}{4} = \frac{2}{3} \cdot \frac{4}{3} \cdot \frac{4}{5} \cdot \frac{6}{5} \cdot \left( 1 - \cfrac{2}{25 + \cfrac{1 \cdot 3}{24 + \cfrac{3 \cdot 5}{24 + \cfrac{5 \cdot 7}{24 + \dots}}}} \right)$ |
| Prompt | Extraction pipeline (Appendix I.6) with an added prompt for identifying the value. |
| Output | $a_n$ = 24, $b_n$ = (2n-1)*(2n+1), value = pi / 4. |

PSLQ [26] addresses this by finding the correct Möbius transformation between the constant of interest and the formula's value, relying solely on the algebraic expression extracted by the LLM. Given the prior belief that a formula computes a constant and that a nontrivial integer relation exists between the formula's empirical value and the constant of interest, the integer relation found is likely the correct one. The validity of collected formulas is strengthened further when equivalence proofs are found between them.

After classifying and extracting, GPT-4o still produces some expressions that do not compute $\pi$. The PSLQ-validation stage removes these false positives by finding the precise limit of each formula in terms of $\pi$. Of the 660 candidates, 147 are not Diophantine formulas (e.g. they contain $\pi$ itself in the series term), leaving 535 formulas. 150 of these did not pass validation, and many of these were formulas that do not compute $\pi$ like $-\frac{4}{\pi} \sum_{n=0}^{\infty} \frac{(-1)^n}{(2n+1)^2} = -\frac{4G}{\pi}$ (arXiv 1906.04927), and

formulas with mistakes like $\sum_{i=1}^{\infty} \frac{1}{i^2} = \frac{\pi}{6}$ (arXiv 1811.05831, under proofs.tex). Example for a non-Diophantine formula for $\pi$: $\sum_{n=0}^{\infty} \frac{(\frac{1}{2})_n (\frac{1}{3})_n (\frac{2}{3})_n}{(1)_n^3} (\frac{2}{27})^n = \frac{3\pi}{4\,\Gamma^2(\frac{2}{3})\Gamma^2(\frac{5}{6})}$ (arXiv 2001.08104), as $\pi$ can be computed only when given the irrational denominator.

## I.6   Prompts for harvesting formulas

We utilized the Structured Output feature of OpenAI's models and used a temperature of 0 for all prompts. The return schemas are included with the prompts. LaTeX from prompts is rendered for readability.

### I.6.1   Classification prompts

Initial classification by GPT-4o mini:

**System message:**

> You are a model that classifies whether a LaTeX string is a formula that can be rearranged to calculate the constant
> {constant}. Specifically, we are interested in continued fractions and series.

**User message:**

> Is this a continued fraction or a series that can be rearranged to calculate the constant {constant}? {latex_string}
>
> Structured output: boolean.

### I.6.2   Extraction prompts

For added context during extraction, the second classification query shown in Fig. 3c is actually conducted during the extraction stage. Values in source strings were not collected (see Appendix I.5).

**System message:**

> You are a model that extracts formula information from a LaTeX string.
>
> Your task is to:
> a. Classify the type of formula: series, continued fraction, or neither.
> b. Extract its components and identify the variable.
>
> This information will be used to compute the formula later, so it is critical that the extracted value and components are accurate to ensure correctness.
>
> You will be asked separately about each of the following steps:
> Step 1. Classify the formula: Determine whether the LaTeX string represents a series or a continued fraction that can be rearranged to calculate the constant {constant}.
> Step 2. Identify the formula type: Specify whether it is a series or a continued fraction.
> Step 3. Extract the formula components:
> - For series: Identify the term and the start value.
> - For continued fractions: Identify the partial numerator and partial denominator.
> Step 4. Identify the variable of the formula: Clearly state the variable used in the formula.

**User message: step 1**

> Step 1:
> Is this formula a series or a continued fraction that can be rearranged to calculate the constant
> {constant}?
>
> {latex_string}
>
> Structured output: boolean.

**User message: step 2**

> Step 2:
> Determine the type of formula.
> Is this formula a continued fraction or a series?
>
> {latex_string}
>
> Structured output: 'series' or 'cf' (for continued fraction).

**User message: step 3**

The prompt in step 3 depends on the classification result.

If `formula_type == 'cf'` (continued fraction)

(LaTeX for one-shot example taken from [46] (arXiv 1907.00205).)

> Step 3:
> The formula is a continued fraction. Identify the following components:
> 1. The partial denominator (`an`) as a function of depth (`n`).
> 2. The partial numerator (`bn`) as a function of depth (`n`).
> 3. Any unknown variables or expressions (other than the depth n).
>
> Write each component as a proper SymPy expression. For example:
> The string
> $$\forall z \in \mathbb{C}: \quad 1 + \cfrac{1 \cdot (2 \cdot z - 1)}{4 + \cfrac{2 \cdot (2 \cdot z - 3)}{7 + \cfrac{3 \cdot (2 \cdot z - 5)}{10 + \cfrac{4 \cdot (2 \cdot z - 7)}{13 + \dots}}}} = \frac{2^{2 \cdot z + 1}}{\pi \binom{2 \cdot z}{z}}$$
> has the following:
> - `an`: '3*n + 1'
> - `bn`: 'n*(2*z - (2*n - 1))'
> - `unknowns`: ['z']
>
> The continued fraction is:
> {latex_string}
>
> Structured output: {'an': str, 'bn': str, 'unknowns': list[str]}

If `formula_type == 'series'`

(LaTeX for one-shot example adapted from arXiv 1806.03346. Note the adapted formula is incorrect as the unknown was inserted at random.)

Step 3:
The formula is a series. Identify the following components:
1. The term as a SymPy expression.
2. The dummy variable.
3. The start value of the dummy variable.
4. Any unknown variables (other than the dummy variable).

For example:
The string

$$\pi \cdot z = \frac{22}{7} - 24 \sum_{n=2}^{\infty} \frac{(-1)^n}{(2n+1+z)(2n+2+z)(2n+3)(2n+4)(2n+5)}$$

has the following:
- `Term: '(-1)**n / ((2*n + 1)*(2*n + 2)*(2*n + 3)*(2*n + 4)*(2*n + 5))'`
- `Dummy variable: 'n'`
- `Start: '2'`
- `Unknowns: ['z']`

Pay attention to special symbols like `_symbol` (e.g., `\frac12)_n` ), which often indicate a SymPy `RisingFactorial`. Another symbol to look out for is `H_`, which often means a SymPy `harmonic`.

The series is:
`{latex_string}`

Structured output: {'term': str, 'dummy_var': str, 'start': int, 'unknowns': list[str]}.

**User message: step 4**

(LaTeX for one-shot example taken from arXiv 1806.03346)

Step 4:
Identify the variable used in the formula.
If the formula is a series, focus on the variable used in the outermost summation.
If the formula contains nested summations or other variables, ensure you extract only the variable from the outermost summation and exclude all others.

For example:
The string

$$\pi = \frac{22}{7} - 24 \sum_{n=2}^{\infty} \frac{(-1)^n}{(2n+1)(2n+2)(2n+3)(2n+4)(2n+5)}$$

has the outermost summation variable: `'n'`.

Extract the variable from the formula:
`{latex_string}`.

Structured output: str.

### I.6.3   Code-correcting prompts

After extraction, each of the code components extracted in steps 3, 4 from the above prompts went through an execution test. Faulty code was sent back to GPT-4o for correction to SymPy code that runs properly. A total of only 16 code-correction iterations were needed during a run on 847 classified formulas, a testament to the LLM's ability to write executable code. We are confident that this stage could be removed in future runs with minimal consequences to the size of the formula dataset. Since

the number of corrections is so low, the cost of leaving this stage in is minimal. In short, the correction stage is largely insignificant when using GPT-4o to derive executable code from LaTeX.

The following prompts were used in the loop for up to three iterations:

**System message:**

> You are a helpful assistant tasked with extracting mathematical expressions from strings and rewriting them in proper SymPy format.
> Your output must be valid Python code that can be executed without errors.
> Always focus on processing the original string provided and ensure the response contains only the corrected SymPy expression, formatted as executable Python code.

**User message:**

(Includes the Python error message from a failed execution attempt - e.)

> The last attempt was invalid SymPy code: {`str(e)[:400]`}.
> Last attempt:
> {`string`}
> Task:
> 1. Extract the expression from the **original string** below.
> 2. Rewrite it in proper SymPy format as valid, executable Python code.
> 3. Only return the corrected SymPy expression, formatted as valid Python code.
> Original string:
> {`original_string`}
> Process the **original string** and provide the corrected SymPy expression.
>
> Structured output: str.

# J  Tables of full results

This section contains the full list of order-2 canonical forms (PCFs) derived by our system, grouped in equivalence classes discovered by running the matching algorithm (per Appendix C.1). Formula sources are shown in Table 15. Tables 16 and 17 list the formulas that have been unified by the $\pi$ CMF and those that have not yet been unified, respectively. We believe wider scans of the CMF (Appendix C.5), with deeper, more complicated trajectories, will unify additional formulas from the latter. Enlarging the $\pi$-CMF to higher dimensions and rank will also likely increase the unification percentage.

We invite the reader to explore the formulas and algorithms leading to these tables, available in our project repository https://github.com/RamanujanMachine/euler2ai. Some results are also discussed in the online algorithm demonstration [4].

The supplementary material accompanying this paper contains two json files:

- `pcfs.json`: The full dataset of order-2 canonical forms (PCFs) harvested, along with their sources. 149 PCFs in total.
  Columns:
    - `ab`: list containing $a_n, b_n$ of the PCF, both as strings.
    - `a`: just $a_n$.
    - `b`: just $b_n$.
    - `limit`: string of the symbolic limit of the PCF in terms of $\pi$.
    - `delta`: the empirical irrationality measure of the formula, Eq. (5), as a float.
    - `convergence_rate`: the empirical convergence rate of the formula, Eq. (4), as a float.
    - `sources`: list of dictionaries, each describing a harvested formula that led to the PCF when canonicalized.
      Keys:
        * `type`: 'cf' or 'series'.
        * `formula`: as a string.
        * `formula_limit`: in terms of $\pi$, as a string.
        * `id`: the arXiv id of the source paper, string.
        * `file`: TEX file in which the formula was located, string.
        * `line`: exact line number in the TEX file where the formula was found, int.
        * `equation`: the source equation string.
- `cmf_pcfs.json`: A dataset of order-2 canonical forms (PCFs) sampled from the $\pi$-CMF. 1693 PCFs in total.
  Columns:
  same as `pcfs.json` except for
    - `sources`: list of 2-tuples where each tuple contains a trajectory and a starting point in the CMF, both lists. Generating a trajectory matrix using these (trajectory, starting point) pairs and converting to companion form yields the PCF (Appendix C.5). Trajectories are lists of ints. Starting points are lists of strings for easy storage of rational numbers ($\frac{a}{b}$ is stored as 'a_b').

Table 15: **Formula sources**. Summary of papers from the literature containing $\pi$ formulas, showcasing the success rate of UMAPS (colored indices). Canonical forms are shown in gold if connected to at least one other canonical form by UMAPS and in cyan if unified by the CMF; 98% of papers with $\pi$ formulas had at least one formula connected via canonicalization and UMAPS (gold or cyan) and 50% included a formula unified by the CMF (cyan). Black indices represent formulas that remain unconnected to to other formulas via UMAPS. See Table 4 for statistics in terms of formula counts. Formula indices are consistent with Tables 16,17. Canonical forms 150-153 are not included in this table as they are order-3 recurrences (see Appendix B.6).

\* Tabulated formulas from the data provided in this paper were added manually.

| arXiv source | Canonical forms | arXiv source | Canonical forms | arXiv source | Canonical forms |
|---|---|---|---|---|---|
| 1907.00205* | 1, 2, 3, 13, 20, 21, 22, 28, 29, 30, 31, 32, 48 | 2305.14995 | 44 | 1807.07394 | 96 |
| 2308.11829 | 4 | 2307.05607 | 44 | 1906.07384 | 96 |
| 2307.03086 | 5, 6, 7, 87, 96, 107, 137 | 2307.08063 | 44 | 1908.05123 | 96, 126 |
| 1407.8465 | 7 | 2312.17402 | 44 | 2305.00498 | 96, 126, 137 |
| 2204.08275 | 8, 9, 10, 11, 38, 39, 40, 41, 86, 87 | 2409.06658 | 44 | 2310.04642 | 96 |
| 2405.02776 | 12, 130, 144 | math/0006141 | 44 | 1008.3171 | 106, 113 |
| 2412.12361* | 14, 15, 17, 19, 23, 24, 25, 26, 27, 33, 34, 35, 36, 37 | math/0206179 | 44 | 2001.08104 | 108, 137, 140 |
| 0707.2124 | 16 | math/0402462 | 44 | 1209.2348 | 113 |
| 1507.01703 | 16, 18, 118 | 2305.14367 | 45 | 1302.2898 | 113 |
| 2112.00622 | 16 | 2105.11771 | 49 | 1910.04328 | 113 |
| 2211.11484 | 16, 96, 126 | 2206.08284 | 49 | 2103.07872 | 113, 115, 116 |
| 2310.03699 | 42 | 1601.03180 | 51 | 2203.02631 | 113 |
| 1806.03346 | 43, 44, 45, 46, 47, 48, 50, 51, 52, 59, 60, 61, 62, 63, 64, 65, 66, 67, 68, 69, 70, 71, 72 | 1806.08411 | 51 | 2205.08617 | 113 |
| 0707.2122 | 44 | 1209.5739 | 53 | 2208.07696 | 113 |
| 0707.2500 | 44 | 2411.00280 | 55 | 2305.04935 | 113 |
| 0708.2564 | 44 | 1504.01028 | 73, 126 | 2108.12796 | 114, 115 |
| 0806.0150 | 44 | 1804.08210 | 73, 74, 78, 79 | 0911.2415 | 117 |
| 0807.0872 | 44, 108, 113, 118, 120, 126 | 1805.06568 | 73, 74, 78, 79, 81, 148 | 1103.3893 | 117 |
| 1206.3431 | 44 | 2005.04672 | 73 | 1110.5308 | 117 |
| 1209.3657 | 44 | 2111.10998 | 73, 75 | 1804.00394 | 117 |
| 1301.2584 | 44 | 2204.04535 | 73, 75, 76 | 0708.3307 | 126 |
| 1302.0471 | 44 | 2403.04944 | 77 | 1210.0269 | 126 |
| 1303.1856 | 44 | 1708.04269 | 80 | 1302.5984 | 126, 140, 146 |
| 1310.5610 | 44 | 0712.1332 | 82 | 1303.6228 | 126 |
| 1406.1168 | 44 | 0911.5665 | 82, 87 | 1510.02575 | 126 |
| 1501.05457 | 44, 53 | 1203.1255 | 82, 96 | 1808.03213 | 126 |
| 1511.08568 | 44 | 1302.0548 | 82, 96, 126 | 1901.07962 | 126 |
| 1602.00336 | 44 | 1610.04839 | 82, 107 | 1909.10294 | 126 |
| 1704.02498 | 44 | 1611.02217 | 82, 96, 126 | 1910.07551 | 126 |
| 1801.09181 | 44 | 1911.05456 | 83, 84, 90, 91, 97, 98, 100, 103, 109, 111, 127, 128, 136, 138, 141, 142 | 1912.00765 | 126 |
| 1802.01473 | 44 | 2110.03651 | 85, 89, 92, 93, 94, 95, 96, 98, 101, 102, 104, 105, 110, 112, 122, 123, 124, 126, 127, 134, 135, 139, 143 | 2003.02572 | 126 |
| 1802.01506 | 44, 126 | 1101.0600 | 87 | 2101.09753 | 126 |
| 1809.00998 | 44 | 2210.07238 | 87, 96, 137 | 2109.09877 | 126 |
| 1812.06643 | 44 | math/0503507 | 87 | 2203.16047 | 126, 127, 128 |
| 1906.09629 | 44, 57, 106, 113, 121 | 2212.09965 | 88, 99, 119, 125, 133 | 2210.01331 | 126 |
| 1907.04089 | 44 | 0805.2788 | 96, 126, 137 | 2301.12932 | 126 |
| 1911.12551 | 44 | 1103.6022 | 96 | 2303.05402 | 126 |
| 1912.03214 | 44, 51, 147 | 1104.0392 | 96, 126, 137 | 2310.15207 | 126 |
| 1912.03527 | 44 | 1104.1994 | 96 | 1808.04717 | 128 |
| 2009.10774 | 44 | 1104.3856 | 96, 126, 140 | 1501.06413 | 129 |
| 2104.12412 | 44, 96 | 1410.5514 | 96, 113 | 2305.00626 | 131, 132, 145 |
| 2105.05809 | 44 | 1504.01976 | 96 | 0704.2438 | 137 |
| 2106.04517 | 44 | 1512.04608 | 96, 108, 126 | 1004.4623 | 137 |
| 2110.07457 | 44 | 1604.00193 | 96 | 1802.04616 | 137 |
| 2203.09465 | 44, 53, 54, 56, 149 | 1604.01106 | 96, 126 | 0909.2387 | 149 |
| 2206.07174 | 44, 51, 58 | 1609.07276 | 96 | math/0502582 | 149 |
| 2212.13687 | 44 | 1804.02695 | 96, 140 | | |

Table 16: **Formulas unified by the $\pi$ Conservative Matrix Field (CMF)** as shown in Fig. 6. Clusters of formulas harvested from the literature are given in terms of their corresponding trajectory in the CMF. Dashes indicate a formula's canonical form (CF) is the same as the formula collected. Each canonical form has a numbered row and some are followed by additional source formulas.

| Cluster | | Formula | Value | Canonical form (CF) | CF value | Convergence rate |
|---|---|---|---|---|---|---|
| (1, 1, 2) $\delta = -0.21$ | | | | | | |
| | 1 | $\mathrm{PCF}(2n+1, n^2)$ | $\frac{4}{\pi}$ | - | - | 1.76 |
| | 2 | $\mathrm{PCF}(2n+3, n^2+2n)$ | $\frac{4}{-2+\pi}$ | $\mathrm{PCF}(2n+3, n^2+2n)$ | $\frac{4}{-2+\pi}$ | 1.76 |
| | 3 | $\mathrm{PCF}(2n+5, n^2+4n)$ | $\frac{8}{-8+3\pi}$ | $\mathrm{PCF}(2n+5, n^2+4n)$ | $\frac{8}{-8+3\pi}$ | 1.76 |
| | 4 | $\mathrm{PCF}(-48n^3 - 108n^2 - 70n - 12, -64n^6 - 96n^5 + 52n^3 + 15n^2)$ | $\frac{10}{-4+\pi}$ | $\mathrm{PCF}(-48n^3 - 108n^2 - 70n - 12, -64n^6 - 96n^5 + 12n^4 + 52n^3 + 15n^2)$ | $\frac{10}{-4+\pi}$ | 3.53 |
| (3, 1, 1) $\delta = -0.45$ | | | | | | |
| | 5 | $\sum_{k=1}^{\infty} \frac{(-4)^k (280k - 51)\binom{2k}{k}}{k\binom{3k}{k}\binom{6k}{3k}}$ | $-6\pi - 10$ | $\mathrm{PCF}(29120n^3 + 110616n^2 + 132106n + 49845, 33868800n^6 + 173940480n^5 + 322863792n^4 + 252606312n^3 + 62976828n^2 - 9173682n - 2336310)$ | $\frac{229050 + 137430\pi}{154 - 45\pi}$ | 3.29 |
| | 6 | $\sum_{k=1}^{\infty} \frac{(-4)^k (952k - 201)\binom{2k}{k}}{\binom{3k}{k}\binom{6k}{3k}}$ | $-15\pi - 42$ | $\mathrm{PCF}(99008n^3 + 369416n^2 + 427970n + 153045, 391523328n^6 + 2379573504n^5 + 5231138352n^4 + 4800421464n^3 + 1334845140n^2 - 258125598n - 61614540)$ | $\frac{12874680 + 4598100\pi}{872 - 225\pi}$ | 3.29 |
| | 7 | $\sum_{k=1}^{\infty} \frac{(-4)^k (7k - 1)\binom{2k}{k}}{k(2k-1)\binom{3k}{k}\binom{6k}{3k}}$ | $-\frac{\pi}{4}$ | $\mathrm{PCF}(728n^3 + 2822n^2 + 3469n + 1360, 21168n^6 + 89208n^5 + 130380n^4 + 78570n^3 + 15162n^2 - 1458n - 390)$ | $\frac{130\pi}{16 - 5\pi}$ | 3.30 |
| (2, 1, 1) $\delta = -0.48$ | | | | | | |
| | 8 | $\sum_{k=1}^{\infty} \frac{(-2)^k k(126k + 29)}{\binom{4k}{2k}}$ | $-\frac{65}{3} - 2\pi$ | $\mathrm{PCF}(1764n^4 + 8596n^3 + 14767n^2 + 10143n + 2053, 508032n^8 + 4552128n^7 + 16355000n^6 + 29955532n^5 + 29379250n^4 + 14855861n^3 + 3459708n^2 + 293364n)$ | $\frac{3372\pi + 36530}{15 - \pi}$ | 2.07 |
| | 9 | $\sum_{k=1}^{\infty} \frac{(-2)^{k-1} (6k - 1)}{k(2k-1)\binom{4k}{2k}}$ | $\frac{\pi}{4}$ | $\mathrm{PCF}(84n^3 + 328n^2 + 411n + 164, 1152n^6 + 4800n^5 + 6968n^4 + 4220n^3 + 838n^2 - 95n - 33)$ | $\frac{33\pi}{10 - 3\pi}$ | 2.08 |
| | 10 | $\sum_{k=0}^{\infty} \frac{(-2)^k (30k - 7)}{\binom{4k}{2k}}$ | $-\frac{32}{3} - \frac{\pi}{2}$ | $\mathrm{PCF}(420n^3 + 232n^2 - 121n - 44, 28800n^6 + 960n^5 - 57352n^4 - 8996n^3 + 27766n^2 + 4697n - 2553)$ | $\frac{-1472 - 69\pi}{3\pi + 22}$ | 2.08 |
| | 11 | $\sum_{k=1}^{\infty} \frac{(-2)^k (18k + 1)}{(2k-1)\binom{4k}{2k}}$ | $-\frac{3\pi}{2} - 1$ | $\mathrm{PCF}(252n^3 + 1004n^2 + 1321n + 591, 10368n^6 + 58176n^5 + 116024n^4 + 100084n^3 + 38458n^2 + 5843n + 222)$ | $\frac{444 + 666\pi}{32 - 9\pi}$ | 2.08 |
| | 12 | $\sum_{j=0}^{\infty} \frac{64^{-j} (3j+2)\left(112j^2 + 144j + 41\right)\left(\frac{1}{4}\right)_{(j)}\left(\frac{1}{2}\right)_{(j)}\left(\frac{3}{4}\right)_{(j)} j!}{\left(\frac{9}{8}\right)_{(j)}\left(\frac{11}{8}\right)_{(j)}\left(\frac{13}{8}\right)_{(j)}\left(\frac{15}{8}\right)_{(j)}}$ | $\frac{105\pi}{4}$ | $\mathrm{PCF}(1397760n^7 + 11104768n^6 + 36657792n^5 + 65007040n^4 + 66685140n^3 + 39437842n^2 + 12401823n + 1591920, -29595009024n^{14} - 248739004416n^{13} - 868936056832n^{12} - 1606677430272n^{11} - 1605350604800n^{10} - 651457732608n^9 + 282238639104n^8 + 446685049344n^7 + 175738683712n^6 - 1178920416n^5 - 20395524172n^4 - 5128349922n^3 - 106481538n^2 + 98454690n + 8419950)$ | $\frac{935550\pi}{-328 + 105\pi}$ | 4.16 |

| Cluster | | Formula | Value | Canonical form (CF) | CF value | Convergence rate |
|---|---|---|---|---|---|---|
| $(1, 0, 0)$ $\delta = -0.65$ | | | | | | |
| | 13 | $PCF(3n+1, -2n^2+n)$ | $\frac{2}{\pi}$ | - | - | 0.69 |
| | 14 | $PCF(3n+2, -2n^2-n+1)$ | $\frac{\pi+4}{2+\pi}$ | - | - | 0.69 |
| | 15 | $PCF(3n+4, -2n^2-3n+2)$ | $\frac{12+4\pi}{\pi+4}$ | - | - | 0.69 |
| | 16 | $\sum_{i=1}^{\infty} \frac{2^i}{i\binom{2i}{i}}$ | $\frac{\pi}{2}$ | $PCF(3n+4, -2n^2-3n-1)$ | $\frac{\pi}{-2+\pi}$ | 0.69 |
| | | $\sum_{k=1}^{\infty} \frac{2^k}{k\binom{2k}{k}}$ | $\frac{\pi}{2}$ | | | |
| | | $\sum_{k=0}^{\infty} \frac{k!}{(2k+1)!!}$ | $\frac{\pi}{2}$ | | | |
| | | $\sum_{j=0}^{\infty} \frac{2^{j+1}}{(2j+1)\binom{2j}{j}}$ | $\pi$ | | | |
| | 17 | $PCF(3n+5, -2n^2-5n+3)$ | $\frac{84+27\pi}{6\pi+20}$ | - | - | 0.69 |
| | 18 | $\sum_{i=1}^{\infty} \frac{2^i}{(2i+1)\binom{2i}{i}}$ | $-1 + \frac{\pi}{2}$ | $PCF(3n+7, -2n^2-7n-6)$ | $\frac{12-6\pi}{8-3\pi}$ | 0.69 |
| | 19 | $PCF(3n^2+9n+5, -2n^4-9n^3 - 9n^2+n+3)$ | $\frac{102\pi+357}{85+34\pi}$ | - | - | 0.69 |
| | 20 | $PCF(3n, -2n^2+3n)$ | $\frac{2}{2+\pi}$ | $PCF(3n, -2n^2+3n)$ | $\frac{2}{2+\pi}$ | 0.70 |
| | 21 | $PCF(3n+3, -2n^2+n)$ | $\frac{4}{-8+3\pi}$ | - | - | 0.70 |
| | 22 | $PCF(3n+3, -2n^2-n)$ | $-\frac{2}{-4+\pi}$ | - | - | 0.70 |
| | 23 | $PCF(3n+4, -2n^2+n)$ | $\frac{12}{-44+15\pi}$ | - | - | 0.70 |
| | 24 | $PCF(3n+4, -2n^2-n)$ | $-\frac{2}{-10+3\pi}$ | - | - | 0.70 |
| | 25 | $PCF(3n+5, -2n^2+n)$ | $\frac{48}{-320+105\pi}$ | - | - | 0.70 |
| | 26 | $PCF(3n+5, -2n^2-n)$ | $-\frac{4}{-48+15\pi}$ | - | - | 0.70 |
| | 27 | $PCF(3n+5, -2n^2-3n)$ | $\frac{6}{-8+3\pi}$ | - | - | 0.70 |
| | 28 | $PCF(3n+3, -2n^2-n+1)$ | $1 + \frac{\pi}{2}$ | - | - | 0.70 |
| | 29 | $PCF(3n+4, -2n^2-n+1)$ | $-\frac{\pi}{-4+\pi}$ | - | - | 0.70 |
| | 30 | $PCF(3n+5, -2n^2-n+1)$ | $\frac{4-3\pi}{-20+6\pi}$ | - | - | 0.70 |
| | 31 | $PCF(3n+5, -2n^2-3n+2)$ | $\frac{2\pi+8}{\pi}$ | - | - | 0.70 |
| | 32 | $PCF(3n+6, -2n^2-3n+2)$ | $\frac{8}{-8+3\pi}$ | - | - | 0.70 |
| | 33 | $PCF(3n+6, -2n^2-5n+3)$ | $\frac{15\pi+48}{8+3\pi}$ | - | - | 0.70 |
| | 34 | $PCF(3n+7, -2n^2-3n+2)$ | $\frac{32-6\pi}{-64+21\pi}$ | - | - | 0.70 |
| | 35 | $PCF(3n+7, -2n^2-5n+3)$ | $3 + \frac{9\pi}{8}$ | - | - | 0.70 |
| | 36 | $PCF(3n+8, -2n^2-5n+3)$ | $-\frac{9\pi}{-32+9\pi}$ | - | - | 0.70 |
| | 37 | $PCF(3n+9, -2n^2-5n+3)$ | $\frac{32-15\pi}{-96+30\pi}$ | - | - | 0.70 |
| | 38 | $\sum_{k=0}^{\infty} \frac{4^k k}{\binom{4k}{2k}}$ | $\frac{2}{3} + \frac{\pi}{4}$ | $PCF(20n^3+86n^2+123n+59, -64n^6-416n^5 - 1004n^4-1090n^3 - 504n^2-72n)$ | $\frac{64}{\pi}+24$ | 1.38 |
| | 39 | $\sum_{k=1}^{\infty} \frac{4^k(12k^2+1)}{\binom{4k}{2k}}$ | $\frac{50}{3} + \frac{11\pi}{2}$ | $PCF(240n^4+1320n^3+2792n^2+2726n+1043, -9216n^8-78336n^7-265920n^6-456096n^5-413728n^4-195792n^3-53804n^2-13194n-1764)$ | $\frac{19600+6468\pi}{16+11\pi}$ | 1.38 |
| | 40 | $\sum_{k=1}^{\infty} \frac{4^k(3k-1)}{k(2k-1)\binom{4k}{2k}}$ | $\frac{\pi}{2}$ | $PCF(60n^3+214n^2+237n+80, -576n^6-2208n^5-2860n^4-1330n^3+46n^2+178n+30)$ | $\frac{30\pi}{-8+3\pi}$ | 1.39 |
| | 41 | $\sum_{k=1}^{\infty} \frac{4^k(12k-5)}{(2k-1)\binom{4k}{2k}}$ | $2 + \frac{3\pi}{2}$ | $PCF(240n^3+884n^2+994n+321, -9216n^6-43008n^5-65536n^4-31904n^3+4900n^2+6914n+1140)$ | $\frac{912+684\pi}{-16+9\pi}$ | 1.39 |
| $(-1, 3, 3)$ $\delta = -0.91$ | | | | | | |
| | 42 | $\sum_{k=1}^{\infty} \frac{16^k(22k^2-17k+3)\binom{4k}{2k}}{k(4k-3)(4k-1)\binom{3k}{k}\binom{6k}{3k}}$ | $2\pi$ | $PCF(3784n^4+15292n^3+21230n^2+11981n+2304, -3345408n^8-13229568n^7-16699200n^6-4912608n^5+4514544n^4+2792280n^3+27384n^2-173304n-20520)$ | $\frac{6840\pi}{-32+15\pi}$ | 0.52 |

| Cluster | Formula | Value | Canonical form (CF) | CF value | Convergence rate |
|---|---|---|---|---|---|
| (1, 1, 1) $\delta = -1.00$ | | | | | |
| 43 | PCF$(4,4n^2 - 1)$ | $\frac{2+\pi}{-2+\pi}$ | - | - | 0.00 |
| 44 | $\sum_{k=0}^{\infty} \frac{(-1)^k}{2k+1}$ | $\frac{\pi}{4}$ | PCF$(2,4n^2 + 4n + 1)$ | $\frac{\pi}{4-\pi}$ | 0.00 |
| | $\sum_{l=0}^{\infty} \frac{(-1)^l}{2l+1}$ | $\frac{\pi}{4}$ | | | |
| | $\sum_{m=0}^{\infty} \frac{(-1)^m}{2m+1}$ | $\frac{\pi}{4}$ | | | |
| | $\sum_{n=0}^{\infty} \frac{(-1)^n}{2n+1}$ | $\frac{\pi}{4}$ | | | |
| | $\sum_{n=1}^{\infty} \frac{(-1)^h}{2n-1}$ | $-\frac{\pi}{4}$ | | | |
| | $\sum_{j=1}^{\infty} \frac{(-1)^{j-1}}{2j-1}$ | $\frac{\pi}{4}$ | | | |
| | $\sum_{k=1}^{\infty} \frac{(-1)^{k+1}}{2k-1}$ | $\frac{\pi}{4}$ | | | |
| | $\sum_{n=1}^{\infty} \frac{(-1)^{n+1}}{2n-1}$ | $\frac{\pi}{4}$ | | | |
| | $\sum_{n=1}^{\infty} \frac{(-1)^{n-1}}{2n-1}$ | $\frac{\pi}{4}$ | | | |
| | $\sum_{\nu=0}^{\infty} \frac{(-1)^{\nu}}{2\nu+1}$ | $\frac{\pi}{4}$ | | | |
| | $\sum_{p=0}^{\infty} \frac{(-1)^p}{p+\frac{1}{2}}$ | $\frac{\pi}{2}$ | | | |
| 45 | $\sum_{k=1}^{\infty} \frac{(-1)^k}{(2k-1)(2k+1)}$ | $\frac{1}{2} - \frac{\pi}{4}$ | PCF$(4,4n^2 + 8n + 3)$ | $\frac{-6+3\pi}{10-3\pi}$ | 0.00 |
| | $\sum_{n=1}^{\infty} \frac{(-1)^{n-1}}{(2n-1)(2n+1)}$ | $-\frac{1}{2} + \frac{\pi}{4}$ | | | |
| 46 | PCF$(4,4n^2 - 8n + 3)$ | $\frac{3\pi+10}{2+\pi}$ | - | - | 0.00 |
| 47 | PCF$(6,4n^2 + 4n - 3)$ | $\frac{3\pi}{-8+3\pi}$ | - | - | 0.00 |
| 48 | PCF$(6,4n^2 - 4n + 1)$ | $3 + \pi$ | - | - | 0.00 |
| | $\sum_{n=1}^{\infty} \frac{(-1)^{n+1}}{n(n+1)(2n+1)}$ | $\pi - 3$ | | | |
| 49 | $\sum_{n=1}^{\infty} \frac{(-1)^{n+1}}{2n+1}$ | $1 - \frac{\pi}{4}$ | PCF$(2,4n^2 + 12n + 9)$ | $\frac{-36+9\pi}{8-3\pi}$ | 0.00 |
| | $\sum_{\ell=1}^{\infty} \frac{(-1)^{\ell}}{2\ell+1}$ | $-1 + \frac{\pi}{4}$ | | | |
| 50 | $\sum_{n=1}^{\infty} \frac{(-1)^{n-1}}{(2n-1)(2n+1)(2n+3)}$ | $-\frac{1}{3} + \frac{\pi}{8}$ | PCF$(6,4n^2 + 12n + 5)$ | $\frac{-40+15\pi}{48-15\pi}$ | 0.00 |
| 51 | $\sum_{k=2}^{\infty} \frac{(-1)^{k+1}}{k(k-1)(2k-1)}$ | $3 - \pi$ | PCF$(6,4n^2 + 12n + 9)$ | $\frac{-27+9\pi}{19-6\pi}$ | 0.00 |
| | $\sum_{k=1}^{\infty} \frac{(-1)^{k+1}}{2k(2k+1)(2k+2)}$ | $-\frac{3}{4} + \frac{\pi}{4}$ | | | |
| | $\sum_{n=1}^{\infty} \frac{(-1)^{n-1}}{2n(2n+1)(2n+2)}$ | $-\frac{3}{4} + \frac{\pi}{4}$ | | | |
| | $\sum_{n=1}^{\infty} (-1)^{n+1} \left( -\frac{4}{2n+1} + \frac{1}{n+1} + \frac{1}{n} \right)$ | $-3 + \pi$ | | | |
| | $\sum_{n=1}^{\infty} \frac{2(-1)^n \left( \frac{1}{n+1} + \frac{1}{n} \right)}{(2n+1)^2}$ | $6 - 2\pi$ | | | |
| 52 | $\sum_{n=1}^{\infty} \frac{(-1)^{n-1}}{(2n-1)(2n+1)(2n+3)(2n+5)}$ | $-\frac{11}{90} + \frac{\pi}{24}$ | PCF$(8,4n^2 + 16n + 7)$ | $\frac{-308+105\pi}{332-105\pi}$ | 0.00 |
| 53 | $\sum_{n=1}^{\infty} \frac{1}{(4n-3)(4n-1)}$ | $\frac{\pi}{8}$ | PCF$(32n^2 + 64n + 38, -256n^4 - 512n^3 - 352n^2 - 96n - 9)$ | $\frac{9\pi}{-8+3\pi}$ | 0.00 |
| | $\sum_{k=1}^{\infty} \left( -\frac{1}{4k-1} + \frac{1}{4k-3} \right)$ | $\frac{\pi}{4}$ | | | |
| | $\sum_{n=1}^{\infty} \left( -\frac{1}{4n-1} + \frac{1}{4n-3} \right)$ | $\frac{\pi}{4}$ | | | |
| 54 | $\sum_{n=1}^{\infty} \frac{1}{(4n-3)(4n-1)(4n+1)}$ | $-\frac{1}{8} + \frac{\pi}{16}$ | PCF$(32n^2 + 80n + 66, -256n^4 - 768n^3 - 800n^2 - 336n - 45)$ | $\frac{90-45\pi}{46-15\pi}$ | 0.00 |
| 55 | $\sum_{n=1}^{\infty} \frac{1}{16n^2-1}$ | $\frac{1}{2} - \frac{\pi}{8}$ | PCF$(32n^2 + 96n + 78, -256n^4 - 1024n^3 - 1504n^2 - 960n - 225)$ | $\frac{900-225\pi}{52-15\pi}$ | 0.00 |
| 56 | $\sum_{n=1}^{\infty} \frac{1}{(2n-1)(2n+1)(4n-1)(4n+1)}$ | $-\frac{1}{2} + \frac{\pi}{6}$ | PCF$(32n^2 + 96n + 110, -256n^4 - 1024n^3 - 1504n^2 - 960n - 225)$ | $\frac{225-75\pi}{47-15\pi}$ | 0.00 |
| 57 | $\sum_{k=1}^{\infty} \left( -\frac{1}{4k+3} + \frac{1}{4k+1} \right)$ | $-\frac{2}{3} + \frac{\pi}{4}$ | PCF$(32n^2 + 128n + 134, -256n^4 - 1536n^3 - 3424n^2 - 3360n - 1225)$ | $\frac{9800-3675\pi}{304-105\pi}$ | 0.00 |
| 58 | $\sum_{n=1}^{\infty} \frac{3}{n(n+1)(4n+1)(4n+3)}$ | $\frac{19}{3} - 2\pi$ | PCF$(32n^2 + 128n + 166, -256n^4 - 1536n^3 - 3424n^2 - 3360n - 1225)$ | $\frac{23275-7350\pi}{1321-420\pi}$ | 0.00 |
| 59 | PCF$(8,4n^2 - 1)$ | $\frac{\pi+4}{4-\pi}$ | PCF$(8,4n^2 - 1)$ | $\frac{\pi+4}{4-\pi}$ | 0.01 |
| 60 | PCF$(8,4n^2 + 8n - 5)$ | $\frac{20-15\pi}{44-15\pi}$ | - | - | 0.01 |
| 61 | PCF$(10,4n^2 + 4n - 3)$ | $\frac{6}{10-3\pi}$ | - | - | 0.01 |
| 62 | PCF$(10,4n^2 - 4n + 1)$ | $5 + \frac{16}{\pi}$ | - | - | 0.01 |
| 63 | $\sum_{n=1}^{\infty} \frac{(-1)^{n-1}}{n(2n+1)(2n+2)(2n+3)(4n-2)}$ | $\frac{5}{36} - \frac{\pi}{24}$ | PCF$(10,4n^2 + 12n + 5)$ | $\frac{-50+15\pi}{94-30\pi}$ | 0.01 |
| 64 | PCF$(10,4n^2 + 12n - 7)$ | $\frac{224-105\pi}{320-105\pi}$ | - | - | 0.01 |
| 65 | $\sum_{n=1}^{\infty} \frac{(-1)^{n-1}}{(2n-1)(2n+1)(2n+3)(2n+5)(2n+7)}$ | $-\frac{2}{63} + \frac{\pi}{96}$ | PCF$(10,4n^2 + 20n + 9)$ | $\frac{-960+315\pi}{992-315\pi}$ | 0.01 |
| 66 | $\sum_{n=2}^{\infty} \frac{(-1)^n}{(2n+1)(2n+2)(2n+3)(2n+4)(2n+5)}$ | $\frac{11}{84} - \frac{\pi}{24}$ | PCF$(10,4n^2 + 28n + 45)$ | $\frac{-14850+4725\pi}{3958-1260\pi}$ | 0.01 |
| 67 | $\sum_{n=1}^{\infty} \frac{(-1)^{n-1}}{(2n-1)^2(2n+1)^2(2n+3)^2(2n+5)^2}$ | $-\frac{7}{4050} + \frac{\pi}{1728}$ | PCF$(32n + 80,16n^4 + 128n^3 + 312n^2 + 224n + 49)$ | $\frac{-10976+3675\pi}{11552-3675\pi}$ | 0.01 |
| 68 | $\sum_{n=1}^{\infty} \frac{(-1)^{n-1}}{(2n-1)^2(2n+1)^2(2n+3)^2(2n+5)^2(2n+7)^2(2n+9)^2}$ $\frac{41}{44651250} - \frac{\pi}{3456000}$ | | PCF$(48n + 168,16n^4 + 192n^3 + 664n^2 + 528n + 121)$ | $\frac{-2540032+800415\pi}{2514432-800415\pi}$ | 0.01 |
| 69 | PCF$(24,4n^2 - 1)$ | $\frac{75\pi+256}{256-75\pi}$ | - | - | 0.02 |
| 70 | $\sum_{n=1}^{\infty} \frac{(-1)^{n-1}}{(2n-1)^2(2n+1)^2(2n+3)^2(2n+5)^2(2n+7)^2(2n+9)^2(2n+11)^2(2n+13)^2}$ $-\frac{14789}{1342217914153750} + \frac{\pi}{28449792000}$ | | PCF$(64n + 288,16n^4 + 256n^3 + 1144n^2 + 960n + 225)$ | $\frac{-181727232+57972915\pi}{182128640-57972915\pi}$ | 0.02 |
| | * | | | | |

| Cluster | | Formula | Value | Canonical form (CF) | CF value | Convergence rate |
|---|---|---|---|---|---|---|
| (0, 0, 1) $\delta = -1.00$ | | | | | | |
| | 71 | PCF(2,$n^2$) | $\frac{2}{4-\pi}$ | - | - | 0.00 |
| | 72 | PCF(1,$n(n+1)$) | $\frac{2}{\pi-2}$ | - | - | 0.00 |
| | 73 | $\sum_{n=0}^{\infty} \frac{4^{-2n}\binom{2n}{n}^2}{n+1}$ | $\frac{4}{\pi}$ | PCF($8n^2+16n+9,-16n^4-32n^3-20n^2-4n$) | $\frac{4}{4-\pi}$ | 0.00 |
| | | $\sum_{n=0}^{\infty} \frac{2^{-4n-3}\binom{2n}{n}\binom{2n+2}{n+1}}{2n+1}$ | $\frac{1}{\pi}$ | | | |
| | | $\sum_{n=0}^{\infty} \frac{\left(\left(\frac{1}{2}\right)_{(n)}\right)^2}{n!(n+1)!}$ | $\frac{4}{\pi}$ | | | |
| | | $\sum_{k=0}^{\infty} \frac{\left(\left(\frac{1}{2}\right)_{(k)}\right)^2}{(k+1)k!^2}$ | $\frac{4}{\pi}$ | | | |
| | | $\sum_{n=0}^{\infty} \frac{4\left(\left(\frac{1}{2}\right)_{(n)}\right)^2}{2^{(n)}n!}$ | $\frac{16}{\pi}$ | | | |
| | | $\sum_{n=0}^{\infty} \frac{\left(\left(\frac{1}{2}\right)_{(n)}\right)^2}{(n+1)n!^2}$ | $\frac{4}{\pi}$ | | | |
| | 74 | $\sum_{n=0}^{\infty} \frac{\left(\left(\frac{1}{2}\right)_{(n)}\right)^2}{n!(n+2)!}$ | $\frac{16}{9\pi}$ | PCF($8n^2+20n+13,-16n^4-48n^3-36n^2-8n$) | $\frac{32}{32-9\pi}$ | 0.00 |
| | | $\sum_{n=0}^{\infty} \frac{\left(\left(\frac{1}{2}\right)_{(n)}\right)^2}{(n+1)(n+2)n!^2}$ | $\frac{16}{9\pi}$ | | | |
| | 75 | $\sum_{n=0}^{\infty} \frac{4^{-2n}\binom{2n}{n}^2}{(n+1)^2}$ | $-4+\frac{16}{\pi}$ | PCF($8n^2+20n+17,-16n^4-48n^3-52n^2-24n-4$) | $\frac{-16+4\pi}{-16+5\pi}$ | 0.00 |
| | | $\sum_{n=0}^{\infty} \frac{2^{-4n-4}\binom{2n}{n}\binom{2n+2}{n+1}}{(n+1)(2n+1)}$ | $\frac{4-\pi}{2\pi}$ | | | |
| | 76 | $\sum_{n=1}^{\infty} \frac{2^{4n}}{n^2(2n+1)\binom{2n}{n}^2}$ | $-4+2\pi$ | PCF($8n^2+24n+19,-16n^4-64n^3-92n^2-56n-12$) | $\frac{24-12\pi}{8-3\pi}$ | 0.00 |
| | 77 | $\sum_{i=1}^{\infty} \frac{(2i-1)!!^2}{(i+1)(2i-1)(2i)!!^2}$ | $\frac{-\frac{8}{3}+\pi}{\pi}$ | PCF($8n^2+28n+27,-16n^4-80n^3-140n^2-100n-24$) | $\frac{192-72\pi}{64-21\pi}$ | 0.00 |
| | 78 | $\sum_{n=1}^{\infty} \frac{\left(\left(\frac{1}{2}\right)_{(n)}\right)^2}{(n+1)!^2}$ | $-5+\frac{16}{\pi}$ | PCF($8n^2+36n+45,-16n^4-112n^3-292n^2-336n-144$) | $\frac{-2304+720\pi}{-256+81\pi}$ | 0.00 |
| | 79 | $\sum_{n=1}^{\infty} \frac{\left(\left(\frac{1}{2}\right)_{(n)}\right)^2}{(n+1)!(n+2)!}$ | $\frac{256-81\pi}{18\pi}$ | PCF($8n^2+40n+57,-16n^4-128n^3-372n^2-468n-216$) | $\frac{-18432+5832\pi}{-2048+651\pi}$ | 0.00 |
| | 80 | $\sum_{n=1}^{\infty} \frac{\left(\left(\frac{1}{2}\right)_{(n)}\right)^2}{(n+2)(n+1)!^2}$ | $\frac{256-81\pi}{18\pi}$ | | | |
| | | $\sum_{n=1}^{\infty} \frac{16^n}{(2n+1)^2(2n+3)^2\binom{2n}{n}^2}$ | $-\frac{28}{9}+\pi$ | PCF($8n^2+44n+65,-16n^4-144n^3-484n^2-720n-400$) | $\frac{11200-3600\pi}{704-225\pi}$ | 0.00 |
| | 81 | $\sum_{n=1}^{\infty} \frac{\left(\left(\frac{1}{2}\right)_{(n)}\right)^2}{(n+2)!^2}$ | $\frac{2048-651\pi}{108\pi}$ | PCF($8n^2+44n+73,-16n^4-144n^3-468n^2-648n-324$) | $\frac{-73728+23436\pi}{-8192+2607\pi}$ | 0.00 |

Table 17: **Additional formulas for $\pi$ that were automatically harvested from arXiv, clustered, and proven equivalent within the clusters, but not unified**. We expect these formulas to be unified in the future, by enlarging the $\pi$-CMF (Eq. (25)) to new dimensions, i.e. adding matrices and directions to walk in, and by improving the matching algorithm. This table complements Table 16; the formulas are mutually exclusive and were collected using the same harvesting pipeline. These formulas were then connected among themselves as described in Appendix C.1, resulting in clusters.

| Cluster | Formula | Value | Canonical form (CF) | CF value | Convergence rate |
|---------|---------|-------|---------------------|----------|------------------|
| $\delta = -0.29$ | | | | | |
| 82 | $\sum_{k=0}^{\infty} \frac{(-1)^k 882^{-2k}(21460k+1123)\left(\frac{1}{4}\right)_{(k)}\left(\frac{1}{2}\right)_{(k)}\left(\frac{3}{4}\right)_{(k)}}{k!^3}$ | $\frac{3528}{\pi}$ | PCF($534215282560n^4 + 1630601631968n^3 + 1686512782328n^2 + 618081838666n + 27955409115,366856790423961600n^8 + 588680355780034560n^7 - 56045383774765056n^6 - 487988770034755584n^5 - 247923828204062976n^4 - 34298642100691584n^3$) | $\frac{239018472}{-3528+1123\pi}$ | 13.56 |
| | $\sum_{n=0}^{\infty} \frac{(-1)^n 882^{-2n}(21460n+1123)\left(\frac{1}{4}\right)_{(n)}\left(\frac{1}{2}\right)_{(n)}\left(\frac{3}{4}\right)_{(n)}}{n!^3}$ | $\frac{3528}{\pi}$ | | | |
| | $\sum_{k=0}^{\infty} \frac{(-1)^k 882^{-2k-1}(21460k+1123)\left(\frac{1}{4}\right)_{(k)}\left(\frac{1}{2}\right)_{(k)}\left(\frac{3}{4}\right)_{(k)}}{k!^3}$ | $\frac{4}{\pi}$ | | | |
| | $\sum_{n=0}^{\infty} \frac{\left(-\frac{1}{777924}\right)^n(21460n+1123)\left(\frac{1}{4}\right)_{(n)}\left(\frac{1}{2}\right)_{(n)}\left(\frac{3}{4}\right)_{(n)}}{n!^3}$ | $\frac{3528}{\pi}$ | | | |
| | $\sum_{n=0}^{\infty} \frac{(-1)^n 882^{-2n-1}(21460n+1123)\left(\frac{1}{4}\right)_{(n)}\left(\frac{1}{2}\right)_{(n)}\left(\frac{3}{4}\right)_{(n)}}{n!^3}$ | $\frac{4}{\pi}$ | | | |
| 83 | $\sum_{k=0}^{\infty} \frac{(-199148544)^{-k}\left(413512k^2-50826k-3877\right)\binom{2k}{k}^2\binom{4k}{2k}}{(2k-1)(4k-1)}$ | $-\frac{12180}{\pi}$ | PCF($10293775858432n^5 + 29616108563136n^4 + 26989135212112n^3 + 6208537419732n^2 - 1554777635156n - 96512721945,136211370064688185344n^{10} - 1015899809989452189696n^9 - 264690152463365701632n^8 + 176607156266987249664n^7 + 130723278949247781888n^6 - 76446780832924992000n^5 - 7169513755287663360n^4 + 4112854339342396032n^3$) | $\frac{4370293620}{-12180+3877\pi}$ | 13.56 |
| 84 | $\sum_{k=0}^{\infty} \frac{(-199148544)^{-k}\left(1424799848k^2+1533506502k+108685699\right)\binom{2k}{k}^2\binom{4k}{2k}}{(k+1)(2k-1)(4k-1)}$ | $\frac{341446000}{\pi}$ | PCF($35468306308982528n^5 + 180047738533689024n^4 + 332745102731042192n^3 + 272631301503072468n^2 + 89876772716256332n + 5411146610376015,161712967678730132721264230 4n^{10} + 428958552689457343506048614 4n^9 - 283366210981584591028224000n^8 - 578121362136863737845475737 6n^7 - 103927897759426752285201715 2n^6 + 195228558726217305788352128 00n^5 + 656926263945042965550190 08n^4 - 100482263421913916885155968n^3 - 1599880200791331634560n^2$) | $\frac{1047212167162854000}{-341446000+108685699\pi}$ | 13.56 |
| 85 | $\sum_{k=0}^{\infty} (-199148544)^{-k}\, k^3\left(643835623600k^2 - 1361740501968k + 711617288021\right)\frac{\binom{2k}{k}^2\binom{4k}{2k}}{}$ | $\frac{11907}{5\pi}$ | PCF($16027345274169049600n^5 + 46238118052856925952n^4 + 42393702411206957504n^3 + 10025851657990883552n^2 - 2313657716521897662n - 156579723130929021,330207651328 8022382676285849600 00n^{10} + 140995778801021981269171839959040 0n^9 + 120387428592332982632029268272 7424n^8 - 232910652624788403294094340 9971200n^7 - 401507715278567256196515497 4486528n^6 - 2233673407143085897463553 69486336n^5 + 23878496722883367446506 32714720000n^4 + 10480946458200502716153 04567478400n^3$) | $\frac{1948560328369940813539200}{-32934190464+10479317245\pi}$ | 13.56 |
| $\delta = -0.29$ | | | | | |
| 86 | $\sum_{k=1}^{\infty} \frac{2^{-k}(5k-2)}{k(2k-1)\binom{3k}{k}}$ | $\frac{\pi}{6}$ | PCF($145n^3 + 517n^2 + 572n + 188, -1350n^6 - 4995n^5 - 6186n^4 - 2565n^3 + 420n^2 + 540n + 96$) | $\frac{8\pi}{-3+\pi}$ | 2.60 |
| 87 | $\sum_{k=0}^{\infty} \frac{2^{-k}(25k-3)}{\binom{3k}{k}}$ | $\frac{\pi}{2}$ | PCF($725n^3 + 713n^2 + 160n + 4, -33750n^6 - 8775n^5 + 63564n^4 + 15957n^3 - 30972n^2 - 4644n + 3696$) | $\frac{22\pi}{\pi+6}$ | 2.60 |
| | $\sum_{n=0}^{\infty} \frac{2^{-n}(50n-6)}{\binom{3n}{n}}$ | $\pi$ | | | |

| Cluster | Formula | Value | Canonical form (CF) | CF value | Convergence rate |
|---|---|---|---|---|---|
| $\delta = -0.34$ | | | | | |
| 88 | $\sum_{n=1}^{\infty} \frac{(2n+1)(3n+1)(14n+11)\binom{2n}{n}}{(2n-1)(4n+1)^2(4n+3)\binom{4n}{2n}^2}$ | $\frac{5}{3} - \frac{\pi}{2}$ | $\text{PCF}(21840n^6 + 264512n^5 + 1320976n^4 + 3480988n^3 + 5103299n^2 + 3944955n + 1255590, -7225344n^{12} - 128163840n^{11} - 1015908352n^{10} - 4751034368n^9 - 14575174464n^8 - 30842090208n^7 - 46062845088n^6 - 48807290832n^5 - 36316590692n^4 - 18450538102n^3 - 6054381960n^2 - 1147592250n - 94594500)$ | $\frac{191100-57330\pi}{22-7\pi}$ | 4.16 |
| $\delta = -0.49$ | | | | | |
| 89 | $\sum_{k=0}^{\infty} (-82944)^{-k} k^3 \left(2428400k^2 - 5044368k + 2584321\right) \binom{2k}{k}^2 \binom{4k}{2k}$ | $\frac{243}{5\pi}$ | $\text{PCF}(25099942400n^5 + 73089143552n^4 + 68107981504n^3 + 16786867552n^2 - 3892088262n - 560080521, 1956525061570560000n^{10} + 8502121389529497600n^9 + 7771802427851341824n^8 - 13080888260893900800n^7 - 24151221026510819328n^6 - 2397288296314791936n^5 + 13672048552352582400n^4 + 6215310504603446400n^3)$ | $\frac{64935193276800}{-279936+52745\pi}$ | 5.77 |
| 90 | $\sum_{k=0}^{\infty} \frac{(-82944)^{-k}\left(1144k^2-102k-19\right)\binom{2k}{k}^2\binom{4k}{2k}}{(2k-1)(4k-1)}$ | $-\frac{60}{\pi}$ | $\text{PCF}(11824384n^5 + 34473792n^4 + 32217904n^3 + 8117004n^2 - 1648652n - 198015, 434207195136n^{10} - 294532153344n^9 - 867809230848n^8 + 536815558656n^7 + 429636897792n^6 - 240626557440n^5 - 23568433920n^4 + 13014131328n^3)$ | $\frac{61380}{-60+19\pi}$ | 5.78 |
| 91 | $\sum_{k=0}^{\infty} \frac{(-82944)^{-k}\left(3224k^2+4026k+637\right)\binom{2k}{k}^2\binom{4k}{2k}}{(k+1)(2k-1)(4k-1)}$ | $\frac{2000}{\pi}$ | $\text{PCF}(33323264n^5 + 175163712n^4 + 340624496n^3 + 302122284n^2 + 116510516n + 13200945, 3448538136576n^{10} + 10337057243136n^9 + 2209893580800n^8 - 11941876998144n^7 - 3960532620288n^6 + 3945642969600n^5 + 453938141952n^4 - 202478424192n^3 - 13492448640n^2)$ | $\frac{15774000}{-2000+637\pi}$ | 5.78 |
| 92 | $\sum_{k=0}^{\infty} (-82944)^{-k} \left(2475740800k^2 + 4950772932k + 2475031103\right) \binom{2k}{k+1}^2 \binom{4k}{2k}$ | $-\frac{2238840}{\pi}$ | $\text{PCF}(25589256908800n^6 + 307024146260352n^5 + 1483653045985888n^4 + 3682571629531512n^3 + 4933895619830194n^2 + 3372826099775973n + 919254248208450, 2033552151394532720640000n^{12} + 3761955061322637241211200n^{11} + 3084452313459100903780224n^{10} + 1476028855603509052602580992n^9 + 4565376629008757961859399680n^8 + 9536441599068207233102567424n^7 + 13623723826190777310302174976n^6 + 13142549025479964764182704768n^5 + 8192764123025109436770174720n^4 + 2979947458777007024574045696n^3 + 4802431275030561124142131120n^2)$ | $\frac{965363871612406468680}{-2063314944+660102989\pi}$ | 5.78 |

| Cluster | Formula | Value | Canonical form (CF) | CF value | Convergence rate |
|---|---|---|---|---|---|
| $\delta = -0.50$ | | | | | |
| 93 | $\sum_{k=0}^{\infty} 4096^{-k} k^3 \left(198k^2 - 425k + 210\right) \binom{2k}{k}^3$ | $-\frac{1}{21\pi}$ | $\mathbf{PCF}(102960n^5 + 299344n^4 + 276000n^3 + 61356n^2 - 22843n - 4600, -160579584n^{10} - 675569664n^9 - 526950400n^8 + 1278275584n^7 + 2128862208n^6 + 232229376n^5 - 1054909440n^4 - 441262080n^3)$ | $\frac{2101248}{512-357\pi}$ | 4.15 |
| | $\sum_{k=0}^{\infty} 4096^{-k} k^2 \left(5544k^3 - 11900k^2 + 5880k\right) \binom{2k}{k}^3$ | $-\frac{4}{3\pi}$ | | | |
| 94 | $\sum_{k=0}^{\infty} 4096^{-k} k^2 \left(504k^2 - 314k - 11\right) \binom{2k}{k}^3$ | $\frac{4}{3\pi}$ | $\mathbf{PCF}(262080n^5 + 1419280n^4 + 2902704n^3 + 2814416n^2 + 1289208n + 220475, -1040449536n^{10} - 8587837440n^9 - 28095496192n^8 - 44392325120n^7 - 29396893696n^6 + 5069131776n^5 + 17276599296n^4 + 6864182784n^3 + 209392128n^2)$ | $\frac{76142592}{2048-537\pi}$ | 4.15 |
| 95 | $\sum_{k=0}^{\infty} 4096^{-k} \left(k^2 + 1\right) \left(126504k^2 - 921334k - 109205\right) \binom{2k}{k}^3$ | $-\frac{1063412}{3\pi}$ | $\mathbf{PCF}(65782080n^7 - 279217360n^6 - 1232128016n^5 - 1846926048n^4 - 1980776344n^3 - 1616603667n^2 - 653455030n - 57721030, -65549361217536n^{14} + 856473516638208n^{13} - 1849611118133248n^{12} - 5920093389561856n^{11} - 1760186613374976n^{10} + 6901075810324480n^9 - 5014055826994176n^8 - 23245912331160064n^7 - 5991956674019328n^6 + 13900726975086592n^5 + 9537554582155264n^4 + 1737844908349440n^3)$ | $\frac{1922723334840}{-1063412+327615\pi}$ | 4.15 |
| 96 | $\sum_{k=0}^{\infty} 4096^{-k} \left(42k + 5\right) \binom{2k}{k}^3$ | $\frac{16}{\pi}$ | $\mathbf{PCF}(21840n^4 + 67952n^3 + 73008n^2 + 29508n + 2607, -7225344n^8 - 12558336n^7 - 876544n^6 + 8491008n^5 + 5127168n^4 + 890368n^3)$ | $\frac{752}{16-5\pi}$ | 4.16 |
| | $\sum_{n=0}^{\infty} 2^{-12n} \left(42n + 5\right) \binom{2n}{n}^3$ | $\frac{16}{\pi}$ | | | |
| | $\sum_{n=0}^{\infty} 2^{-12n-4} \left(42n + 5\right) \binom{2n}{n}^3$ | $\frac{1}{\pi}$ | | | |
| | $\sum_{n=0}^{\infty} 4096^{-n} \left(n + \frac{5}{42}\right) \binom{2n}{n}^3$ | $\frac{8}{21\pi}$ | | | |
| | $\sum_{m=0}^{\infty} \frac{4096^{-m}(42m+5)(2m)!^3}{m!^6}$ | $\frac{16}{\pi}$ | | | |
| | $\sum_{n=0}^{\infty} \frac{64^{-2n}(42n+5)(2n)!^3}{n!^6}$ | $\frac{16}{\pi}$ | | | |
| | $\sum_{k=0}^{\infty} \frac{64^{-k}(42k+5)\left(\left(\frac{1}{2}\right)^{(k)}\right)^3}{k!^3}$ | $\frac{16}{\pi}$ | | | |
| | $\sum_{n=0}^{\infty} \frac{64^{-n}(42n+5)\left(\left(\frac{1}{2}\right)^{(n)}\right)^3}{n!^3}$ | $\frac{16}{\pi}$ | | | |
| | $\sum_{n=0}^{\infty} \frac{2^{-6n}(42n+5)\left(\left(\frac{1}{2}\right)^{(n)}\right)^3}{n!^3}$ | $\frac{16}{\pi}$ | | | |
| | $\sum_{n=0}^{\infty} \frac{2^{-6n}\left(n+\frac{5}{42}\right)\left(\left(\frac{1}{2}\right)^{(n)}\right)^3}{n!^3}$ | $\frac{8}{21\pi}$ | | | |
| | $\sum_{k=0}^{\infty} \frac{64^{-k}\left(\frac{21k}{8}+\frac{5}{16}\right)\left(\left(\frac{1}{2}\right)^{(k)}\right)^3}{k!^3}$ | $\frac{1}{\pi}$ | | | |
| 97 | $\sum_{k=0}^{\infty} \frac{4096^{-k}\left(28k^2-4k-1\right)\binom{2k}{k}^3}{(2k-1)^2}$ | $-\frac{3}{\pi}$ | $\mathbf{PCF}(14560n^5 + 41264n^4 + 36272n^3 + 6488n^2 - 3578n - 489, -3211264n^{10} + 2523136n^9 + 6930432n^8 - 4874240n^7 - 3977216n^6 + 2521088n^5 + 611328n^4 - 365056n^3)$ | $\frac{69}{3-\pi}$ | 4.16 |

| Cluster | Formula | Value | Canonical form (CF) | CF value | Convergence rate |
|---|---|---|---|---|---|
| $\delta = -0.50$ (cont.) | | | | | |
| 98 | $\sum_{k=0}^{\infty} \frac{4096^{-k}\left(42k^2-3k-1\right)\binom{2k}{k}^3}{(2k-1)^3}$ | $\frac{27}{8\pi}$ | PCF($21840n^5 + 63120n^4 + 58976n^3 + 15348n^2 - 2925n - 550, -7225344n^{10} + 11870208n^9 + 7790592n^8 - 21516288n^7 + 5676032n^6 + 7635456n^5 - 5004288n^4 + 856064n^3$) | $\frac{1026}{-27+8\pi}$ | 4.16 |
| 99 | $\sum_{n=0}^{\infty} \frac{4096^{-n}(2n+1)(6n+1)(14n-3)\binom{2n}{n}^3}{(2n-1)^2}$ | $-\frac{8}{\pi}$ | PCF($43680n^5 + 128624n^4 + 121776n^3 + 31176n^2 - 6934n - 1305, -28901376n^{10} - 11698176n^9 + 97304576n^8 + 12771328n^7 - 101953536n^6 + 12584960n^5 + 32441344n^4 - 10053120n^3$) | $\frac{1848}{8-3\pi}$ | 4.16 |
| 100 | $\sum_{k=0}^{\infty} \frac{4096^{-k}\left(56k^2+118k+61\right)\binom{2k}{k}^3}{(k+1)^2}$ | $\frac{192}{\pi}$ | PCF($29120n^5 + 206288n^4 + 567664n^3 + 758432n^2 + 493160n + 125163, -12845056n^{10} - 99090432n^9 - 309805056n^8 - 509747200n^7 - 479764480n^6 - 260722688n^5 - 76624896n^4 - 9241088n^3 + 358400n^2 + 120320n$) | $\frac{45120}{192-61\pi}$ | 4.16 |
| 101 | $\sum_{k=0}^{\infty} \frac{4096^{-k}\left(42k^2+81k+38\right)\binom{2k}{k}^3}{(k+1)^3}$ | $-512 + \frac{1728}{\pi}$ | PCF($21840n^5 + 172320n^4 + 529856n^3 + 789396n^2 + 566379n + 155809, -7225344n^{10} - 60383232n^9 - 210518016n^8 - 398905344n^7 - 447026176n^6 - 299017728n^5 - 111091200n^4 - 16026624n^3 + 2519040n^2 + 1073664n + 82432$) | $\frac{-139104+41216\pi}{-864+275\pi}$ | 4.16 |
| 102 | $\sum_{k=0}^{\infty} 4096^{-k} k \left(210k^2 - 5k + 1\right) \binom{2k}{k}^3$ | $\frac{4}{3\pi}$ | PCF($109200n^5 + 764320n^4 + 2076080n^3 + 2737956n^2 + 1761115n + 444325, -180663600n^{10} - 1888051200n^9 - 8364728320n^8 - 20346408960n^7 - 29314630656n^6 - 24973633024n^5 - 11606320128n^4 - 2246787072n^3 - 55296n^2 - 11487744n$) | $\frac{22975488}{1024-309\pi}$ | 4.16 |
| 103 | $\sum_{k=0}^{\infty} \frac{4096^{-k}\left(420k^2+992k+551\right)\binom{2k}{k}^3}{(k+1)^2(2k-1)}$ | $-\frac{1728}{\pi}$ | PCF($218400n^5 + 1599440n^4 + 4564144n^3 + 6337720n^2 + 4282754n + 1126485, -722534400n^{10} - 5219450880n^9 - 14278672384n^8 - 18150825984n^7 - 9547198464n^6 + 916187136n^5 + 3095921664n^4 + 1071358464n^3 + 48646144n^2 - 21106176n$) | $\frac{3392064}{-1728+551\pi}$ | 4.16 |
| 104 | $\sum_{k=0}^{\infty} \frac{4096^{-k} k \left(2128k^2+4050k+1861\right)\binom{2k}{k}^3}{(k+1)^3}$ | $31232 - \frac{98112}{\pi}$ | PCF($1106560n^6 + 15359184n^5 + 86935120n^4 + 255921504n^3 + 411374988n^2 + 340256619n + 112128678, -18548260864n^{12} - 376648237056n^{11} - 3433659121664n^{10} - 18548547182592n^9 - 65961076154368n^8 - 162097652928512n^7 - 280802726061056n^6 - 342734209244672n^5 - 288669949054976n^4 - 159659246702592n^3 - 52134516228096n^2 - 7603919511552n$) | $\frac{-133626321174528+42537276407808\pi}{-133955584+42639411\pi}$ | 4.16 |

| Cluster | Formula | Value | Canonical form (CF) | CF value | Convergence rate |
|---|---|---|---|---|---|
| $\delta = -0.50$ (cont.) | | | | | |
| 105 | $\sum_{k=0}^{\infty} \frac{4096^{-k} k^2 \left(78162 k^2 + 145175 k + 64431\right) \binom{2k}{k}^3}{(k+1)^3}$ | $-1321984 + \frac{4153360}{\pi}$ | $\mathrm{PCF}(40644240 n^7 + 603553472 n^6 + 3740699664 n^5 + 12504251940 n^4 + 24268130979 n^3 + 27259698627 n^2 + 16367928552 n + 4050187164, -25023685607424 n^{14} - 555894186860544 n^{13} - 5601703483531264 n^{12} - 33854715025600512 n^{11} - 136645305543681024 n^{10} - 387937847576512000 n^9 - 794055668810237952 n^8 - 1180136086421543424 n^7 - 1263161025479455744 n^6 - 948859671284379648 n^5 - 474348094193147904 n^4 - 141516190384201728 n^3 - 19023203258744832 n^2)$ | $\frac{-15328454371098624 0 + 48789345068384256 \pi}{-2126520320 + 676891779 \pi}$ | 4.16 |
| $\delta = -0.59$ | | | | | |
| 106 | $\sum_{k=0}^{\infty} (-1)^k 2^{-10k} \left( \frac{1}{10k+9} - \frac{4}{10k+7} - \frac{4}{10k+5} - \frac{64}{10k+3} + \frac{256}{10k+1} - \frac{1}{4k+3} - \frac{32}{4k+1} \right)$ | $64\pi$ | $\mathrm{PCF}(2684352000000000 n^{13} + 38090960000000000 n^{12} + 245000065120000000 n^{11} + 944869784312000000 n^{10} + 2435218351929120000 n^9 + 4424192355790080000 n^8 + 5821651713360384000 n^7 + 5609901140010836800 n^6 + 3952412820744570848 n^5 + 2007080164781810560 n^4 + 712485836218990920 n^3 + 166987020313319310 n^2 + 23083685299707597 n + 1415404977216930, 70506250240000000000000000000 n^{26} + 101374230528000000000000000000 n^{25} + 652528456826880000000000000000 n^{24} + 2465108672380928000000000000000 n^{23} + 5988171776162529280000000000000 n^{22} + 9507079778423275200000000000000 n^{21} + 9154983459598486732800000000000 n^{20} + 3155400509259243847680000000000 n^{19} - 4399616995545381312921600000000 n^{18} - 7236476374495916090982400000000 n^{17} - 4265372330790924839813120000000 n^{16} - 3967010516772650588569600000 0 n^{15} + 1608243322433796856212357120000 n^{14} + 1003875367531610931241943040000 n^{13} + 142208650082332906941579264000 n^{12} - 1313531726703519671553359872 00 n^{11} - 7633586386562807646890334617 6 n^{10} - 114520032395465020863781273 60 n^9 + 3524016088399505544133738496 n^8 + 17071901979908862507925504 n^7 + 18603073285535104741461196 8 n^6 - 32329095769907407247671296 n^5 - 10027373335724022275321856 n^4 - 5826353539459875202621 44 n^3 + 77843351835543790568448 n^2 + 115515364964990259548 16 n + 4177897462902030330330600)$ | $\frac{787032791496000 \pi}{31669 - 10080\pi}$ | 6.93 |
| | $\sum_{k=0}^{\infty} (-1)^k 2^{-10k} \left( \frac{1}{640k+576} - \frac{1}{256k+192} - \frac{1}{160k+112} - \frac{1}{160k+80} - \frac{1}{10k+3} + \frac{4}{10k+1} - \frac{1}{8k+2} \right)$ | $\frac{}{\pi}$ | | | |
| $\delta = -0.61$ | | | | | |
| 107 | $\sum_{k=0}^{\infty} \frac{(-64)^{-k} \left(28k^2 + 10k + 1\right) \binom{2k}{k}^5}{(6k+1) \binom{3k}{k} \binom{6k}{3k}}$ | $\frac{3}{\pi}$ | $\mathrm{PCF}(23296 n^7 + 125248 n^6 + 276544 n^5 + 320864 n^4 + 207200 n^3 + 72320 n^2 + 12192 n + 801, 21676032 n^{14} + 69672960 n^{13} + 53256192 n^{12} - 48660480 n^{11} - 89803776 n^{10} - 28091904 n^9 + 23292672 n^8 + 20539776 n^7 + 5356800 n^6 + 92352 n^5 - 164880 n^4 - 17784 n^3)$ | $\frac{117}{-3+\pi}$ | 3.30 |
| | $\sum_{n=0}^{\infty} \frac{\left(-\frac{1}{27}\right)^n \left(28n^2 + 10n + 1\right) \left(\left(\frac{1}{2}\right)_{(n)}\right)^5}{(6n+1) \left(\frac{1}{6}\right)_{(n)} \left(\frac{5}{6}\right)_{(n)} n!^3}$ | $\frac{3}{\pi}$ | | | |

| Cluster | Formula | Value | Canonical form (CF) | CF value | Convergence rate |
|---|---|---|---|---|---|
| $\delta = -0.64$ | | | | | |
| 108 | $\sum_{n=0}^{\infty} \frac{1458^{-n}(15n+2)(2n)!(3n)!}{n!^5}$ | $\frac{27}{4\pi}$ | PCF($3915n^4 + 12132n^3 + 13047n^2 + 5354n + 520, -984150n^8 - 1738665n^7 - 137781n^6 + 1151091n^5 + 668979n^4 + 107406n^3$) | $\frac{918}{27-8\pi}$ | 2.60 |
| | $\sum_{n=0}^{\infty} 2^{-n}3^{-6n}\left(n+\frac{2}{15}\right)\binom{2n}{n}^2\binom{3n}{n}$ | $\frac{9}{20\pi}$ | | | |
| | $\sum_{n=0}^{\infty} \frac{\left(\frac{27}{2}\right)^{-n}(15n+2)\left(\frac{1}{3}\right)_{(n)}\left(\frac{1}{2}\right)_{(n)}\left(\frac{2}{3}\right)_{(n)}}{n!^3}$ | $\frac{27}{4\pi}$ | | | |
| 109 | $\sum_{k=0}^{\infty} \frac{1458^{-k}\left(111k^2-7k-4\right)\binom{2k}{k}^2\binom{3k}{k}}{(2k-1)(3k-1)}$ | $-\frac{45}{4\pi}$ | PCF($28971n^5 + 82089n^4 + 74487n^3 + 17735n^2 - 4602n - 872, -53892054n^{10} + 33743223n^9 + 114043302n^8 - 66518334n^7 - 58348674n^6 + 31906143n^5 + 5149170n^4 - 2770200n^3$) | $\frac{4500}{45-16\pi}$ | 2.60 |
| 110 | $\sum_{k=0}^{\infty} 1458^{-k}k^3\left(900k^2 - 2097k + 929\right)\binom{2k}{k}^2\binom{3k}{k}$ | $-\frac{27}{20\pi}$ | PCF($234900n^5 + 683883n^4 + 611136n^3 + 69813n^2 - 136828n - 45024, -3542940000n^{10} - 13604889600n^9 - 4483826766n^8 + 40519719045n^7 + 52000563627n^6 + 2841693759n^5 - 19266142005n^4 - 4537514700n^3$) | $\frac{131876100}{6561-10720\pi}$ | 2.60 |
| 111 | $\sum_{k=0}^{\infty} \frac{1458^{-k}\left(1524k^2+899k+263\right)\binom{2k}{k}^2\binom{3k}{k}}{(k+1)(2k-1)(3k-1)}$ | $\frac{3375}{4\pi}$ | PCF($397764n^5 + 1757115n^4 + 2799174n^3 + 2058041n^2 + 755034n + 130504, -10158947424n^{10} - 17064898560n^9 + 13492053522n^8 + 24444146385n^7 - 13486145949n^6 - 12680237907n^5 + 6562189971n^4 + 1130544378n^3 - 579595824n^2$) | $\frac{9065250}{3375-1052\pi}$ | 2.60 |
| 112 | $\sum_{k=0}^{\infty} 1458^{-k}\left(56025k^2 + 112584k + 56551\right)\binom{2k}{k+1}^2\binom{3k}{k}$ | $\frac{6615}{4\pi}$ | PCF($14622525n^6 + 176113899n^5 + 855263871n^4 + 2136724149n^3 + 2887845868n^2 + 1997567096n + 553123200, -13729113933750n^{12} - 254250324752175n^{11} - 2087327331723069n^{10} - 10004733078193332n^9 - 31006251982463400n^8 - 64925286141963699n^7 - 93024351789956001n^6 - 90051659766526338n^5 - 56364085245495780n^4 - 20596254103734696n^3 - 3336437076176160n^2$) | $\frac{19513829338920}{321489-90064\pi}$ | 2.60 |

| Cluster | Formula | Value | Canonical form (CF) | CF value | Convergence rate |
|---------|---------|-------|---------------------|----------|------------------|

$\delta = -0.68$

| | | | | | |
|---|---|---|---|---|---|
| 113 | $\sum_{k=0}^{\infty} \frac{16^{-k}\left(120k^2+151k+47\right)}{512k^4+1024k^3+712k^2+194k+15}$ | $\pi$ | $\text{PCF}(1044480n^6 + 7458304n^5 + 21619136n^4 + 32438504n^3 + 26456230n^2 + 11070645n + 1852434, -60397977600n^{12} - 393593487360n^{11} - 1039730212864n^{10} - 1408627638272n^9 - 974252605440n^8 - 202269474816n^7 + 170414705664n^6 + 128014877184n^5 + 25005952192n^4 - 4802519872n^3 - 2768834448n^2 - 394428960n - 18316800)$ | $\frac{71550\pi}{-47+15\pi}$ | 2.77 |

$\frac{\sum_{i=0}^{\infty} 16^{-i}\left(-\frac{1}{8i+6}-\frac{1}{8i+5}-\frac{2}{8i+4}+\frac{4}{8i+1}\right)}{\pi}$

$\frac{\sum_{k=0}^{\infty} 16^{-k}\left(-\frac{1}{8k+6}-\frac{1}{8k+5}-\frac{2}{8k+4}+\frac{4}{8k+1}\right)}{\pi}$

$\frac{\sum_{m=0}^{\infty} 16^{-m}\left(-\frac{1}{8m+6}-\frac{1}{8m+5}-\frac{2}{8m+4}+\frac{4}{8m+1}\right)}{\pi}$

$\frac{\sum_{n=0}^{\infty} 16^{-n}\left(-\frac{1}{8n+6}-\frac{1}{8n+5}-\frac{2}{8n+4}+\frac{4}{8n+1}\right)}{\pi}$

$\frac{\sum_{k=0}^{\infty} 2^{-4k}\left(-\frac{1}{8k+6}-\frac{1}{8k+5}-\frac{2}{8k+4}+\frac{4}{8k+1}\right)}{\pi}$

| | | | | | |
|---|---|---|---|---|---|
| 114 | $\sum_{n=0}^{\infty} \frac{16^{-n}\left(40n^2-7n+1\right)\left(-\frac{1}{2}\right)_{(n)}\left(-\frac{3}{8}\right)_{(n)}\left(-\frac{1}{4}\right)_{(n)}\left(\frac{1}{8}\right)_{(n)}}{\left(\frac{1}{2}\right)_{(n)}\left(\frac{3}{4}\right)_{(n)}\left(\frac{9}{8}\right)_{(n)}\left(\frac{13}{8}\right)_{(n)}}$ | $\frac{5\pi}{16}$ | $\text{PCF}(348160n^6 + 1270272n^5 + 1612608n^4 + 788632n^3 + 85322n^2 - 12295n + 5514, -6710886400n^{12} + 9059696640n^{11} + 11161042944n^{10} - 18064867328n^9 - 3976396800n^8 + 11906629632n^7 - 1939592192n^6 - 2448268800n^5 + 933823680n^4 - 839872n^3 - 33462672n^2 + 1279968n + 391680)$ | $\frac{510\pi}{16-5\pi}$ | 2.78 |

| | | | | | |
|---|---|---|---|---|---|
| 115 | $\sum_{k=0}^{\infty} \frac{16^{-k}\left(120k^2+235k+99\right)\left(-\frac{3}{8}\right)_{(k)}\left(\frac{1}{8}\right)_{(k)}\left(\frac{1}{2}\right)_{(k)}\left(\frac{3}{4}\right)_{(k)}}{\left(\frac{9}{8}\right)_{(k)}\left(\frac{13}{8}\right)_{(k)}\left(\frac{5}{2}\right)_{(k)}\left(\frac{11}{4}\right)_{(k)}}$ | $\frac{63\pi}{2}$ | $\text{PCF}(1044480n^6 + 10094080n^5 + 39436736n^4 + 79283912n^3 + 85961374n^2 + 47272809n + 10188954, -60397977600n^{12} - 538548633600n^{11} - 1946303856640n^{10} - 3616349880320n^9 - 3524063330304n^8 - 1407537070080n^7 + 354994965504n^6 + 533609175552n^5 + 138379472320n^4 - 20669097856n^3 - 15123977328n^2 - 2291475168n - 109831680)$ | $\frac{28602\pi}{22-7\pi}$ | 2.78 |

$\sum_{n=0}^{\infty} \frac{16^{-n}\left(120n^2+235n+99\right)\left(-\frac{3}{8}\right)_{(n)}\left(\frac{1}{8}\right)_{(n)}\left(\frac{1}{2}\right)_{(n)}\left(\frac{3}{4}\right)_{(n)}}{\left(\frac{9}{8}\right)_{(n)}\left(\frac{13}{8}\right)_{(n)}\left(\frac{5}{2}\right)_{(n)}\left(\frac{11}{4}\right)_{(n)}}$
$\frac{63\pi}{2}$

| Cluster | Formula | Value | Canonical form (CF) | CF value | Convergence rate |
|---|---|---|---|---|---|
| $\delta = -0.68$ | | | | | |
| 116 | $\sum_{k=0}^{\infty} 16^{-k}\left(-\frac{1}{8k+7}+\frac{4}{8k+4}+\frac{4}{8k+3}+\frac{8}{8k+2}\right)$ | $2\pi$ | PCF($4177920n^7 + 31889408n^6 + 100376832n^5 + 167914400n^4 + 159965424n^3 + 85913234n^2 + 23812437n + 2611170, -966367641600n^{14} - 7135014420480n^{13} - 21160699887616n^{12} - 30977820721152n^{11} - 19361501806592n^{10} + 5089203585024n^9 + 16296553857024n^8 + 9574298615808n^7 + 722307942400n^6 - 1770106902528n^5 - 932890659904n^4 - 197879557824n^3 - 13151503632n^2 + 1452925152n + 199825920$) | $\frac{780570\pi}{-65+21\pi}$ | 2.77 |
| $\delta = -0.71$ | | | | | |
| 117 | $\sum_{k=0}^{\infty} \frac{16^{-k}\binom{2k}{k}}{2k+1}$ | $\frac{\pi}{3}$ | PCF($20n^2+44n+25, -64n^4-96n^3-48n^2-8n$) | $\frac{\pi}{-3+\pi}$ | 1.39 |
| 118 | $\sum_{k=0}^{\infty} \frac{2^{-4k}\binom{2k}{k}}{2k+1}$ 
 $\sum_{q=0}^{\infty} \frac{3\cdot 2^{-4q}\binom{2q}{q}}{2q+1}$ 
 $\sum_{i=1}^{\infty} \frac{16^{-i}\binom{2i}{i}}{2i+1}$ | $\frac{\pi}{3}$ 
 $\pi$ 
 $-1+\frac{\pi}{3}$ | PCF($20n^2+84n+89, -64n^4-352n^3-720n^2-648n-216$) | $\frac{216-72\pi}{25-8\pi}$ | 1.39 |
| 119 | $\sum_{n=1}^{\infty} \frac{16^{-n}\binom{2n}{n}}{2n+1}$ 
 $\sum_{n=0}^{\infty} \frac{16^{-n}(6n+5)\binom{2n}{n}}{(2n-1)(2n+1)(2n+3)}$ | $-1+\frac{\pi}{3}$ 
 $-\frac{\pi}{2}$ | PCF($120n^3+460n^2+514n+189, -2304n^6-7296n^5-4480n^4+2880n^3+1264n^2-264n$) | $\frac{33\pi}{10-3\pi}$ | 1.39 |
| $\delta = -0.74$ | | | | | |
| 120 | $\sum_{n=0}^{\infty} (-1)^n 4^{-n}\left(\frac{1}{4n+3}+\frac{2}{4n+2}+\frac{2}{4n+1}\right)$ | $\pi$ | PCF($1920n^5+11296n^4+25016n^3+25646n^2+11905n+1962, 1638400n^{10}+8355840n^9+15609856n^8+11624448n^7-371968n^6-5753472n^5-3107120n^4-303408n^3+229828n^2+74472n+6624$) | $\frac{414\pi}{10-3\pi}$ | 1.39 |

| Cluster | Formula | Value | Canonical form (CF) | CF value | Convergence rate |
|---|---|---|---|---|---|

$\delta = -0.74$
(cont.)

121 $\sum_{k=0}^{\infty} -16^{-k-1}\left(\frac{1}{8k+7} + \frac{2}{8k+6} + \frac{2}{8k+5} - \frac{4}{8k+3} - \frac{8}{8k+2} - \frac{8}{8k+1}\right)$

Value: $\frac{\pi}{4}$

CF: PCF($34225520640n^{11} + 406696493056n^{10} + 2143316606976n^9 + 6597952929792n^8 + 13147113750528n^7 + 17745614598144n^6 + 16486173371904n^5 + 10483446296768n^4 + 4438464202992n^3 + 1179435143940n^2 + 174547215315n + 10695273750, -64851834634135142400n^{22} - 77065596823563927552 0n^{21} - 3990364910235726905344n^{20} - 1165978791006482255052 8n^{19} - 2047478120290275714662 4n^{18} - 2000973501937303840358 4n^{17} - 4470827785003632427008n^{16} + 1479266653480626526617 6n^{15} + 2020516571640639153766 4n^{14} + 1026246192018929326489 6n^{13} - 1171047348656337321984n^{12} - 4651261809826751053824n^{11} - 2619491664736064372736n^{10} - 472035066965010677760n^9 + 169226552586375200768n^8 + 1137623682230599352 32n^7 + 23112713279753342976n^6 + 4465658932083916 8n^5 - 813054260841253632n^4 - 140873401622019456n^3 - 7987170386065680n^2 + 142566415135200n + 21743515584000$)

CF value: $\frac{3145763250\pi}{-109+35\pi}$

Convergence rate: 2.77

$\sum_{k=0}^{\infty} 16^{-k}\left(-\frac{1}{32k+28} - \frac{1}{16k+12} - \frac{1}{16k+10} + \frac{1}{8k+3} + \frac{2}{8k+2} + \frac{2}{8k+1}\right)$

Value: $\pi$

---

$\delta = -0.75$

122 $\sum_{k=0}^{\infty} 256^{-k}k^3\left(6k^2 - 19k + 6\right)\binom{2k}{k}^3$

Value: $-\frac{1}{12\pi}$

CF: PCF($240n^5 + 608n^4 + 120n^3 - 1008n^2 - 1193n - 440, -9216n^{10} - 19968n^9 + 61952n^8 + 190720n^7 + 60096n^6 - 166752n^5 - 74304n^4 + 41472n^3$)

CF value: $\frac{1728}{-8+21\pi}$

Convergence rate: 1.38

123 $\sum_{k=0}^{\infty} 256^{-k}\left(k^2 + 1\right)\left(192k^2 - 626k - 103\right)\binom{2k}{k}^3$

Value: $-\frac{1373}{3\pi}$

CF: PCF($7680n^7 + 1840n^6 - 41056n^5 - 84504n^4 - 105848n^3 - 90009n^2 - 37922n - 4370, -9437184n^{14} + 47382528n^{13} + 13908992n^{12} - 156557824n^{11} - 127080960n^{10} + 257140480n^9 + 117536448n^8 - 613944736n^7 - 357328896n^6 + 270441472n^5 + 247601920n^4 + 49146240n^3$)

CF value: $\frac{1474602}{-1373+309\pi}$

Convergence rate: 1.38

124 $\sum_{k=0}^{\infty} 256^{-k}\left(13608k^2 + 25050k + 10589\right)\binom{2k}{k+1}^3$

Value: $27296 - \frac{84604}{\pi}$

CF: PCF($544320n^8 + 8676912n^7 + 59045024n^6 + 223752600n^5 + 516148260n^4 + 742418115n^3 + 651298770n^2 + 319555044n + 67415544, -47405481984n^{16} - 1146343145472n^{15} - 12680362939392n^{14} - 84945259660800n^{13} - 384414182215168n^{12} - 1240661079383808n^{11} - 2936757342030912n^{10} - 5159671581960928n^9 - 6724120738987392n^8 - 6414378191822976n^7 - 4348938878740480n^6 - 1981782604878336n^5 - 543034218313728n^4 - 67406723610624n^3$)

CF value: $\frac{-538566289956864+173758988353536\pi}{-21658624+6938529\pi}$

Convergence rate: 1.38

| Cluster | Formula | Value | Canonical form (CF) | CF value | Convergence rate |
|---|---|---|---|---|---|
| $\delta = -0.75$ (cont.) | | | | | |
| 125 | $\sum_{n=0}^{\infty} \frac{256^{-n} n^2 \binom{2n}{n}^3}{(2n-3)(2n-1)}$ | $-\frac{1}{12\pi}$ | PCF($40n^3 + 148n^2 + 166n + 55, -256n^6 - 896n^5 - 832n^4 + 96n^3 + 288n^2$) | $\frac{72}{-8+3\pi}$ | 1.39 |
| 126 | $\sum_{k=0}^{\infty} 256^{-k} (6k+1) \binom{2k}{k}^3$ | $\frac{4}{\pi}$ | PCF($240n^4 + 736n^3 + 792n^2 + 336n + 39, -9216n^8 - 16896n^7 - 2560n^6 + 9984n^5 + 6336n^4 + 1120n^3$) | $\frac{28}{4-\pi}$ | 1.39 |
| | $\sum_{n=0}^{\infty} 256^{-n} (6n+1) \binom{2n}{n}^3$ | $\frac{4}{\pi}$ | | | |
| | $\sum_{n=0}^{\infty} 2^{-8n} (6n+1) \binom{2n}{n}^3$ | $\frac{4}{\pi}$ | | | |
| | $\sum_{n=0}^{\infty} 256^{-n} \left(n+\frac{1}{6}\right) \binom{2n}{n}^3$ | $\frac{2}{3\pi}$ | | | |
| | $\sum_{n=0}^{\infty} \frac{2^{-8n}(6n+1)(2n)!^3}{n!^6}$ | $\frac{4}{\pi}$ | | | |
| | $\sum_{k=0}^{\infty} (-1)^k 4^{-k} (6k+1) \left(-\frac{1}{2} \atop k\right)^3$ | $\frac{4}{\pi}$ | | | |
| | $\sum_{k=0}^{\infty} \frac{4^{-k}(6k+1)\left(\left(\frac{1}{2}\right)_{(k)}\right)^3}{k!^3}$ | $\frac{4}{\pi}$ | | | |
| | $\sum_{n=0}^{\infty} \frac{4^{-n}(6n+1)\left(\left(\frac{1}{2}\right)_{(n)}\right)^3}{n!^3}$ | $\frac{4}{\pi}$ | | | |
| | $\sum_{n=0}^{\infty} \frac{2^{-2n}(6n+1)\left(\left(\frac{1}{2}\right)_{(n)}\right)^3}{n!^3}$ | $\frac{4}{\pi}$ | | | |
| | $\sum_{k=0}^{\infty} \frac{4^{-k}\left(\frac{3k}{2}+\frac{1}{4}\right)\left(\left(\frac{1}{2}\right)_{(k)}\right)^3}{k!^3}$ | $\frac{1}{\pi}$ | | | |
| 127 | $\sum_{k=0}^{\infty} \frac{256^{-k} k(6k-1)\binom{2k}{k}^3}{(2k-1)^3}$ | $\frac{1}{2\pi}$ | PCF($240n^4 + 1280n^3 + 2504n^2 + 2120n + 651, -9216n^8 - 47616n^7 - 94720n^6 - 91904n^5 - 43840n^4 - 7712n^3 + 896n^2 + 352n$) | $\frac{176}{16-5\pi}$ | 1.39 |
| 128 | $\sum_{k=0}^{\infty} \frac{256^{-k}(12k^2-1)\binom{2k}{k}^3}{(2k-1)^2}$ | $-\frac{2}{\pi}$ | PCF($480n^5 + 1296n^4 + 1088n^3 + 208n^2 - 94n - 21, -36864n^{10} + 18432n^9 + 89088n^8 - 44544n^7 - 50944n^6 + 25472n^5 + 7744n^4 - 3872n^3$) | $\frac{22}{2-\pi}$ | 1.39 |
| $\delta = -0.75$ | | | | | |
| 129 | $\sum_{n=0}^{\infty} \frac{2^{-6n}\left(2100n^2+1160n+63\right)\left(\frac{1}{10}\right)_{(n)}\left(\frac{3}{10}\right)_{(n)}\left(\frac{7}{10}\right)_{(n)}\left(\frac{9}{10}\right)_{(n)}}{(2n+1)\left(\frac{1}{2}\right)_{(n)} n!^3}$ | $\frac{200}{\pi}$ | PCF($1365000000n^6 + 6886000000n^5 + 13628850000n^4 + 13283880000n^3 + 6461275900n^2 + 1346342040n + 61108047, -28224000000000000n^{12} - 101740800000000000n^{11} - 96724480000000000n^{10} + 470735360000000000n^9 + 126021900800000000n^8 + 54539290880000000n^7 - 12829349632000000n^6 - 14719958809600000n^5 - 3316987367040000n^4 - 201577965120000n^3$) | $\frac{125609400}{200-63\pi}$ | 4.16 |
| $\delta = -0.77$ | | | | | |
| 130 | $\sum_{j=0}^{\infty} \frac{4^{-j}\left(48j^2+76j+31\right)\left(\frac{3}{4}\right)_{(j)}\left(\frac{3}{2}\right)_{(j)}}{(j+1)(4j+1)\left(\frac{11}{8}\right)_{(j)}\left(\frac{15}{8}\right)_{(j)}}$ | $\frac{21\pi}{2}$ | PCF($15360n^6 + 121088n^5 + 387904n^4 + 645552n^3 + 588500n^2 + 279001n + 53940, -37748736n^{12} - 346030080n^{11} - 1354301440n^{10} - 2947039232n^9 - 3879428096n^8 - 3125897216n^7 - 1452979712n^6 - 302704640n^5 + 15363632n^4 + 10035756n^3 - 4167378n^2 - 1802088n - 175770$) | $\frac{58590\pi}{-62+21\pi}$ | 1.39 |

| Cluster | Formula | Value | Canonical form (CF) | CF value | Convergence rate |
|---|---|---|---|---|---|
| $\delta = -0.77$ | | | | | |
| 131 | $\sum_{j=0}^{\infty} \frac{\left(\frac{27}{4}\right)^{-j}\left(368j^3+400j^2+118j+9\right)\left(\left(\frac{1}{4}\right)_{(j)}\right)^2\left(\frac{1}{2}\right)_{(j)}\left(\left(\frac{3}{4}\right)_{(j)}\right)^2}{\left(\frac{4}{3}\right)_{(j)}\left(\frac{5}{3}\right)_{(j)}j!^3}$ | $\frac{32}{\pi}$ | PCF($1460224n^8 + 10254336n^7 + 30198016n^6 + 48411008n^5 + 45785184n^4 + 25791664n^3 + 8263468n^2 + 1327140n + 77175, -239628976128n^{16} - 1359633973248n^{15} - 2762609786880n^{14} - 1761717780480n^{13} + 1754200080384n^{12} + 3409371168768n^{11} + 1472080920576n^{10} - 671023939584n^9 - 863464676352n^8 - 256765588992n^7 + 24587248896n^6 + 31003596672n^5 + 6642741888n^4 + 476340480n^3$) | $\frac{257760}{32-9\pi}$ | 1.91 |
| $\delta = -0.77$ | | | | | |
| 132 | $\sum_{j=1}^{\infty} \frac{\left(\frac{256}{27}\right)^{-j}\left(32976j^5+67896j^4+52844j^3+19156j^2+3167j+180\right)\left(\left(\frac{1}{8}\right)_{(j)}\right)^2\left(\frac{1}{2}\right)_{(j)}\left(\left(\frac{5}{8}\right)_{(j)}\right)^2 j!}{j\left(\frac{7}{8}\right)_{(j)}\left(\frac{9}{8}\right)_{(j)}\left(\frac{4}{3}\right)_{(j)}\left(\frac{11}{8}\right)_{(j)}\left(\frac{13}{8}\right)_{(j)}\left(\frac{5}{3}\right)_{(j)}}$ | $-149+60\pi$ | PCF($2687675904n^{11} + 56471427072n^{10} + 535315804416n^9 + 3021521573952n^8 + 11281513939616n^7 + 29252467524640n^6 + 53743687923752n^5 + 69953516648740n^4 + 63210207439520n^3 + 37759710234253n^2 + 13419573808347n + 2149364047698, -623425631475990528n^{22} - 19087986092113133568n^{21} - 275348137430420029440n^{20} - 2488293166274605744128n^{19} - 15800171388967065010176n^{18} - 74944821953597729955840n^{17} - 275584389035044674392064n^{16} - 804682855975385924425728n^{15} - 1895673749567702243880960n^{14} - 3640715324669684509461504n^{13} - 5736185376897136561716480n^{12} - 7436162403767341956209280n^{11} - 7931362314765912632102016n^{10} - 6939790953137810855789760n^9 - 4951447084034730814786848n^8 - 2853248226731004784120704n^7 - 1309335440321591091369804n^6 - 468917015270723575079178n^5 - 127291005254093541357168n^4 - 25070604686415888998310n^3 - 3339540353061295760916n^2 - 264619352541786580800n - 9163515571913102400$) | $\frac{50569030378352688-203633679375846720\pi}{4460671-1441440\pi}$ | 2.25 |
| $\delta = -0.78$ | | | | | |
| 133 | $\sum_{j=1}^{\infty} \frac{(-16)^j\left(40j^2-12j-1\right)\binom{2j}{j}}{j(2j-1)^2(4j+1)\binom{4j}{2j}^2}$ | $8-4\pi$ | PCF($7680n^6 + 70656n^5 + 261696n^4 + 498528n^3 + 513624n^2 + 270156n + 56295, 26214400n^{12} + 259522560n^{11} + 1106640896n^{10} + 2656960512n^9 + 3926888448n^8 + 3637020672n^7 + 1998611456n^6 + 483591168n^5 - 103859648n^4 - 110540832n^3 - 31619952n^2 - 3824280n - 145800$) | $\frac{-32400+16200\pi}{16-5\pi}$ | 1.39 |

| Cluster | Formula | Value | Canonical form (CF) | CF value | Convergence rate |
|---|---|---|---|---|---|
| $\delta = -0.80$ | | | | | |
| 134 | $\sum_{k=0}^{\infty} (-1024)^{-k} k^3 \left(400k^2 - 496k + 327\right) \binom{2k}{k}^2 \binom{4k}{2k}$ | $\frac{9}{5\pi}$ | $\mathrm{PCF}(38400n^5 + 99584n^4 + 25408n^3 - 156256n^2 - 188394n - 68607, 655360000n^{10} + 3945267200n^9 + 8690139136n^8 + 7602667520n^7 + 1455075328n^6 + 2245892096n^5 + 6993980160n^4 + 4109212800n^3)$ | $-\frac{12566400}{128 + 385\pi}$ | 1.38 |
| 135 | $\sum_{k=0}^{\infty} (-1024)^{-k} \left(19840k^2 + 39324k + 19481\right) \binom{2k}{k+1}^2 \binom{4k}{2k}$ | $-\frac{1000}{\pi}$ | $\mathrm{PCF}(1904640n^6 + 22504064n^5 + 106510496n^4 + 256884424n^3 + 330495838n^2 + 213084771n + 53326350, 1612290457600n^{12} + 29769513041920n^{11} + 243609643319296n^{10} + 1163447016357888n^9 + 3591108030627840n^8 + 7484964318959616n^7 + 10668120581775104n^6 + 10265556721549952n^5 + 6381908722722560n^4 + 2314389737823744n^3 + 371767948231680n^2)$ | $\frac{3816723456000}{-102400 + 47187\pi}$ | 1.38 |
| 136 | $\sum_{k=0}^{\infty} \frac{(-1024)^{-k}\left(8k^2 - 2k - 1\right)\binom{2k}{k}^2\binom{4k}{2k}}{(k+1)(2k-1)(4k-1)}$ | $-\frac{16}{5\pi}$ | $\mathrm{PCF}(96n^3 + 464n^2 + 634n + 261, 4096n^6 + 10240n^5 + 6912n^4 + 128n^3 - 640n^2)$ | $\frac{80}{16 - 5\pi}$ | 1.39 |
| 137 | $\sum_{k=0}^{\infty} (-1024)^{-k} \left(20k + 3\right) \binom{2k}{k}^2 \binom{4k}{2k}$ | $\frac{8}{\pi}$ | $\mathrm{PCF}(1920n^4 + 6368n^3 + 7288n^2 + 3146n + 315, 1638400n^8 + 2949120n^7 + 262144n^6 - 1910784n^5 - 1054976n^4 - 150144n^3)$ | $\frac{552}{-8 + 3\pi}$ | 1.39 |
| | $\sum_{n=0}^{\infty} (-1)^n \, 2^{-10n} \left(20n + 3\right) \binom{2n}{n}^2 \binom{4n}{2n} \frac{8}{\pi}$ | | | | |
| | $\sum_{k=0}^{\infty} \frac{(-4)^{-k}(20k+3)\left(\frac{1}{4}\right)_{(k)}\left(\frac{1}{2}\right)_{(k)}\left(\frac{3}{4}\right)_{(k)}}{k!^3} \quad \frac{8}{\pi}$ | | | | |
| | $\sum_{n=0}^{\infty} \frac{(-1)^n 4^{-n}(20n+3)\left(\frac{1}{4}\right)_{(n)}\left(\frac{1}{2}\right)_{(n)}\left(\frac{3}{4}\right)_{(n)}}{n!^3} \frac{8}{\pi}$ | | | | |
| | $\sum_{n=0}^{\infty} \frac{(-1)^n 2^{-2n}(20n+3)\left(\frac{1}{4}\right)_{(n)}\left(\frac{1}{2}\right)_{(n)}\left(\frac{3}{4}\right)_{(n)}}{n!^3} \frac{8}{\pi}$ | | | | |
| | $\sum_{n=0}^{\infty} \frac{\left(-\frac{1}{4}\right)^n (20n+3)\left(\frac{1}{4}\right)_{(n)}\left(\frac{1}{2}\right)_{(n)}\left(\frac{3}{4}\right)_{(n)}}{n!^3} \quad \frac{8}{\pi}$ | | | | |
| 138 | $\sum_{k=0}^{\infty} \frac{(-1024)^{-k}\left(40k^2 - 2k - 1\right)\binom{2k}{k}^2\binom{4k}{2k}}{(2k-1)(4k-1)}$ | $-\frac{4}{\pi}$ | $\mathrm{PCF}(3840n^5 + 13248n^4 + 14608n^3 + 4676n^2 - 644n - 165, 6553600n^{10} - 3932160n^9 - 13500416n^8 + 7626752n^7 + 6695936n^6 - 3568128n^5 - 367360n^4 + 194176n^3)$ | $\frac{148}{-4 + \pi}$ | 1.39 |
| $\delta = -0.86$ | | | | | |
| 139 | $\sum_{k=0}^{\infty} 648^{-k} k^3 \left(854k^2 - 3639k + 910\right) \binom{2k}{k}^2 \binom{4k}{2k}$ | $-\frac{243}{14\pi}$ | $\mathrm{PCF}(96502n^5 + 166951n^4 - 357406n^3 - 1238824n^2 - 1267179n - 461835, -1890387072n^{10} + 42057792n^9 + 28244340360n^8 + 34524320580n^7 - 71756129178n^6 - 143426873097n^5 - 43072909110n^4 + 22847151600n^3)$ | $\frac{677882520}{-2187 + 8750\pi}$ | 0.92 |
| 140 | $\sum_{k=0}^{\infty} 648^{-k} \left(7k + 1\right) \binom{2k}{k}^2 \binom{4k}{2k}$ | $\frac{9}{2\pi}$ | $\mathrm{PCF}(791n^4 + 2374n^3 + 2482n^2 + 1007n + 105, -127008n^8 - 226800n^7 - 17334n^6 + 149769n^5 + 82134n^4 + 11664n^3)$ | $\frac{216}{9 - 2\pi}$ | 0.93 |
| | $\sum_{n=0}^{\infty} \frac{\left(\frac{81}{32}\right)^{-n}(7n+1)\left(\frac{1}{4}\right)_{(n)}\left(\frac{1}{2}\right)_{(n)}\left(\frac{3}{4}\right)_{(n)}}{n!^3}$ | $\frac{9}{2\pi}$ | | | |

| Cluster | Formula | Value | Canonical form (CF) | CF value | Convergence rate |
|---------|---------|-------|---------------------|----------|------------------|
| $\delta = -0.86$ (cont.) | | | | | |
| 141 | $\sum_{k=0}^{\infty} \frac{648^{-k}\left(122k^2+3k-5\right)\binom{2k}{k}^2\binom{4k}{2k}}{(2k-1)(4k-1)}$ | $-\frac{21}{2\pi}$ | PCF($13786n^5 + 35841n^4 + 29614n^3 + 7104n^2 - 965n - 285, -38579328n^{10} + 17392320n^9 + 83657448n^8 - 39260700n^7 - 41524002n^6 + 20115135n^5 + 2277882n^4 - 1108080n^3$) | $\frac{2520}{21-10\pi}$ | 0.93 |
| 142 | $\sum_{k=0}^{\infty} \frac{648^{-k}\left(1903k^2+114k+41\right)\binom{2k}{k}^2\binom{4k}{2k}}{(k+1)(2k-1)(4k-1)}$ | $\frac{343}{2\pi}$ | PCF($215039n^5 + 720798n^4 + 810302n^3 + 328875n^2 + 34877n + 8700, -9386692128n^{10} - 5817973392n^9 + 23052928050n^8 + 11193948009n^7 - 19783391157n^6 - 6108011388n^5 + 6029620560n^4 + 339444756n^3 - 305057340n^2$) | $\frac{705894}{343-82\pi}$ | 0.93 |
| 143 | $\sum_{k=0}^{\infty} 648^{-k}\left(2107k^2+4359k+2249\right)\binom{2k}{k+1}^2\binom{4k}{2k}$ | $\frac{567}{\pi}$ | PCF($238091n^6 + 2907189n^5 + 14353943n^4 + 36596679n^3 + 50726276n^2 + 36214482n + 10426185, -11507051808n^{12} - 214464248208n^{11} - 1771630180902n^{10} - 8542913475033n^9 - 26632332249786n^8 - 56090452945077n^7 - 80827637149008n^6 - 78693324310107n^5 - 49539739902066n^4 - 18209558822580n^3 - 2967860914200n^2$) | $\frac{11876722200}{2916-415\pi}$ | 0.93 |
| $\delta = -0.86$ | | | | | |
| 144 | $\sum_{j=0}^{\infty} \frac{4^{-j}\left(18j^2+19j+6\right)\left(\left(\frac{1}{6}\right)_{(j)}\right)^2 j!}{\left(\frac{13}{12}\right)_{(j)}\left(\frac{19}{12}\right)_{(j)}\left(\frac{11}{6}\right)_{(j)}}$ | $\frac{35\pi}{18}$ | PCF($19440n^5 + 103464n^4 + 208296n^3 + 197430n^2 + 89543n + 16560, -60466176n^{10} - 298971648n^9 - 527726016n^8 - 341467488n^7 + 40616640n^6 + 115455024n^5 + 1562292n^4 - 29814726n^3 - 9904998n^2 - 1164570n - 45150$) | $\frac{9030\pi}{-108+35\pi}$ | 1.39 |

| Cluster | Formula | Value | Canonical form (CF) | CF value | Convergence rate |
|---|---|---|---|---|---|
| $\delta = -0.89$ | | | | | |
| 145 | $\sum_{j=0}^{\infty} \frac{4^{-j}\left(108j^2+72j+5\right)\left(\left(\frac{1}{6}\right)_{(j)}\right)^2\left(\left(\frac{5}{6}\right)_{(j)}\right)^2}{\left(\frac{3}{2}\right)_{(j)}j!^3}$ | $\frac{18}{\pi}$ | PCF($699840n^6 + 3545856n^5 + 7069680n^4 + 6990624n^3 + 3499812n^2 + 776232n + 43505, -78364164096n^{12} - 300395962368n^{11} - 325066162176n^{10} + 76429246464n^9 + 351120365568n^8 + 187384679424n^7 - 7603808256n^6 - 29844910080n^5 - 7099617600n^4 - 491508000n^3$) | $\frac{83250}{18-5\pi}$ | 1.39 |
| 146 | $\sum_{n=0}^{\infty} \frac{2^{-8n}3^{-6n}\left(25-108n^2\right)\binom{4n}{2n}\binom{6n}{3n}\binom{6n}{4n}}{(6n-5)^2}$ | $\frac{3}{5\pi}$ | PCF($699840n^6 + 2239488n^5 + 2302992n^4 + 557280n^3 - 385164n^2 - 189000n - 12925, -78364164096n^{12} + 39182082048n^{11} + 301847150592n^{10} - 150923575296n^9 - 352141572096n^8 + 176070786048n^7 + 157361356800n^6 - 78680678400n^5 - 22320360000n^4 + 11160180000n^3$) | $\frac{155625}{3-5\pi}$ | 1.39 |
| $\delta = -1.00$ | | | | | |
| 147 | $\sum_{k=1}^{\infty}(-1)^{k+1}\left(\frac{1}{6k-1}+\frac{1}{6k-5}\right)$ | $\frac{\pi}{3}$ | PCF($72n^2 + 144n + 62, 5184n^6 + 15552n^5 + 13104n^4 + 288n^3 - 3428n^2 - 980n - 75$) | $\frac{75\pi}{18-5\pi}$ | 0.00 |
| $\delta = -1.00$ | | | | | |
| 148 | $\sum_{n=0}^{\infty} \frac{\left(\frac{1}{6}\right)_{(n)}\left(\frac{5}{6}\right)_{(n)}}{(n+1)!^2}$ | $\frac{648-180\pi}{25\pi}$ | PCF($72n^2 + 180n + 149, -1296n^4 - 3888n^3 - 4068n^2 - 1656n - 180$) | $\frac{-3240+900\pi}{-648+205\pi}$ | 0.00 |
| $\delta = -1.00$ | | | | | |
| 149 | $\sum_{n=1}^{\infty} \frac{1}{n(2n-1)(4n-3)}$ | $\frac{\pi}{3}$ | PCF($16n^3 + 52n^2 + 66n + 31, -64n^6 - 224n^5 - 308n^4 - 212n^3 - 77n^2 - 14n - 1$) | $\frac{\pi}{-3+\pi}$ | 0.00 |
| | $\sum_{n=1}^{\infty} \frac{3}{4n(4n-3)(4n-2)}$ | $\frac{\pi}{8}$ | | | |
| | $\sum_{n=0}^{\infty} \frac{1}{(n+1)(2n+1)(4n+1)}$ | $\frac{\pi}{3}$ | | | |

