# OpenReview forum: "From Euler to AI: Unifying Formulas for Mathematical Constants"
_NeurIPS.cc/2025/Conference — NeurIPS 2025 poster_

### Official Review · Reviewer_5mQZ · 2025-06-28

**Clarity:** 3
**Significance:** 2
**Originality:** 3
**Rating:** 3
**Confidence:** 3

**Summary:**

This paper presents a new AI methodology to unify different formulas which correspond to the same mathematical object. In particular, the paper is focused on one use case where the object is the constant $\pi$, but also authors extended to $e$ or $\zeta(3)$.

The methodology can be described by the following pipeline:
- Extraction: regex and an LLM is used to extract mathematical expressions from ArXiV papers and make it more standardized
- Validation: use PSLQ to check if the new formula actually calculates $\pi$ or if the LLM makes some mistakes
- Unification: all formulas are converted into the canonical form using RISC and using UMAPS algorithm and finding a matrix $U$ which satisfies the equation (8).
The unification step is the final goal of this work, where it's possible to prove equivalence of different formulas.

**Questions:**

Q1. I would be more interested if we can develop a system which, based on this work, is able to calculate some constants with faster convergence rate, which is an active research area.

Q2. Can you elaborate more on why you believe unification is very important for unifying knowledge across different subjects? In particular, most of these objects are generated from pure math and then used in other fields, but the source is the pure math.

**Ethical Concerns:**

["NO or VERY MINOR ethics concerns only"]

**Final Justification:**

The two main points were solved, although I'm still struggling to see the significance of these claims.
The current manuscript is insufficient and newly added contents (during rebuttal and discussions) are substantial and require rigorous re-evaluation.

**Limitations:**

yes

**Quality:**

2

**Strengths And Weaknesses:**

For the strengths component

S1. The paper is very well written, easy to follow and, although I don't have expertise in this area, seems pretty novel as well.

S2. A good benchmarking exercise has been done, although it's very focused on $\pi$ only.

I see two main weaknesses

W1. I don't see how this paper is interesting for an AI conference. There is no advancement in AI, nor AI has been used in a clever or interesting way. The AI part is just for filtering and generating equations which needs to be the same as the TeX formulas. Then, there is no data/training/evaluation using AI.

W2. The contributions part from this work is quite limited. Showing that formula $x_1$ is the same as formula $x_2$ cannot be seen as a novel contribution (as the authors acknowledged, there are already specialized systems) and it's not bringing any meaningful and additional knowledge to the scientific community.

---

> ### Author Rebuttal · Authors · 2025-07-31
>
> - *“The paper is very well written, easy to follow and, although I don't have expertise in this area, seems pretty novel as well.”*
>
> We thank the reviewer for this positive feedback.
>
> ---
>
> - *“A good benchmarking exercise has been done, although it's very focused on $\pi$ only.”*
>
> Running the entire pipeline for other constants like Catalan’s or zeta(3) is possible with only slight modifications. The benchmark is demonstrated on $\pi$ because it has many known formulas that make a large-scale statistical analysis more precise.
>
> ---
>
> - *“W1. …There is no advancement in AI, nor AI has been used in a clever or interesting way. The AI part is just for filtering and generating equations which needs to be the same as the TeX formulas. Then, there is no data/training/evaluation using AI.”*
>
> We appreciate the reviewer’s comment and welcome the opportunity to clarify how this work fits within the scope and interests of the AI community, and NeurIPS in particular. Our submission does not involve training a new model, but applies LLMs to a domain where AI has had limited reach: number theory. By combining LLM-based extraction with symbolic validation tools and algorithms like PSLQ and UMAPS, we enable the first successful application of LLMs in this space.
>
> A key component is LLM-based filtering. Because mathematical papers vary widely in definitions and notation, and no deterministic ruleset can precisely isolate expressions related to targets like $\pi$, we rely on LLMs to interpret and filter candidate formulas across thousands of documents. This usage is essential for identifying useful mathematical content at scale.
>
> Our pipeline reflects a broader trend in AI for mathematics: combining LLMs with formal or symbolic reasoning. PAL (NeurIPS 2022), Parsel and DELI (NeurIPS 2023), and Self-Discover (NeurIPS 2024) all used existing LLMs alongside external computation, achieving strong results without model training. We now cite these works to put our contribution in this broader context of LLM integration with computer algebra systems (CAS).
>
> NeurIPS has actively encouraged this kind of interdisciplinary infrastructure work, especially as LLMs begin to interface more deeply with formal tools, numerical methods, and structured mathematical knowledge. We focused on number theory, a domain that remains underexplored in current AI pipelines.
>
> ---
>
> - *“W2. …Showing that formula $x_1$ is the same as formula $x_2$ cannot be seen as a novel contribution…and it's not bringing any meaningful and additional knowledge to the scientific community.”*
>
> We appreciate the reviewer’s comments and opportunity to explain the contributions of our work. Our algorithm is the first to unify formulas in bulk, with no prior work achieving this. This approach has multiple novel contributions:
>
> 1. **Structure discovery**: The system automatically finds implications of the form FormulaA = LimitA iff FormulaB = LimitB, resulting in a streamlined theory of $\pi$ computation. The knowledge contained in 381 formulas from 151 papers is compactified to 136 unique canonical forms which are further condensed into just 32 formulas. This technique greatly increases the efficiency of formula proving, as proof of any formula in a cluster proves all the others systematically, verifiably and automatically.
>
> 2. **Novel dataset to gauge mathematical reasoning**: Our results provide a dataset of equivalent formula pairs, useful for benchmarking LLMs on the ability to transfer and unify mathematical knowledge. To demonstrate this contribution, we tested state-of-the-art LLMs, asking them to prove the equivalence of formula pairs (see Section 4), showing that this benchmark is indeed useful for gauging their math reasoning.
>
> 3. **CMFs for formula generation**: The CMF generates infinitely many $\pi$-formulas, some of which yield faster convergence than the original formulas. Such formulas can be used for irrationality proofs - for example, automatically generalizing formulas to reveal Apéry’s creative sequence along a diagonal, illustrating how unifying structures can uncover deep number-theoretic insights.
>
> 4. **Exploring individually the 1021 equivalences identified in our work could lead to advances in mathematics**: Similar situations have happened in the past, where equivalent formulas for constants revealed connections among special functions, continued fractions, lattice sums, Fourier–theta dualities, modular forms, and Feynman integrals. Euler’s comparison of two formulas for $\pi$ - the Wallis product and Brouncker’s continued fraction - revealed the hidden structure of beta and gamma functions, now central to analysis and physics. In the 1800s, efforts to compute $\pi$ via Buffon’s needle experiments and Gauss’s lattice point counts culminated in the Poisson summation formula, connecting geometric probability and theta-function theory. In the 20th century, Ramanujan’s many rapidly converging series for $1/\pi$ were traced to modular forms and elliptic functions, leading to both the fastest known $\pi$-computation algorithms and the theory of Ramanujan–Sato series. In each case, the breakthrough emerged not from a single formula, but from uncovering connections among different formulas. Our work enables this kind of discovery at an unprecedented scale.
>
> ---
>
> - *“Q1. I would be more interested if we can develop a system which, based on this work, is able to calculate some constants with faster convergence rate, which is an active research area.”*
>
> Although this was not in our original plan for this paper (was planned for a future work), this comment made us use our developed formalism to find a faster converging formula, compared with the known ones. We are glad to report that we found such a faster converging formula and it is now added to the findings of the paper. The *normalized* convergence rate we found is 1.76, for a formula in the CMF with trajectory (-1,-1,0). In comparison, the trajectory (1,1,2) is the best pre-existing formula found in the CMF and has a *normalized* convergence rate of 0.88. We added a discussion of how this result was achieved to Appendix E.
>
> **Additional details**:
>
> To make a fair test of convergence, one has to *normalize* the convergence rate by the “length” or “complexity” of the trajectory. This normalization avoids trivial improvements, such as artificially accelerating convergence by taking every k-th step, which can misleadingly inflate the raw rate. The normalized rate is the convergence rate divided by the number of CMF steps (i.e., the trajectory's length in the lattice). This means that trajectories along the same line such as (1,2,5) and (2,4,10) will have the same normalized convergence rate.
>
> Using this metric, we are now able to map the convergence landscape over the CMF directions. In the revised appendix, we include:
> - A visualization ("convergence map") over the unit sphere of directions.
> - A detailed discussion on the convergence rate and how CMF trajectories change this rate in a continuous (in the spherical angle) manner.
>
> In summary, the CMF framework not only unifies symbolic formulas but also enables structured exploration of convergence behavior, which may help guide the discovery of new, fast-converging formulas in future work.
>
> ---
>
> - *“Q2. Can you elaborate more on why you believe unification is very important for unifying knowledge across different subjects? In particular, most of these objects are generated from pure math and then used in other fields, but the source is the pure math.”*
>
> We appreciate the opportunity to clarify this point.
>
> Mathematical progress often arises from finding new ways to conceptualize ideas that appear across different areas. One of the most far-reaching examples is the development of category theory, which provides a language for describing patterns and relationships shared across seemingly unrelated mathematical structures. Initially, this language didn't introduce new results but offered a unifying lens that has since become foundational in many fields, enabling new questions (such as - given a functor what is its adjoint) and methods of reasoning (such as - this object is an initial object in the category).
>
> In a similar spirit, our proposed work aims to unify the many expressions for mathematical constants that appear in varied domains. These constants, such as $\pi$ or logarithmic values, frequently emerge in formulas tied to geometry, topology, number theory, and mathematical physics. Although these formulas originate in pure mathematics, they are scattered across different subfields and often studied in isolation.
>
> As a concrete case, recent work by Bettin and Drapeau (Modularity and Value Distribution of Quantum Invariants of Hyperbolic Knots, 2020) demonstrates that constants related to volume and modularity—particularly $\pi$—arise in surprisingly structured ways. They show that the asymptotic behavior of quantum invariants, such as the colored Jones polynomial of the figure-eight knot, encodes the hyperbolic volume of the knot complement. This leads to a precise numerical expression involving $\pi$ - found in our CMF by UMAPS as coboundary to the direction (1,1,1). One can imagine a direct use of  the CMF for this problem  by identifying possible relationships between the parameters defining a knot (like its crossing number or twist pattern) and the directions within the CMF that represent the derived formula for the volume. Then, by just analyzing the CMF (which is easy to compute), one could reveal new insights into the invariants of different knot types.
>
> Unification, in this sense, is not just about organizing known results. It creates a higher-level language for recognizing deep patterns and making new connections—often across fields that traditionally develop in isolation. By framing diverse formulas within a common structure, the CMF enables a more global view of mathematical knowledge and opens new directions for inquiry grounded in pure mathematics.

---

### Official Review · Reviewer_3XHw · 2025-06-29

**Clarity:** 2
**Significance:** 3
**Originality:** 3
**Rating:** 5
**Confidence:** 4

**Summary:**

This work searches a large mathematical literature (drawn from arXiv) and extracts convergent sequence representations of standard mathematical constants such as $\pi$ and $e$, then uses LLMs to bring them to a standard form.  Then, several mathematical techniques are used to find relations between superficially different sequence representations: computation of invariants ("dynamical metrics") such as convergence rate and irrationality measure, clustering, then an algorithm for locating recurrence relations within a more general structure called a "conservative matrix field".  The search step uses LLMs and several evaluations and comparisons between LLMs are given.  Many relations between sequence representations are found.  For $\pi$ the fraction of representations satisfying relations is quantified and many canonical forms listed.

**Questions:**

In section 3 describing the literature extraction, the fraction which lead to validated formulas, while large, seems small enough (maybe 15%) to wonder how representative the final results are of the full range of sequence representations in the literature.  For example, it is stated that there are 3389 computations of $\pi$, which are categorized by type as series, continued fraction, or neither, resulting in 1678 formula candidates.  How many of these 1678 formula candidates were continued fractions?  What are the other formulas, are they just explicit series with no other apparent structure?  Then in 3.2 which verifies that the sequences represent $\pi$, it is said that "Out of the 682 candidates, 407 were validated as $\pi$ formulas and passed on for canonicalization."   Where did these numbers come from?  In 3.3 it says "Out of the 407 validated formulas, 402 are found to have 146 representations as order-2 recurrences, and 5 as order-3 recurrences."   Is this to say that almost all of the validated formulas were in fact continued fractions?

The biggest question the results raised in my mind was to what extent the authors believe the "conservative matrix field" approach unifies all sequence representations or not.  In the final table 10 many conjectures are stated about such relations.  Are there representations of $\pi$ which the authors believe are *not* instances of the 3d CMF of D.2 ?  I suppose the CMF concept could be generalized to Z^D for higher D, is it possible that any recurrence relation could be accommodated this way, or are there constraints which prevent combining some pairs of recurrences into the same CMF ?  The simplest such constraints might be relations between the "dynamical metrics," is this so and where is it mentioned in the text?

**Ethical Concerns:**

["NO or VERY MINOR ethics concerns only"]

**Final Justification:**

The method and results of this paper are interesting to mathematicians and the quality of the writing and analysis, while not great, is adequate.  The authors addressed my most pressing concerns.  I feel the other reviewers are just not very interested in this topic and are rejecting it for this reason.  Perhaps it should go to a different conference focused on AI for mathematics.

**Limitations:**

Yes.

**Quality:**

3

**Strengths And Weaknesses:**

Strengths: the paper asks and studies an interesting "meta-mathematical" question about the many solutions to a well known problem (convergent sequence representations of standard constants) by automated literature search and canonicalization.  Most automated literature searches are fairly superficial and I know of few if any other works which automate such a mathematically interesting analysis.

Weaknesses: I find the mathematical part of the paper poorly written, explaining the concepts and methods in a very disjointed way.  The relation to previous work other than O. David's or the authors is unclear, and the invention of new terminology for standard mathematical concepts, such as "conservative matrix field" for a flat connection, is not reassuring.

---

> ### Author Rebuttal · Authors · 2025-07-31
>
> - *“Strengths: the paper asks and studies an interesting "meta-mathematical" question about the many solutions to a well known problem (convergent sequence representations of standard constants) by automated literature search and canonicalization…I know of few if any other works which automate such a mathematically interesting analysis.”*
>
> We thank the reviewer for the interest and the positive feedback.
>
> ***
>
> - *“I find the mathematical part of the paper poorly written, explaining the concepts and methods in a very disjointed way.”*
>
> We are sorry to hear. In our revised manuscript, we have compacted the introduction and added a summary of terminology, aiming to create a more continuous description of the mathematical concepts needed in the rest of the paper. Section 2 is supported by this added context. We have also explained where each mathematical term fits in the different stages of unification. The mathematical part of the appendix has been organized for better readability.
>
> ***
>
> - *“The relation to previous work… is unclear, and the invention of new terminology for standard mathematical concepts, such as "conservative matrix field" for a flat connection, is not reassuring.”*
>
> We thank the reviewer for pointing this out. Please note that the Conservative Matrix Field (CMF) terminology is not new, as it was introduced in (Elimelech et al. PNAS 2024), and used in several papers since then. Part of these papers mentioned that CMFs indeed satisfy the defining properties of a discrete flat connection. But importantly, CMFs have more specific properties arising from their domain and rational entries, which are critical for their useful behaviors regarding formula discovery and mathematical convergence. This is why we choose this terminology. We now also mention the relation to flat connections when defining the CMFs for the first time.
>
> In addition, Appendix D now includes a discussion on the broader mathematical contexts of CMFs, and their special properties that distinguish them. Specifically, the cocycle perspective plays a central role in our algorithmic framework for symbolic unification using coboundary equivalence, with the equivalences between trajectories naturally expressed in terms of coboundary transformations.
>
> ***
>
> - *“In section 3 …it is stated that there are 3389 computations of $\pi$ , which are categorized by type as series, continued fraction, or neither, resulting in 1678 formula candidates. How many of these 1678 formula candidates were continued fractions?”
> “What are the other formulas, are they just explicit series with no other apparent structure?”*
>
> 1613 are series vs. 65 continued fractions (Appendix G.4).
> To clarify, 3389 is the number of formulas singled out by GPT-4o mini as pi computing, before being extracted and validated. The other 3389-1678 formulas are mostly false positives from GPT-4o mini’s classification.
> Checking 20 random examples, all were false positives, discarded when double checked by GPT-4o. Example: $g ( x ) = \frac{\pi}{2} + \sum_{j = 1}^{\infty}a_nx^n$ (arXiv 2206.11256).
>
> ***
>
> - *“Then in 3.2…it is said that "Out of the 682 candidates, 407 were validated as formulas and passed on for canonicalization." Where did these numbers come from?”*
>
> After classifying and extracting, GPT-4o produces some expressions that do not compute $\pi$. The PSLQ-validation stage removes these false positives by finding the precise limit of each formula in terms of $\pi$. Of the 682 candidates, 147 are not diophantine formulas (e.g. they contain $\pi$ itself in the series term), leaving 525 formulas. 118 of these did not pass validation. Many of which were formulas that do not compute $\pi$ like $-\frac{4}{\pi}\sum_{n=0}^{\infty}\frac{(-1)^n}{(2n+1)^{2}} = -\frac{4 G}{\pi}$ (arXiv 1906.04927), and formulas with mistakes like $ \sum_{i = 1}^{\infty} \frac{1}{i^2} = \frac{\pi}{6} $ (arXiv 1811.05831, under proofs.tex).
>
> Example for a non-diophantine formula for $\pi$: $\sum_{n = 0}^{\infty} \frac{ ( \frac12 ) _n ( \frac13 ) _n ( \frac23 ) _n}{ ( 1 ) _n^3} ( \frac{2}{27} ) ^n = \frac{3\pi}{4 \, \Gamma^2 ( \frac23 ) \Gamma^2 ( \frac56 ) }$ (arXiv 2001.08104), as $\pi$ can be computed only when given the irrational denominator.
>
> ***
>
> - *“In 3.3 it says "Out of the 407 validated formulas, 402 are found to have 149 representations as order-2 recurrences, and 5 as order-3 recurrences." Is this to say that almost all of the validated formulas were in fact continued fractions?”*
>
> Almost all of the validated formulas are series (364 series vs. 43 continued fractions) because series are the most common forms of formulas being studied in the literature.
>
> Interestingly, a series must be hypergeometric to yield a polynomial recurrence of order 2, shown in a new lemma (Lemma 5). Order-2 recurrences can always be transformed to an equivalent continued fraction, as shown in Lemma 3. So, in total, 402 of 407 formulas were transformed to order-2 recurrences (continued fractions). The other 5 were transformed to order-3 recurrences (these were initially given in a form of series).
>
> ***
>
> - *“The biggest question the results raised in my mind was to what extent the authors believe the "conservative matrix field" approach unifies all sequence representations or not. In the final table 10 many conjectures are stated about such relations. Are there representations of $\pi$ which the authors believe are not instances of the 3d CMF of D.2?”*
>
> We believe that additional formulas from Table 10 may yet hide in the 3d CMF of D.2 (Eq. 6). There are two theoretical limiting factors to unification that may have prevented these from being found so far: (1) UMAPS fit depth and (2) sampling of additional CMF representative formulas.
>
> More generally, to unify all formulas, one would need a “larger” CMF, which will contain the 3d CMF of D.2 and additional dimensions. To see this, note that order-3 recurrences, such as the examples in Appendix A.3, need higher rank CMFs (3x3 matrices).
>
> We are glad to report that we now have a 5d rank-3 CMF that contains our previous 3d rank-2 CMF (Eq. 2). This CMF also contains order-3 recurrences converging to $\pi$.
>
> ***
>
> - *“I suppose the CMF concept could be generalized to Z^D for higher D, is it possible that any recurrence relation could be accommodated this way…”*
>
> Yes! We hope to find CMFs of higher dimension, containing even more unique directions for formula search to accommodate all formulas for $\pi$.
> Our CMF was constructed using the Gauss Hypergeometrics function 2F1(z)=F(a,b;c;z), using its contiguous relations and differential equation. The CMF object was shown in (arXiv 2507.08138) to be generalizable to D-finite functions, like generalized hypergeometric functions - pFqs, using their contiguous relations and annihilators (which serve the same role as the differential equation in our construction). For a pFq CMF, the number matrices (the CMF’s dimension) = p+q (indeed 3 for ours) and the matrix dimension (the CMF’s rank) = max(p, q+1) (2 for ours). CMFs with higher rank are promising for finding deeper recurrences.
>
> The eventual hope is finding one of a few appropriate D-finite functions that can create CMFs that will capture all the formulas of $\pi$, and in a similar way appropriate functions and CMFs for any other constant of interest.
>
> ***
>
> - *“…or are there constraints which prevent combining some pairs of recurrences into the same CMF? The simplest such constraints might be relations between the "dynamical metrics," is this so and where is it mentioned in the text?”*
>
> The most rigid constraint which prevents combining two recurrences is computing separate irrational constants. Separate constants require different CMFs and admission of a formula to a CMF requires consistency between the constants the two compute.
>
> Relations between the dynamical metrics of a formula do not seem to impose constraints given the wide range of dynamical metric values found for formulas generated by the same CMF. To demonstrate this, we have added a dynamical metric “scan” of the CMF (Eq. 6), showing the convergence rate and irrationality measure as a function of direction (on the unit sphere). Fig. 6 and Table 9 also provide multiple dynamical metric values from this CMF, showcasing its versatility.

---

### Official Review · Reviewer_7riT · 2025-07-01

**Clarity:** 2
**Significance:** 1
**Originality:** 1
**Rating:** 2
**Confidence:** 4

**Summary:**

This paper is a step towards automation of mathematical discovery and includes the use of machine learning techniques. The paper focuses on the extraction of specific recurrences that describe the mathematicl constant pi from the mathematical literature.

The method includes the use of scraping, various extraction and validation steps (confusingly some of which are called "retrieval") and an approach to classify and unify the formulas found. Starting from about 278m formulas scraped from arxiv, the formulas are filtered down to 402 that fit the criteria.

I cannot judge whether the recurrences extracted here are interesting to mathematicians in any form. There seem to be some recent papers out there (from the list of references) where mathematicians have explore such formulas. So perhaps there is some interest, but the paper does not explain the significance of this field in mathematics.

As far as I can tell, the only machine learning techniques used are prompting (Appendix F.2 and G.5).

The paper discusses some validation of the methodology in section 4.2, but the "ground truth" is just an unspecified "merging" of the output of 3 language models. The paper does not answer the question whether any of the formulas found bear any resemblance to the original formulas found in papers.

The "retrieval" or "clustering" methods described in in the paper appear to be classical algorithms that are very specific to this kind of recurrence the authors explore. It is unclear to me how these methods generalize to other mathematical discovery methods or whether there is any interesting conclusions for AI research in these methods.

**Questions:**

PSLQ seems to play an important role. It is used to "fix" the formulas extracted by GPT-4o. Could you explain how the process works and how often it is required?

The example in section 5.1 states "This equivalence demonstrates how two previously distinct mathematical expressions, discovered
over a century apart, are now proven to be equivalent by an automated process." To me this reads like an overstatement, as the paper they cite did very explicitly point out the relation to Ramanujan's earlier work. Could you please point out what exactly was discovered here that hasn't been discovered before? It is also okay to state that your method automatically rediscovered some knowledge, which validates the approach.

From how many different papers do your 402 final formulas originate?

**Ethical Concerns:**

["NO or VERY MINOR ethics concerns only"]

**Limitations:**

yes

**Paper Formatting Concerns:**

This paper relies very heavily on the very long appendix. Many methods are only described in the appendix, and in total, the paper is 66 pages long.

**Quality:**

2

**Strengths And Weaknesses:**

Strengths:
- Automatic discovery of mathematical knowledge is a very interesting topic in AI.

Weaknesses:
- It is unclear whether the concrete output of this paper is interesting to mathematicians.
- the paper does seem to contain new methods or insights for AI research
- The paper does not discuss much in terms of validation - How accurate are any of the processing steps? Does the overall pipeline produce the expected formulas?
- The code is not available so I could not dive deeper to clarify the many questions I encountered. The authors promise, however, to release some code in the future.

---

> ### Author Rebuttal · Authors · 2025-07-31
>
> * *“The method includes the use of scraping, various extraction and validation steps (confusingly some of which are called "retrieval")”*
>
> We apologize for the confusion. The retrieval stage refers to collecting equation candidates that match $\pi$-computing patterns from the database of scraped equations. This is now clarified in the text.
>
> ---
>
> * *“I cannot judge whether the recurrences extracted here are interesting to mathematicians…the paper does not explain the significance of this field in mathematics.”
> “It is unclear whether the concrete output of this paper is interesting to mathematicians.”*
>
> The concrete output of this paper is of interest to mathematicians. We can generally classify these significant outputs in 4 types:
>
> 1. **Novel algebraic structure.** The discovery that many historic formulas are generated by a single algebraic structure, the CMF, is mathematically significant. It opens up a new perspective on classical results, connecting them through a shared generative mechanism. The system automatically finds implications of the form FormulaA = LimitA iff FormulaB = LimitB, resulting in a streamlined theory of formula equivalences. The knowledge contained in 381 unique formulas from 151 papers is condensed into 136 canonical forms, which are reduced further into just 32 formulas, proof of which proves all the others systematically, verifiably and automatically.
>
> 2. **Active research on proving formulas for special constants.** For example, Ramanujan-type formulas for $\pi$ have occupied decades of deep mathematical research, with major contributions by Chowla (1928), the Borwein brothers (1987), the Chudnovsky brothers, Zudilin and Guillera. Moreover, a 2025 paper by Cohen and Zudilin raises interest in polynomial continued fraction formulas for constants (arXiv 2501.10090). Many such efforts (e.g., the unification of just two formulas in (Wei et al., Extensions of Ramanujanʼs two formulas for $1/\pi$. Journal of Number Theory, 2013) are regarded as significant advances. Our work achieves results of this kind - including automated proving - on a much larger scale than anything done before.
>
> 3. **A new method in automated theorem proving.** UMAPS itself is of interest for mathematicians as a novel automated proving method.
> For example, since submission UMAPS has proven equivalent entire infinite classes of formulas, such as generalizations of the famous Madhava-Leibniz series, the class of formulas discovered empirically by Raayoni et al. (2021), and classes from (arXiv 1104.0392, p. 5), (1805.06568, p. 3), (1806.08411, p. 6).
>
> 4. **Specific equivalences connecting papers from different fields**, including quantum field theory and string theory, commutative algebra and ring theory, classical analysis and differential equations, operator theory and functional analysis, and probability theory and stochastic processes. Each of the 1021 specific discovered equivalences could be of importance, and should be analyzed independently. Equivalent formulas for constants have in the past revealed deep mathematical connections among special functions, continued fractions, lattice sums, Fourier–theta dualities, modular forms, and Feynman integrals.
>
> ---
>
> * *“…the only machine learning techniques used are prompting”*
>
> In addition to prompting, our clustering algorithm is also a novel contribution to the broader field of machine learning, unique in its combination of symbolic and numerical clustering: Converting to canonical forms unites disparate looking formulas, after which clustering according to the irrationality measure decreases runtime by a factor of 20. Utilizing the transitivity of coboundary equivalence for rigorous clustering during the graph growing algorithm (Appendix B.1) further improves runtime, for a total improvement factor of >200 compared to the no-clustering case. We now expand on this method and its value for efficient formula exploration in Section 3.5.
>
> This work marks the first large-scale application of AI to number theory, and specifically **the first time that a LLM is used in this field** in any form. This achievement is possible due to our integration of LLMs with automated verification and proving. So even though this work looks different from most conventional works in machine learning, it is unique in its **contribution to AI for math**, for which we believe NeurIPS is an ideal platform.
>
> ---
>
> * *“The paper does not answer the question whether any of the formulas found bear any resemblance to the original formulas found in papers.”
> “How accurate are any of the processing steps? Does the overall pipeline produce the expected formulas?”*
>
> Every result in our paper is validated using symbolic computation - without LLMs. Specifically, the formulas found are proven, so every resemblance to an original formula is verified. The LLMs are used for formula harvesting and not for validation. The benchmark used for testing the LLM performance uses as ground truth extracted formulas that passed numerical validation via PSLQ.
>
> Following the referee’s comment here, we revise the manuscript to clarify this important point, now explicitly saying that our methodology provides validation of absolute precision. Once two formulas are found related, they are automatically proven equivalent by our algorithm. This means that 381 of the formulas (found in 151 papers) were proven equivalent to each other in 136 distinct forms, which prove one another in 32 clusters.
>
> ---
>
> * *“The code is not available...”*
>
> Code producing all the final results was available in the supplementary material. We apologize if this was not made clear in the original submission, and now explicitly mention it in the main text as well.
>
> ---
>
> * *“PSLQ seems to play an important role. ...Could you explain how the process works and how often it is required?”*
>
> PSLQ is used during the extraction of every formula. Limits appearing in the equations of origin are sometimes inconsistent with the pure form of the formula collected, due to missing symbolic manipulations (see the example in Appendix G.7 Table 8), so they are not used directly.
>
> The process starts by measuring the convergence rate numerically, then computing the formula to depth 2,000,000 (if slow convergence) or 50,000 (otherwise). This high-precision numerical value is fed into PSLQ, which returns an integer mobius of $\pi$ that equals this numerical limit if one exists. If PSLQ fails, the formula is discarded as it does not compute pi. PSLQ thus acts as both a critical validation step and for complementing the extraction by the LLMs.
>
> This check is critical: without PSLQ, the fraction of incorrectly extracted formulas is on the order of tens of percent.
>
> ---
>
> * *“...how these methods generalize to other mathematical discovery methods or whether there are any interesting conclusions for AI research in these methods.”
> “the paper does not seem to contain new methods or insights for AI research”*
>
> The main insight for AI research our paper promotes is: Discovery that is verifiable and even automatically proven can be achieved by pairing LLMs with new tools from symbolic mathematics. To show this, we have successfully paired LLMs with UMAPS and with PSLQ, resulting in an automated system for equivalence discovery.
>
> Moreover, the approach shown here moves beyond proof-checking and into symbolic structure discovery, which is a frontier direction for AI in scientific reasoning.
>
> Additional contributions to AI research:
> 1. Our work provides a novel dataset to gauge mathematical reasoning: A dataset of hundreds of equivalent formula pairs, useful for benchmarking LLMs on the ability to transfer mathematical knowledge (see Table 2 and Appendix F.2). We tested frontier models and they currently fail in the majority of cases (which are proven by UMAPS), indicating that this dataset will be useful for testing future LLM developments.
> 2. Our framework can apply to other areas of AI for math: We provide an equivalence pipeline for recurrence relations and rational sequences, applicable across number theory and computational math, unmatched in handling millions of expressions in bulk.
> 3. More generally, equivalence detection as developed here could become a crucial capability for certain generative AI use cases, aiding originality verification, data leak detection, and plagiarism identification.
>
> ---
>
> * *“The example in section 5.1 states "This equivalence demonstrates how two previously distinct mathematical expressions, discovered over a century apart, are now proven to be equivalent by an automated process." To me this reads like an overstatement...what exactly was discovered here that hasn't been discovered before?”*
>
> The mentioned paper (Sun 2020) indeed explicitly points out the relation between their 97 new formulas to Ramanujan's earlier work (Ramanujan 1914). However, these new formulas are presented as novel. They are only mentioned as belonging to the same family (having the same overall structure); **not proven to be equivalent to Ramanujan’s**.
>
> Our work discovers and then proves the equivalence of 16 of Sun’s formulas to a total of 134 other formulas, including one of Ramanujan's, in 7 clusters - each one implies the other is correct too. This is what we meant by *“demonstrates how two previously distinct mathematical expressions…are now proven to be equivalent”*.
>
> We revise the introduction to clarify what was known and what was discovered here, and also elaborate on this in Section 3.5 where UMAPS is discussed, and in Section 5.
>
> ---
>
> * *“From how many different papers do your 402 final formulas originate?”*
>
> 154 papers (see Table 4 in p. 8). This is now better clarified in Section 5.
> Note that all 154 are included in the supplementary material.

---

### Official Review · Reviewer_xauJ · 2025-07-01

**Clarity:** 2
**Significance:** 2
**Originality:** 3
**Rating:** 3
**Confidence:** 4

**Summary:**

The paper introduces an end-to-end, LLM-based pipeline that tries to unify the many disparate formulas that mathematicians and physicists have discovered for fundamental constants such as $\pi$, e, $\xi(3)$, and Catalan’s constant. The work can be categorized as AI for Experimental Mathematics.

Within the study 455k arXiv papers are scraped using LLMs. The LLM filters strings that seem to compute the target constant and rewrites the LaTeX into SymPy code. A code/LLM-feedback loop plus a PSLQ check keeps only numerically correct formulas.
Then, each formula is converted to the minimal polynomial recurrence (via the RISC guesser). Recurrences with identical minimal polynomials are collapsed; dynamical fingerprints, namely the irrationality measure and the exponential convergence rate, cluster the rest. The authors use a theoretical machinery that is called the Conservative Matrix Field (CMF), a 3D lattice of 2 × 2 rational matrices whose paths encode families of equivalent formulas. A new algorithm, UMAPS, searches for and proves coboundary equivalence between two recurrences by empirically fitting a polynomial Möbius map.

As a result,  for $\pi$, 381 of 407 formulas are proved linked; 188 collapse into a single CMF. Famous identities of Euler, Gauss, Brouncker and several modern machine-generated series are shown to be equivalent. Limited experiments show the approach extends to other constants. An interesting obcservation from the paper is that the current LLMs very rarely find such equivalences.

**Questions:**

Please refer to the six weaknesses listed above. In particular, addressing the the following convincingly would raise my score.
1. Can you prove (or empirically demonstrate on an exhaustive stress-test) that the UMAPS coboundary solver is sufficient, i.e. it finds a transformation whenever two recurrences are truly coboundary-equivalent, and that the solution is unique?
2. Please provide a sensitivity study for the key empirical thresholds: $\delta$-bin size & similarity window, fitting depth N, polynomial-degree cap, and the 200-term limit in recurrence guessing. Ideally plot linkage percentage and runtime versus each parameter.
3. Can you prove that the guessed recurrence for each formula is minimal and globally valid, or else report the empirical error rate when minimality fails? If rational rather than polynomial coefficients arise, how do you detect and handle order inflation?
4. Please formalise CMF for rational entries. It would be great to provide a proof of existence for any d-finite constant, clarify whether rank is unique, and give conditions under which every lattice direction corresponds to a formula.

**Ethical Concerns:**

["NO or VERY MINOR ethics concerns only"]

**Final Justification:**

I maintain my score, but would like to reduce my confidence to 4. I think the paper requires a significant rewrite of new content and a much deeper re-evaluation of the formal parts.

**Limitations:**

I believe the authors have not sufficiently addressed the study’s limitations. I recommend adding a concise section in the paper that explicitly summarizes the key methodological gaps, such as hyper-parameter sensitivity, incomplete completeness guarantees, reliance on empirically inferred minimality, and questions of scalability and domain transfer to other areas of symbolic computation and formula harvesting.

**Paper Formatting Concerns:**

Nothing of concern from the formatting perspective.

**Quality:**

1

**Strengths And Weaknesses:**

# Strengths:
**Significance:** The paper addresses an important gap in AI-guided experimental mathematics: connecting many scattered symbolic formulas. While automated theorem proving and conjecture generation have seen progress, the systematic, large-scale unification of symbolic formulas is an unexplored frontier. The work's idea is indeed of significance and its success points toward a new paradigm in how mathematical knowledge can be structured. I believe this could benefit computer-assisted mathematics, OEIS-style databases, and AI-driven discovery.

**Originality:** I think the proposed framework is original. It combines state-of-the-art LLMs for knowledge extraction with deep, novel symbolic algorithms. The CMF idea along with the empirical coboundary solver are new and non-trivial. I think this is the largest automated “formula harvesting’’ study to date.

**Clarity:** Complex concepts are introduced with clarity and supported by well-chosen examples and figures. The appendices are indeed providing the necessary details for a deep understanding of the main text.

**Quality:** The methodology and pipeline and the claims are well-supported by the experimental results. The mathematical foundations are solid, and the appendices provide detailed proofs for the key lemmas that elaborate on the UMAPS algorithm. I have reviewed the proofs of Lemma 1, 2, and 3 and find them to be correct.

# Weaknesses:
1. Lemma 1 proves only a necessary condition for coboundary equivalence. No theorem/lemma is given that the empirical fitting procedure (steps 1-6 in Appendix B.3) will always find a coboundary when one exists, nor that a solution returned is unique. In other words, the paper does not provide a corresponding sufficiency proof or a guarantee of completeness.
2. As far as I understood, the pipeline incorporates several ad-hoc choices, e.g. $\delta$-clustering granularity 0.05 and similarity window 0.03, Maximum fitting depth N=40 and unspecified polynomial-degree caps in UMAPS, and the use of N=200 partial sums to guess recurrences with the RISC tool. Then, no sensitivity study is provided. So it is unclear how robust results are to these (hyper)parameters. For example, the $\delta$ threshold controls which pairs ever reach the (costly) UMAPS stage, too tight a window could miss valid links, while too loose one inflates runtime.
3. Converting every formula to a minimal polynomial recurrence (canonical form) is an important part of the paper. Subsequent equivalence proofs assume minimality, yet minimality is inferred empirically: RISC is run on the first 200 terms, then the candidate recurrence is “shown to obey” the series. Do we know formally that the guessed recurrence is indeed minimal (or even correct) for all n? Consequently, I think for constants whose true minimal recurrence has rational rather than polynomial coefficients, the inflation procedure may raise the order or degree (a common issue in computer algebra) and break minimality.
4. Again, as far as I can tell, the authors extend the CMF from polynomial to rational entries ﻿but does not prove it's existence, uniqueness of rank for a given constant, or whether every direction indeed corresponds to a formula. The analogy with a conservative vector field is appealing yet informal. I think an algebraic geometric treatment is missing.
5. I totally understand how challenging it is to write an AI for Mathematics paper, given the extensive cross-referencing between the main text and the appendices, but I think the exposition could still be made clearer. In particular, consider a short introduction of technical terms (that are heavily relied on Appendix at the moment) and making the introduction less repetitive, and avoiding tautology.
6. The manuscript also lacks discussion on some important related studies that bear directly on its core tasks of harvesting, discovering and clustering identities and conjectures, such as:
- Paule & Schneider (JSC 2024), Creative Telescoping for Hypergeometric Double Sums
- Hashemi et al. (ICLR 2025) Can Transformers do Enumerative Geometry?
- Jiang et al. (2025) Enhancing Deep Symbolic Regression via Reasoning Equivalent Expressions
- Tian et al., (Nat. Commun. 2025) Interactive Symbolic Regression with a Co-design Mechanism
- Romera-Paredes et al., (Nature 2024) Mathematical discoveries from program search with large language models
- Studies on Symbolic computation on Knowledge Graph and relational databases

I think commenting on these studies would situate the contribution within the established AI-guided symbolic-computation literature, and strengthen claims of soundness and completeness. I recommend discussing their relation to the current study.

---

> ### Author Rebuttal · Authors · 2025-07-31
>
> * *“The paper addresses an important gap in AI-guided experimental mathematics … The work's idea is indeed of significance and its success points toward a new paradigm in how mathematical knowledge can be structured.”
> “The methodology and pipeline and the claims are well-supported by the experimental results. The mathematical foundations are solid…”*
>
> We thank the reviewer for their interest and appreciate their positive feedback.
>
> ---
>
> * *“1. Can you prove…that the UMAPS coboundary solver is sufficient...and that the solution is unique?”*
>
> Lemma 1 implies UMAPS is sufficient: if two formulas are coboundary then their limits are related by $U(1)$. So during the application of UMAPS, $U(1),…,U(N)$ will be correct measurements of $U(n)$, and thus the fit will work provided N is large enough. Following this comment, sufficiency is now highlighted in Section 3.5 by renaming Lemma 1 to “sufficiency of UMAPS” and by including the short corollary. Lemma 2 generalizes Lemma 1 and shows sufficiency for general order recurrences.
>
> Besides sufficiency, we now show in a new lemma (Lemma 4) the uniqueness of the coboundary matrix for the $2\times 2$ case for constants not algebraic of degree 2. The idea: assume $U1(n), U2(n)$ are two coboundary matrices between $A(n),B(n)$ $\Rightarrow A(n) = U1(n) B(n) U1(n+1)^{-1} = U2(n) B(n) U2(n+1)^{-1}$. So $B(n)$ is coboundary to itself with $U3(n) = U1(n)^{-1}U2(n)$. Define $L_{B_k} = \prod_{n=k}^\infty B(n)$, the limit of the recurrence when started at index $k$. We have from Lemma 1 $L_{B_k} = U3(k)( L_{B_k})$. Since $L_{B_k}$ is not algebraic of degree 2 and $U3(k)$ acts as a rational mobius, we conclude $U3(k) = I_{2 \times 2}$ for any k. Therefore $U1(n) = U2(n)$ for all $n$.
>
> ---
>
> * *“…the pipeline incorporates several ad-hoc choices, e.g $\delta$-clustering granularity 0.05 and similarity window 0.03, Maximum fitting depth N=40 and unspecified polynomial-degree caps in UMAPS, and the use of N=200 partial sums to guess recurrences with the RISC tool…no sensitivity study is provided.”
> “2. Please provide a sensitivity study…Ideally plot linkage percentage and runtime versus each parameter.”*
>
> We now provide a sensitivity study showing the linkage percentage and runtime for different $\delta$-clustering granularities and similarity thresholds, UMAPS’s fit depth, as well as an analysis of the sensitivity of RISC’s Guess algorithm to its fit depth. All of these have been added to the appendix and are summarized below (parameters used in the paper are denoted in bold). The polynomial-degree caps for UMAPS are derived from the fit depth $N$: the degree “budget” is split equally between the numerator and denominator polynomials, for a degree cap of $\text{floor}(N / 2) - 1$. When the UMAPS fit depth is raised to N=120 for an allowed max degree of 59, no new connections are found.
>
> **Sensitivity study:**
>
> **Linkage percentage** (% of max - 381) **for different $\delta$ clustering granularities and similarity thresholds.**
> | Similarity \ Granularity  | 0.01 | 0.05 | 0.1 | 0.2 | 0.3 | 0.5 |
> |-|-|-|-|-|-|-|
> | 0.001 | 84   | 79   | 63  | 50  | 46  | 59  |
> | 0.003 | 92   | 80   | 65  | 52  | 48  | 60  |
> | 0.005 | 100  | 83   | 67  | 52  | 48  | 60  |
> | 0.01  | 100  | 95   | 73  | 52  | 49  | 61  |
> | 0.02  | 100  | 100  | 78  | 52  | 54  | 61  |
> | **0.03**  | 100  | **100**  | 78  | 52  | 54  | 61  |
> | 0.05  | 100  | 100  | 100 | 70  | 71  | 61  |
> | 0.1   | 100  | 100  | 100 | 100 | 91  | 61  |
> | 0.2   | 100  | 100  | 100 | 100 | 100 | 78  |
> | 0.3   | 100  | 100  | 100 | 100 | 100 | 100 |
> | 0.5   | 100  | 100  | 100 | 100 | 100 | 100 |
>
> Linkage percentage with no clustering criterion (similarity threshold = 2): 100% (381).
>
> **Runtime table for different $\delta$ clustering granularities and similarity thresholds.** The algorithm (Appendix B.1) first matches formulas to find a set of “hubs” (stage 1) and then matches each hub with CMF representatives (stage 2). For coarse granularities (>0.1), the runtime shows 3 phases in its dependence on similarity threshold, delineated via the split into regular text, bold, then regular text (right). This can be explained by the number of hubs from stage 1 (decrease as similarity threshold increases) vs. the number of CMF representatives that meet the similarity criterion in stage 2 (increase as similarity threshold increases). The runtime value results from their product, which is largest for medium similarity threshold values.
>
> | Similarity \ Granularity | 0.01 | 0.05    | 0.1 | 0.2 | 0.3 | 0.5 |
> |-|-|-|-|-|-|-|
> | 0.001   | 0.8 | 0.6 | 0.8 | 0.8 | 0.7 | 0.7 |
> | 0.003   | 0.9 | 0.8 | 1.1 | 1.1 | 1.1 | 1.0 |
> | 0.005   | 1.1 | 0.9 | 1.3 | 1.2 | 1.3 | 1.2 |
> | 0.01     | 1.0 | 0.7  | 1.2 | **2.0** | **2.0** | 1.3 |
> | 0.02       | 1.0  | 1.0 | 1.5 | **3.4** | **3.5** | **1.6** |
> | **0.03**  | 1.0  | **1.0** | 1.4 | **4.7** | **4.9** | **2.4** |
> | 0.05    | 1.0 | 1.0 | 1.0 | **4.0** | **3.9** | **3.2** |
> | 0.1      | 1.0 | 1.0 | 0.9 | 1.0 | **2.9** | **5.4** |
> | 0.2      | 0.9 | 0.9 | 0.9 | 0.9 | 0.9 | **6.1** |
> | 0.3      | 1.0 | 0.9 | 0.9 | 1.0 | 0.9 | 1.0 |
> | 0.5      | 1.0 | 0.9 | 1.1 | 1.0 | 0.9 | 0.9 |
>
> Runtime with no clustering criterion (similarity threshold = 2): 20.0. This highlights the role of clustering in reducing runtime.
>
> **Linkage rate and runtimes for different maximum fitting depths in UMAPS.**
> | UMAPS N | Found relation (x / 407) | Same CMF (x / 407) | Max poly. deg. found | Avg. UMAPS time | Total runtime |
> |-|-|-|-|-|-|
> | 4| 96| 96 | 1| 0.5| 6.7 |
> | 8 | 141| 136| 3| 0.5| 3.7 |
> | 12| 200| 170| 5| 0.7| 3.1 |
> | 16| 301| 185| 7| 0.8| 2.5 |
> | 20| 373| 188| 9| 0.9| 1.2 |
> | 24| 375| 188| 11 | 0.9| 1.0 |
> | 29| 381| 188| 13 | 0.8| 0.8 |
> | 30| 381| 188| 14 | 1.1| 0.9 |
> | **40**| **381**| **188**| **14** | **1.0**| **1.0** |
> | 50| 381| 188| 14 | 1.0| 1.3 |
> | 65| 381| 188| 14 | 1.1| 2.9 |
> | 80| 381| 188| 14 | 1.7| 9.7 |
> | 100 | 381| 188| 14 | 7.1| 48.8|
> | 120 | 381| 188| 14 | 18.3 | 137.3 |
>
> The total runtime has a clear minimum. The two forces at play: Avg. UMAPS time (increases as N increases) and number of hubs from stage 1 of the algorithm (decreases as N increases).
>
> **Success rate and runtime of RISC’s Guess for different fit depths N over 100 random formulas.**
> | Guess N | Recurrences found (x / 100) | Of which correct | Avg. runtime |
> |-|-|-|-|
> |8 | 0 |0 |0.02 |
> |10 |24 | 24 |0.05 |
> |15 |30 | 30 |0.08 |
> |20 |57 | 57 |0.18 |
> |25 |76 | 76 |0.26 |
> |30 |92 | 92 |0.31 |
> |50 |99 | 99 |0.46 |
> | 100 | 100 |100 |0.67 |
> | **200** | **100** | **100** | **1.00** |
> | 500 | 100 |100 |2.79 |
>
> These findings indicate that Guess generates robust fits that generalize well.
>
> ---
>
> * *“3. Can you prove that the guessed recurrence for each formula is minimal and globally valid…”*
>
> Yes. Correctness is proven automatically by substituting the formula of origin into the resulting recurrence. When symbolic simplification is too complicated, 30 random numerical substitutions validate the expression is identically zero: all substitutions must be under 1e-10 to pass this validation. All recurrences are successfully validated this way.
>
> Regarding minimality, it is not guaranteed by the Guess algorithm. However, most formulas found are of order 2 and cannot be order 1 which have a very unique structure and growth. For higher orders, the canonicalization stage is indeed only as good as RISC’s Guess. A new lemma (Lemma 5) now shows that a series must be hypergeometric to yield a polynomial recurrence of order 2, showing the 5 recurrences of order 3 are minimal as well. All instances of known minimal recurrences were successfully found by Guess.
>
> ---
>
> * *“3. If rational rather than polynomial coefficients arise, how do you detect and handle order inflation?”*
>
> The inflation procedure we conduct (described in Appendix C.2) converts the recurrence coefficients to integer polynomials without changing the recurrence order. Minimality is hence preserved.
>
> ---
>
> * *“4. formalize CMF for rational entries…provide a proof of existence for any d-finite constant, clarify whether rank is unique, and give conditions under which every lattice direction corresponds to a formula”*
>
> The CMF is formalized for rational entries in Appendix D. The $\pi$ CMF (Eq. 6) is symbolically shown to satisfy the conserving conditions for the variables $x,y,z$ (above Eq. 6) and thus proven a CMF. This CMF is constructed from the Gauss Hypergeometric function $2F1(z)=F(a,b;c;z)$ using its contiguous relations and differential equation, a method applicable broadly to parameterized D-finite functions like pFqs (discussed in a recent arXiv paper 2507.08138).
>
> To show the uniqueness of the rank for a pFq CMF: the number of matrices (dimension) = p+q (3 for ours) and the matrix dimension (rank) = max(p, q+1) (2 for ours). CMFs with higher rank are promising for finding deeper recurrences.
>
> Regarding correspondence of lattice directions to formulas, trajectories starting $(x=a,y=b,z=c)$ converge to the value $F(a,b;c;1/2)/F’(a,b;c;1/2)$ where $F’ = z \cdot d/dz F$, now included as Lemma 6 in Appendix D.2. The corresponding formula does depend on the direction.
>
> ---
>
> * *“consider a short introduction of technical terms…and making the introduction less repetitive”
> “...lacks discussion on some important related studies…”
> “I recommend adding a concise section in the paper that explicitly summarizes the key methodological gaps, such as hyper-parameter sensitivity…reliance on empirically inferred minimality, and questions of scalability and domain transfer to other areas of symbolic computation and formula harvesting.”*
>
> Thank you for these suggestions. We have now included a brief summary on terms used, and compacted the intro. To the discussion section we added a paragraph about the empirically inferred minimality and how we can prove it, as well as text on scalability and domain transfer. We now also cite the works listed by the reviewer and discuss their relation to this work.

---

> > ### Comment · Reviewer_xauJ · 2025-08-04
> > **Official Reply**
> >
> > - The authors argue that Lemma 1 already implies sufficiency because, given a true coboundary, the measured sequence $U(1),\dots,U(N)$ “will be correct” once $N$ is “large enough. I think this argument is still on existence. It does not bound the required $N$ or show that the least-squares fit used in practice converges uniquely to the true rational function under noise or finite precision.  Therefore the pipeline may still miss valid links for small/medium $N$. A formal statement such as “if degree ≤ d then $N\ge 2d+1$ suffices” is still missing.
> >
> > - Hyper-parameter sensitivity study : A strange result is that for large thresholds almost every parameter combination yields 100 % linkage, suggesting the clustering pass is sometimes redundant! I think this should be discussed and properly addressed.
> >
> > - The authors agree that minimality is not guaranteed by Guess. Lemma 5 merely shows that an order-2 polynomial-coefficient recurrence implies the series is hypergeometric; it does not prove the converse or cover rational coefficients. Thus minimality for order > 2 remains empirical. I am partially satisfied here. The correctness is addressed, but I think minimality is still an open (though mitigated) issue for higher orders and rational-coefficient cases.
> >
> > - Showing that the specific $\pi$ CMF arises from contiguous relations of ${}_2F_1$ is useful but does **not** constitute a general existence theorem for arbitrary d-finite constants. Lemma 6 gives a convergence target involving $F/F'$.  It is unclear how this generalises beyond ${}_2F_1$ or ensures every lattice path corresponds to a known symbolic formula.               The formalism now covers one important class but not the general CMF claims made in the introduction. I think the manuscript should narrow its scope or supply a broader argument.
> >
> > **My thoughts so far:** The paper's introduction and abstract promise a general framework for unifying mathematical knowledge. The CMF is presented as this general structure. However, the theoretical justification provided (including rebuttal) is proper only to a specific class of special functions. It is the authors' responsibility to either (a) provide the general theory that underpins their general claims or (b) narrow their claims to match the scope of their theory.

---

> ### Author Response · Authors · 2025-08-06
> **Comments 1 and 2: Bound on N and Hyper-parameter Sensitivity Study**
>
> We appreciate the reviewer’s careful inspection and thank them for their continued engagement.
>
> ---
>
> - Comment 1.1. *“Lemma 1 … does not … show that the least-squares fit used in practice converges uniquely to the true rational function under noise or finite precision.”*
>
> We thank the reviewer for raising this point and apologize for the misunderstanding.
> There is no least-squares fit, so noise and finite precision are not sources of error. Analytical correctness of the fit is guaranteed by subsequent substitution into the coboundary condition, which also shows $U$’s projective uniqueness via Lemma 4. We now further stress this point in the manuscript.
>
> **Details**:
> When fitting rational functions, the equations are of the form $\frac{P(i)}{Q(i)} = \frac{p_i}{q_i}$, where $P,Q$ are the polynomials of the rational function and $\frac{p_i}{q_i}$ is the empirical value at $i$. This leads to equations of the form $q_i \sum_{k=0}^{deg_P} P_k i^k - p_i \sum_{k=0}^{deg_Q} Q_k i^k = 0$, for the $deg_P + deg_Q + 2$ polynomial coefficients. These coefficients are determined by finding the nullspace of the system of equations - in practice a SymPy matrix of integers (e.g. $q_i i^k$ for $P_k$). SymPy does this via the linear system’s reduced row-echelon form, and uses rational number objects (not floats), so precision is guaranteed.
>
> ---
>
> - Comment 1.2. *“Lemma 1 … does not bound the required $N$ ” “A formal statement such as “if degree ≤ d then $N \geq 2d + 1$ suffices” is still missing.”*
>
> The formal statement for the fit in UMAPS is precisely as suggested by the reviewer:
> If a coboundary matrix exists and every rational-function entry has degree at most $d$, then running UMAPS with $N \geq 2d + 1$ suffices to recover the coboundary matrix. This is now stated in Section 3.5 and we cite Ch. 2.2 of (Stoer and Bulirsch. Introduction to Numerical Analysis 3rd ed., 2002).
>
> (Lemma 1 does not bound the required $N$, as it describes the Mobius relation between the limits of two matrices that are coboundary.)
>
> **Details**:
> Letting the polynomial degrees of the rational function be at most $d$, we have $2d+2$ variables. Scaling the numerator and denominator by a factor results in the same rational function, so there are $2d+1$ true degrees of freedom. Using $N = 2d+1$, we obtain a linear homogeneous system of $2d+1$ equations. This system will have a unique solution up to scale if a coboundary matrix exists (1d nullspace). Taking $N \geq 2d + 1$ adds more linearly dependent equations, maintaining the same nullspace.
>
> The broader question of  bounding $d$ for two general recurrences $A(n), B(n)$ is an open research topic.
> We now make this distinction between cases more explicit in the manuscript.
>
> Our work advances this active research by 1) supplying a novel tool for equivalence detection and rigorous proof and 2) finding 1021 coboundary equivalence examples among preexisting formulas for $\pi$, providing a testbed for coboundary matrix properties as well as equivalence between formulas in general.
>
> ---
>
> - Comment 2. *“Hyper-parameter sensitivity study: A strange result is that for large thresholds almost every parameter combination yields 100 % linkage, suggesting the clustering pass is sometimes redundant!”*
>
> This behavior is actually expected. Larger similarity thresholds imply more **lenient** clustering. As the similarity threshold is made less strict, more formula pairs obey the similarity criterion regardless of their parameters. Then UMAPS is applied to many more formula pairs, most of which fail. The only disadvantage here is that UMAPS is computationally heavy, and thus using a large threshold makes the overall algorithm slower.
>
> In principle, it is possible to completely skip clustering and just apply UMAPS on all formula pairs. The algorithm is then 20 times slower for the test case we presented.
> The main goal of the $\delta$-clustering stage is practical: to make the graph growing algorithm (Appendix B.1) more efficient.

---

> > ### Author Response · Authors · 2025-08-06
> > **Comment 3: Recurrence Minimality**
> >
> > - Comment 3. *“Lemma 5 … does not prove the converse or cover rational coefficients. … The correctness is addressed, but I think minimality is still an open (though mitigated) issue for higher orders and rational-coefficient cases.”*
> >
> > We now utilize Maple’s **MinimalRecurrence** tool (from the LREtools package) to **prove minimality** for all recurrences found by RISC’s Guess. The function receives as input initial conditions and a linear recurrence (polynomial or rational coefficients) and “computes the recurrence of provably minimal order that holds for all but finitely many terms.” "Assuming the input R is correct, the output MinRec will be the recurrence with proved minimal order." - from Maple’s documentation. Additional details in Ch. 6 of (Zhou, Algorithms for Factoring Linear Recurrence Relations. 2022. Doctoral dissertation, Florida State University).
> >
> > This function is now applied to each recurrence found by Guess in our pipeline. When applied to recurrences found in our original dataset, all cases have returned the same recurrence as the inputted recurrence, showing their minimality.
> >
> > The part of the pipeline converting formulas to recurrences now achieves **provable correctness and minimality**. We thank the reviewer for maintaining the importance of this topic and for encouraging us to find and implement this improvement.
> >
> > **Regarding Lemma 5**, we apologize for the misunderstanding - it does indeed prove the converse and covers rational coefficients.
> >
> > Lemma 5: A series with term s(n) and resulting partial sum x(n) obeys the order-2 recurrence $x(n) = (\frac{s(n)}{s(n-1)} + 1) x(n-1) - \frac{s(n)}{s(n-1)} x(n-2)$.
> >
> > Proof: given $x(n) = \sum_{k=n_0}^n s(k)$ we have $x(n) - x(n-1) = s(n)$, so
> > $\frac{x(n) - x(n-1)}{s(n)} = 1 = \frac{x(n-1) - x(n-2)}{s(n-1)}$. By moving around terms, we arrive at the recurrence above.
> >
> > Corollary: a series yields an order-2 **rational** (polynomial, using our inflation) coefficient recurrence if and only if it is hypergeometric.
> >
> > Proof: By definition, a series is hypergeometric if and only if $\frac{s(n)}{s(n-1)}$ is rational, so the above recurrence is rational if and only if the series is hypergeometric.
> >
> > So we show the converse is true, and in fact we have a closed-form method for finding the recurrences for series. The reasons for using Guess despite this achievement are two-fold:
> > 1. Guess finds recurrences numerically using hundreds of terms, stronger in general than symbolic manipulations that can fail to identify whether a series is hypergeometric because of incomplete simplification of  $\frac{s(n)}{s(n-1)}$.
> > 2. Guess finds polynomial recurrences for non-hypergeometric cases, i.e. is not restricted to second-order polynomial recurrences.

---

> > > ### Author Response · Authors · 2025-08-06
> > > **Comment 4 and Summary: D-finite CMFs and General Theory**
> > >
> > > - Comment 4.1. *“Showing that the specific $\pi$ CMF arises from contiguous relations of $_2F_1$ is useful but does not constitute a general existence theorem for arbitrary d-finite constants.”*
> > >
> > > An existence theorem for general D-finite functions is available in the recent arXiv paper 2507.08138, which we now cite. The proof is by construction, using contiguous relations and annihilators to arrive at a set of matrices satisfying the conservative property.
> > >
> > > We hypothesize that any constant which can be obtained as a value of such D-finite functions has its own CMF, which is a topic for future research. Lemma 6 shows conditions under which a CMF constructed from a D-finite function F, holds approximations to special values of the function F and its derivatives.
> > >
> > > ---
> > >
> > > - Comment 4.2. *“Lemma 6 gives a convergence target involving $F/F’$. It is unclear how this generalises beyond $_2F_1$ or ensures every lattice path corresponds to a known symbolic formula. The formalism now covers one important class but not the general CMF claims made in the introduction. I think the manuscript should narrow its scope or supply a broader argument.”*
> > >
> > > We can supply a broader argument and we will now make this revision to the manuscript. Let us explain how this is possible:
> > > We generalize Lemma 6 to include the convergence scheme for general D-finite-derived CMFs, and shift the original Lemma 6 to be a corollary.
> > > The general statement is that every trajectory in a CMF converges to the ratio of the derivatives of the D-finite function.
> > >
> > > The idea:
> > >
> > > Let F be a D-finite function with $n$-discrete parameters satisfying a differential equation of order $r$. The recent arXiv paper 2507.08138 shows how to build a degree $n$ rank $r$ CMF from F.
> > >
> > > In this setting, the generalized Lemma 6 provides conditions on the CMF trajectory such that, for any initial point $(a1,…,an)$, the associated trajectory matrix yields a Diophantine approximation to each of the values $F′/F,F′′/F,…,F^{(r−1)}/F $ evaluated at that point, and therefore leads to Diophantine approximations of any rational function in $(F,F’,...)$ over the point, such as $(F+F’’)/F’$.
> > >
> > > Remarkably, the convergence of the sequence does not depend on the specific value for F at the initial point $(a_1,...,a_n)$, only on the direction of the trajectory. This means that by just changing the evaluation point $(a_1,...,a_n)$, the same CMF structure provides approximations to a wide class of constants.
> > >
> > > For instance, in our  $_2​F_1$​-based CMF, changing the initial point from $(1/2,1/2,1/2)$ to integer values like $(1,1,−1)$ produces a 3-dimensional, rank-2 CMF that converges to $\log⁡(2)$. This also works for larger CMFs. Moreover, the convergence rate and structure remain independent of the initial point, so the CMF encodes simultaneous information about many constants at once.
> > >
> > > ---
> > >
> > > - Summary. “***My thoughts so far:*** *The paper's introduction and abstract promise a general framework for unifying mathematical knowledge. The CMF is presented as this general structure. However, the theoretical justification provided (including rebuttal) is proper only to a specific class of special functions. It is the authors' responsibility to either (a) provide the general theory that underpins their general claims or (b) narrow their claims to match the scope of their theory.”*
> > >
> > > We have now implemented multiple changes to provide the theoretical proofs and citations that underpin our general claims:
> > > 1. A bound on UMAPS $N$ for successfully finding a coboundary matrix is provided.
> > > 2. We now prove recurrence minimality in addition to correctness as part of the pipeline.
> > > 3. We cite the existence theorem for CMFs derived from general D-finite functions, and
> > > 4. generalize Lemma 6 to describe convergence to constants for these CMFs, supporting that unification via CMFs is applicable to many constants that have representations as evaluations of D-finite functions.
> > >
> > > At the same time, we now state more clearly which properties are empirically observed, and which claims are conjectures, not yet proven.

---

### Note · Authors · 2025-08-16

We thank the reviewers again for their careful and constructive feedback, which inspired several new results and improvements, now included in our paper.

We have better positioned the work within the emerging paradigm of LLM–symbolic reasoning integration, similar in spirit to *PAL* (NeurIPS 2022), *Parsel* and *DELI* (NeurIPS 2023), and *Self-Discover* (NeurIPS 2024), but applied to number theory for the first time. We have also clarified the significance of unifying mathematical formulas in the context of AI for mathematics, situating our contribution alongside related efforts such as *Romera-Paredes et al.* (Nature 2024), *Tian et al.* (Nat. Commun. 2025), *Hashemi et al.* (ICLR 2025), and *Paule & Schneider* (JSC 2024).

We have incorporated all edits promised into the revised manuscript. We appreciate the reviewers’ engagement and interest in our application of AI to mathematics.

---

### Decision · Program_Chairs · 2025-09-17

**Decision:**

Accept (poster)

**Comment:**

The paper solves a very interesting and important question: "How are discovered mathematical formulas connected?". The paper relies on LLMs and various mathematical techniques to extract various representations of approximations for, e.g., for $\pi$ and $e$, and convert them into standard forms, after which unification of formulas can be performed. Several relations between different representations are found. For example, for $\pi$, 381 of 407 formulas are proved linked and several well-known representations (e.g., from Euler, Gauss, and Brouncker) are shown to be equivalent.

The paper is well motivated, and the studied problem is very novel/original. Most reviewers (and also I myself, after reading the paper and the discussions) think the work is quite impactful for mathematical research.  The decision on this paper is especially difficult, as despite its novelty and contributions, its writing and analysis could be further improved, which I believe the authors have done a reasonably good job in the discussions with the reviewers. After discussions between the reviewers/ACs and the ACs/SACs, I think this is a good paper related to using LLMs for mathematical explorations, thus a recommendation for its acceptance. I strongly recommend that the authors consider the comments during rebuttal/discussions to revise the paper accordingly in its final version.